# It's Hard to Be Normal: The Impact of Noise on Structure-agnostic Estimation

**Jikai Jin**\*
Stanford University
jkjin@stanford.edu

**Lester Mackey**
Microsoft Research
lmackey@microsoft.com

**Vasilis Syrgkanis**†
Stanford University
vsyrgk@stanford.edu

## Abstract

Structure-agnostic causal inference studies how well one can estimate a treatment effect given black-box machine learning estimates of nuisance functions (like the impact of confounders on treatment and outcomes). Here, we find that the answer depends in a surprising way on the distribution of the treatment noise. Focusing on the partially linear model of Robinson [1988], we first show that the widely adopted double machine learning (DML) estimator is minimax rate-optimal for Gaussian treatment noise, resolving an open problem of Mackey et al. [2018]. Meanwhile, for independent non-Gaussian treatment noise, we show that DML is always suboptimal by constructing new practical procedures with higher-order robustness to nuisance errors. These *ACE* procedures use structure-agnostic cumulant estimators to achieve $r$-th order insensitivity to nuisance errors whenever the $(r + 1)$-st treatment cumulant is non-zero. We complement these core results with novel minimax guarantees for binary treatments in the partially linear model. Finally, using synthetic demand estimation experiments, we demonstrate the practical benefits of our higher-order robust estimators.

## 1 Introduction

Modern machine learning (ML) offers a rich toolbox of flexible methods for modeling complex, high-dimensional functions — ranging from regularized linear regression [Belloni et al., 2014, Zou and Hastie, 2005] and random forests [Breiman, 2001, Biau et al., 2008, Syrgkanis and Zampetakis, 2020] to neural networks [Schmidt-Hieber, 2020, Farrell et al., 2021] and hybrid combinations of these techniques [Džeroski and Ženko, 2004, Chernozhukov et al., 2022]. For statisticians and econometricians, it is natural to ask whether ML can improve both the accuracy and the robustness of estimating a target parameter of interest. Recently, Balakrishnan et al. [2023] introduced the paradigm of structure-agnostic estimation (SAE). SAE enables parameter inference by directly plugging in black-box ML-based estimates of the so-called nuisance functions — the other regression components that impact our observations while not being of primary inferential interest. This framework characterizes the best possible target estimation accuracy in terms of the error of these nuisance estimates. By contrast, the classical semiparametric approach derives optimal error rates under explicit structural assumptions on the nuisance — such as smoothness or sparsity — which must be exploited by the estimator and can render it fragile when those assumptions fail [Stone, 1982, Chen and White, 1999, Belloni et al., 2011].

In the field of causal inference, a typical goal is to estimate the impact of a treatment on an observed outcome in the presence of confounders $X$ that impact both treatment $T$ and outcome $Y$ in largely unknown ways. When all confounders are observed, many causal estimands admit expressions in

---

\*Jikai Jin was supported by NSF Award IIS-2337916.
†Vasilis Syrgkanis was supported by NSF Award IIS-2337916.

terms of regression functions that are themselves estimable from data. Consequently, causal parameter estimation falls squarely within the regime where ML-based nuisance estimates may provide benefits.

Recently, double/debiased machine learning (DML) [Chernozhukov et al., 2018, 2022] was proposed as an efficient way to estimate causal parameters from black-box nuisance estimates. Specifically, given $n$ independent and identically distributed (i.i.d.) observations and nuisance estimates with mean squared error $\epsilon$, these methods achieve $\mathcal{O}(\epsilon^2 + n^{-1/2})$ error rates, improving over the $\mathcal{O}(\epsilon + n^{-1/2})$ rate achieved by naively plugging in the nuisance estimates. In particular, the estimates are $\sqrt{n}$-consistent even when the nuisance estimates converge as slowly as $n^{-1/4}$. Moreover, this rate is structure-agnostic, and recent works [Balakrishnan et al., 2023, Jin and Syrgkanis, 2024] proved that for several important causal parameters, this rate is in fact minimax rate-optimal in the SAE framework. In other words, one cannot achieve smaller estimation estimation rates without strong prior knowledge on the underlying structure of the nuisance. These optimality results provide a strong justification for the popularity of doubly robust learning methods in practice.

A curious exception to this rule can be found in the work of Mackey et al. [2018]. There, the authors study the popular partially linear model of Robinson [1988]:

$$
\begin{aligned}
Y &= \theta_0 \cdot T + f_0(X) + \xi, \quad \mathbb{E}[\xi \mid X, T] = 0 \quad \text{almost surely,} \\
T &= g_0(X) + \eta, \quad \text{and} \quad \mathbb{E}[\eta \mid X] = 0 \quad \text{almost surely.}
\end{aligned}
\tag{1}
$$

Here, the observation $Z = (X, T, Y)$ comprises covariates $X \in \mathcal{X}$, a scalar treatment $T \in \mathbb{R}$, and a scalar outcome $Y \in \mathbb{R}$. The primary objective is the estimation of the parameter $\theta_0$, which represents the average causal effect of $T$ on $Y$. Within this framework, the DML estimator for $\theta_0$ is derived from the Neyman-orthogonal moment condition and typically takes the form:

$$
\hat{\theta}_{\mathrm{DML}} = \left[ \tfrac{1}{n} \sum_{i=1}^n (T_i - \hat{g}(X_i))^2 \right]^{-1} \left[ \tfrac{1}{n} \sum_{i=1}^n (Y_i - \hat{q}(X_i))(T_i - \hat{g}(X_i)) \right],
\tag{2}
$$

where $\hat{g}$ and $\hat{q}$ are machine learning estimators for the nuisance functions $g_0(X) = \mathbb{E}[T \mid X]$ and $q_0(X) = \mathbb{E}[Y \mid X]$, respectively [Chernozhukov et al., 2018]. Common practice involves sample splitting, where one portion of the data is used to train $\hat{g}$ and $\hat{q}$ and another (independent) portion is used to compute $\hat{\theta}_{\mathrm{DML}}$ via (2). However, Mackey et al. [2018] showed that for the model (1), one can design structure-agnostic estimators that are more efficient than DML if the treatment noise $\eta$ is non-Gaussian and independent of $X$.[3] Bonhomme et al. [2024] consider parameter estimation problems in a conditional likelihood setting and show that arbitrarily higher-order orthogonal estimators can be constructed if the conditional likelihood function is known up to the nuisance parameters. These works highlight a gap in our understanding of how noise impacts optimal SAE rates, and this is the main question that we will address in this paper.

Given access to black-box nuisance estimates, we make the following contributions:

- On the one hand, we prove in Section 3.2 the existence of *Gaussian treatment barrier*: for Gaussian treatment, no estimators can achieve a better rate than DML, even when the variance of the treatment is completely known. This implies that leveraging distributional information of the treatment noise as in Mackey et al. [2018] cannot yield a better estimate and that their restrictions to the non-Gaussian setting is not an algorithmic issue.

- On the other hand, for non-Gaussian treatment, we propose a general procedure to construct higher-order orthogonal moment functions and provide structure-agnostic guarantees in Section 4. Then, for treatment noise independent of $X$, we derive in Section 5 a new agnostic cumulant-based estimator (ACE) that achieves $r$-th order insensitivity to nuisance estimate error $\epsilon$ and $\mathcal{O}(\epsilon^r + n^{-1/2})$ error rates for treatment effect estimation whenever the $(r + 1)$-st cumulant of $\eta$ is non-zero. To the best of our knowledge, this is the first structure-agnostic estimator that achieves arbitrarily high-order robustness.

- We complement these findings with additional contributions relevant to this setting. Specifically, we show in Section 3.1 that DML is minimax optimal for (1) with binary treatment, a case not covered in existing lower bounds that do not assume a partially linear outcome model.

---

[3]The structure-agnostic rates are not explicitly stated in Mackey et al. [2018] but can be straightforwardly derived from their analysis.

We also derive new lower and upper bounds for structure-agnostic moment and cumulant estimation in a standard non-parametric model in Section 5.1 that might be of independent interest. Finally, in Section 6, we conduct a synthetic demand estimation experiment, highlighting the benefits of the ACE estimator compared with existing approaches.

**Notation**   We introduce the shorthand $[m] \triangleq \{1, \ldots, m\}$ for each $m \in \mathbb{N}$. For a function $g$ and distribution $P$ on a domain $\mathcal{Z}$, we let $\|g\|_{P,s} \triangleq \|g\|_{L^s(P)} \triangleq (\int |g(z)|^s dP(z))^{1/s}$ with $s \geq 1$ represent the $L^s(P)$ norm. For a vector $\epsilon \in \mathbb{R}^l$, we define $\|\epsilon\|_\infty \triangleq \max_{1 \leq i \leq l} |\epsilon_i|$. For two vectors $\alpha = (\alpha_1, \cdots, \alpha_l), \beta = (\beta_1, \cdots, \beta_l) \in \mathbb{R}^l$, we write $\alpha \leq \beta$ if $\alpha_i \leq \beta_i, 1 \leq i \leq l$.

## 2   Structure-agnostic estimation and minimax error

To evaluate and compare method quality in this work, we adopt the minimax structure-agnostic framework of Balakrishnan et al. [2023]. Notably, structure-agnostic analyses make no explicit assumptions about nuisance smoothness, sparsity, or structure and instead simply assume access to black-box nuisance estimates with certain unobserved error levels.

We first define the class of all data generating distributions $P$ on our data domain $\mathcal{Z}$ and assign to each a target parameter $\theta_0(P)$ and a vector-valued nuisance function $h : \mathcal{Z} \mapsto \mathbb{R}^\ell$.

**Definition 2.1** (Data generating distributions, target parameters, and nuisance functions). *Throughout, we let $\mathcal{P}$ be the set of candidate data generating distributions on the finite dimensional-domain $\mathcal{Z}$, let $\mathcal{H} \subseteq \left(\mathbb{R}^\ell\right)^{\mathcal{Z}}$ be the set of relevant vector-valued nuisance functions, and let $\Phi$ be a deterministic mapping from $\mathcal{P}$ to $\mathcal{H}$. Then, for any $P \in \mathcal{P}$, we say that $\theta_0(P) \in \mathbb{R}$ is the target parameter and $h = \Phi(P) \in \mathcal{H}$ is the nuisance function corresponding to the distribution $P$.*

We will sometimes abuse notation and choose $\mathcal{H}$ to be a subset of $\mathcal{H} \subseteq \left(\mathbb{R}^\ell\right)^{\mathcal{X}}$ where $\mathcal{X}$ is the domain of the covariate component $X$ of $Z$. Departing from prior work [Balakrishnan et al., 2023, Jin and Syrgkanis, 2024], we leave the exact choice of nuisance mapping $\Phi$ unspecified in Definition 2.1. This allows us to study how the choice of nuisance functions affects the estimation error of the target parameter of interest.

Next, we introduce the ground-truth uncertainty sets associated with any nuisance estimate $\hat{h}$ and any target error level $\epsilon$. For any vector-valued function $h^\star : \mathcal{X} \mapsto \mathbb{R}^\ell$, distribution $P \in \mathcal{P}$, and $\epsilon \in \mathbb{R}^\ell_+$, we define $\mathcal{B}_{P,s}(h^\star, \epsilon) := \left\{h_i \in L^s(P) : \|h_i - h_i^\star\|_{P,s} \leq \epsilon_i, \forall i \in [\ell]\right\}$.

**Definition 2.2** (Uncertainty sets). *For any nuisance estimate $\hat{h} \in \mathcal{H}$, error level vector $\epsilon \in \mathbb{R}^\ell_+$, and power $s \geq 1$, we define the uncertainty set $\mathcal{P}_{s,\epsilon}(\hat{h}; \Phi)$ as the set of all $P \in \mathcal{P}_0$ satisfying*

$$\|\hat{h}_i - h_i\|_{P,s} \leq \epsilon_i, \quad for \quad h = \Phi(P) \quad and\ each \quad i \in \{1, 2, \cdots, \ell\}, \tag{3}$$

*or, equivalently, $\Phi(P) \in \mathcal{B}_{P,s}(\hat{h}, \epsilon)$.*

For convenience, we will omit the dependency of $\mathcal{P}_{s,\epsilon}(\hat{h}; \Phi)$ on $\mathcal{P}$, which will always be clear from context. We will sometimes write $\mathcal{P}_{s,\epsilon}(\hat{h})$ when the choice of $\Phi$ is obvious. Finally, for a given estimator $\hat{\theta}$ of a target parameter $\theta_0(P)$, we define the worst-case error over an uncertainty set.

**Definition 2.3** (Minimax estimation error). *For any set of distributions $\mathcal{P}$ over $\mathcal{Z}$, we define the worst-case $(1 - \gamma)$-quantile error of an estimator $\hat{\theta} : \mathcal{Z}^{\otimes n} \mapsto \mathbb{R}$ as $\mathfrak{R}_{n,1-\gamma}(\hat{\theta}; \mathcal{P}) \triangleq \sup_{P \in \mathcal{P}} \mathcal{Q}_{P,1-\gamma}(|\hat{\theta} - \theta_0(P)|)$, where $\mathcal{Q}_{P,1-\gamma}(|\hat{\theta} - \theta_0(P)|)$ is a $(1-\gamma)$-quantile of $|\hat{\theta}(Z_1, \ldots, Z_n) - \theta_0(P)|$ when $Z_i \stackrel{i.i.d.}{\sim} P$. We further define the minimax estimation error of $\mathcal{P}$ as $\mathfrak{M}_{n,1-\gamma}(\mathcal{P}) \triangleq \inf_{\hat{\theta} : \mathcal{Z}^{\otimes n} \mapsto \mathbb{R}} \mathfrak{R}_{1-\gamma}(\hat{\theta}; \mathcal{P})$.*

## 3   Structure-agnostic lower bounds

In this section, we establish structure-agnostic lower bounds for treatment effect estimation in the partially linear model (1).

## 3.1 Optimality of DML for binary treatment

We begin by establishing the minimax rate-optimality of DML when the treatment $T$ is binary. Previous works [Balakrishnan et al., 2023, Jin and Syrgkanis, 2024] have established structure-agnostic lower bounds of a similar form. However, Balakrishnan et al. [2023] consider the estimation of the expected conditional covariance defined as $\mathbb{E}[\text{Cov}(T, Y \mid X)]$, while Jin and Syrgkanis [2024] consider the estimation of the average treatment effect with a different set of nuisance functions. Furthermore, neither work constrains the form of $\mathbb{E}[Y \mid T, X]$, while we assume a partially linear structure for the outcome model. Our result implies that even with this additional assumption, it is still impossible to improve over DML. For convenience, we introduce the following definitions as the "default" choice for the class of data generating distributions and nuisance functions:

**Definition 3.1** (Set of feasible distributions). *We define $\mathcal{P}^\star$ as the set of all distributions of $(X, T, Y)$ generated by (1). Moreover, we define the following subsets of $\mathcal{P}^\star$ as follows:*

- *for any constants $C_\theta, C_\mathsf{T}, C_\mathsf{Y} \in [0, +\infty]$, we use $\mathcal{P}_r(C_\theta, C_\mathsf{T}, C_\mathsf{Y})$ to denote all distributions $P \in \mathcal{P}^\star$ that satisfy $|\theta_0| \leq C_\theta$, $\mathbb{E}[|T|^r]^{1/r} \leq C_\mathsf{T}$, and $\mathbb{E}[|Y|^r]^{1/r} \leq C_\mathsf{Y}$.*

- *for any constants $C_\theta, C_\mathsf{g}, C_\mathsf{q}, \psi_\xi, \psi_\eta \in (0, +\infty]$, we use $\mathcal{P}^\star(C_\theta, C_\mathsf{g}, C_\mathsf{q}; \psi_\xi, \psi_\eta)$ to denote the set of all distributions in $P \in \mathcal{P}^\star$ that satisfy*

  1. *$|\theta_0| \leq C_\theta, |g_0(X)| \leq C_\mathsf{g}, |q_0(X)| \leq C_\mathsf{q}$ a.s. for $(g_0, q_0)(X) = (\mathbb{E}[T \mid X], \mathbb{E}[Y \mid X])$*
  2. *$\xi \mid X$ and $\eta \mid X$ are $\psi_\xi$ and $\psi_\eta$-sub-Gaussian a.s..*

- *for any constants $C_\theta, C_\mathsf{T}, C_\mathsf{Y} \in [0, +\infty]$, we use $\mathcal{P}_\mathsf{b}(C_\theta, C_\mathsf{T}, C_\mathsf{Y})$ to denote all distributions $P \in \mathcal{P}^\star$ that satisfy $|\theta_0| \leq C_\theta, |T| \leq C_\mathsf{T}$, and $|Y| \leq C_\mathsf{Y}$.*

*Finally, we define $\Phi^\star$ as a mapping from $P \in \mathcal{P}^\star$ to the "default" nuisance functions $h_0 = (g_0, q_0)$.*

We are interested in the minimax structure-agnostic estimation error induced by $\mathcal{P} = \mathcal{P}_{s,\epsilon}(\hat{h}, \Phi)$ for some $s \geq 1$, in the sense of Definition 2.3, with $\mathcal{P}$ being chosen as a set of distributions that satisfies certain mild regularity conditions, such as the ones introduced in Definition 3.1. In other words, given black-box nuisance estimates of $h_0$ with $L^s(P)$ error rates, we would like to derive the optimal worst-case estimation error for the treatment effect $\theta_0$.

Our main result in this section is a lower bound for estimating $\theta_0$ when $T$ is a Bernoulli random variable. Our lower bound, Theorem 3.1, is established under the additional assumption that $X$ has a uniform distribution on $\mathcal{X} = [0, 1]^K$; as a consequence, the minimax error rate of DML cannot be improved even when the marginal distribution of $X$ is known.

**Theorem 3.1** (Structure-agnostic lower bound for binary treatment). *Fix any $C_\theta > 0$, $c_q, \delta \in (0, \frac{1}{4})$, and $K \in \mathbb{N}_+$, and let*

$$\mathcal{P} = \{P \in \mathcal{P}_\mathsf{b}(C_\theta, 1, 1) : T \in \{0, 1\}, \text{Var}_P(T \mid X) \geq \delta \text{ and } X \sim \text{Unif}([0, 1]^K)\}. \quad (4)$$

*If $\|\epsilon\|_\infty \leq \delta/2$, then for any estimates $\hat{h} = (\hat{g}, \hat{q})$ with $c_q \leq \hat{q}(X) \leq 1 - c_q$ and $\hat{g}(X)(1 - \hat{g}(X)) \geq A^{-1}\delta$, a.s., we have*

$$\mathfrak{M}_{n,1-\gamma}\left(\mathcal{P}_{2,\epsilon}(\hat{h}, \Phi^\star)\right) \geq c_\gamma \left[A^{-1}\delta^{-1}(c_q\epsilon_1^2 + \epsilon_1\epsilon_2) + \delta^{-1/2}n^{-1/2}\right], \quad (5)$$

*for any $\gamma \in (1/2, 1)$, where $c_\gamma$ is a universal constant that only depends on $\gamma$.*

When $\|\epsilon\|_\infty \leq \delta/2$, this matches the upper bound achieved by DML up to constant factors, stated in Theorem C.1 for completeness. The proof of Theorem 3.1 can be found in Appendix B.

## 3.2 The Gaussian treatment barrier

In the previous section, we established the rate-optimality of DML for binary treatment. Is it possible to improve over DML if we make different distributional assumptions? In the literature, it is not uncommon to model the treatment assignment rule using a Gaussian distribution, *i.e.*, $\eta \mid X \sim \mathcal{N}(0, \sigma(X)^2)$ for some function $\sigma(\cdot)$ [Imai and Van Dyk, 2004, Zhao et al., 2020]. However, in Mackey et al. [2018] it is shown that Gaussianity of the noise variable $\eta$ is in fact a barrier for one to construct second-order orthogonal moments, thereby preventing them from deriving better error

rates than DML by leveraging distributional information of $\eta$. However, it is unclear whether this is an issue specific to their approach or if there indeed exists a fundamental, non-algorithmic barrier for Gaussian treatment.

In this section, we resolve this open question and show that the latter is true: if the treatment noise is Gaussian, then DML is already minimax rate-optimal even when $\eta$ is independent of $X$ and one has *exact* knowledge of its distribution. Our lower bound, stated next, is proved in Appendix D.

**Theorem 3.2** (The Gaussian treatment barrier). *Let $\sigma, C_\theta, C_g, C_q > 0$ be known constants and $\mathcal{P} = \{P \in \mathcal{P}_b(C_\theta, \infty, C_q) : \eta \mid X \sim \mathcal{N}(0, \sigma^2)$ and $|g_0(X)| \leq C_g$ a.s.$\}$. If $\epsilon_1 \epsilon_2 = o(\log^{-1/2} n)$, then for any estimates $\hat{h} = (\hat{g}, \hat{q})$ satisfying $|\hat{g}| \leq C_g, |\hat{q}| \leq C_q$ and any $1 \leq s \leq +\infty$, we have*

$$\mathfrak{M}_{n,1-\gamma}\left(\mathcal{P}_{s,\epsilon}(\hat{h}, \Phi^\star)\right) \geq c_\gamma\left(\sigma^{-2}\epsilon_1\epsilon_2\left(\tfrac{\log(1/\epsilon_1)}{\log n}\right)^3 + \sigma^{-1}n^{-1/2}\right) \tag{6}$$

*for any $\gamma \in (1/2, 1)$, where $c_\gamma > 0$ is a constant that only depends on $\gamma$.*

The assumption that $|\hat{g}| \leq C_g, |\hat{q}| \leq C_q$ is natural, since for any $P_0 \in \mathcal{P}_0$, the ground-truth nuisance functions $g_0(X) = \mathbb{E}[T \mid X]$ and $q_0(X) = \mathbb{E}[Y \mid X]$ must satisfy $|g_0| \leq C_g$ and $|q_0| \leq C_q$ a.s. according to our assumption. Moreover, the lower bound does not depend on the value of $s$, meaning that no improvement is possible even if the nuisance estimates have small $L^\infty$-error.

Under the same assumptions, one can show that DML attains the minimax error rate up to the factor $\left(\frac{\log(1/\epsilon_1)}{\log n}\right)^3$ and matches the minimax error rate whenever $\epsilon_1 = \mathcal{O}(n^{-c})$ for some positive constant $c$. For completeness, we include the details in Theorem C.2. Notably, compared with the lower bound in Theorem 3.1, the term $\Theta(\epsilon_1^2)$ disappears here since the variance $\sigma^2$ is assumed to be known. Theorem 3.2 establishes the existence of a method-agnostic *Gaussian treatment barrier* as suggested (but not proved) in Mackey et al. [2018].

The proof of Theorem 3.2 in Appendix D is based on a constrained risk inequality for testing composite hypothesis developed in Cai and Low [2011] combined with novel constructions of the fuzzy hypotheses using moment matching techniques. For Gaussian treatment, we show that such hypotheses can be constructed in a way such that their induced target parameters $\theta_0$ are well-separated by leveraging a recursive property of Hermite polynomials.

## 4 Structure-agnostic upper bounds

Section 3 established the rate optimality of DML for binary and Gaussian treatments but left open the possibility of improvement for other treatment distributions. To exploit this opportunity, we will first introduce a new procedure that yields fast estimation rates whenever $\eta \mid X$ is non-Gaussian and the cumulants of $\eta \mid X$ are estimated accurately and next, in Section 5, show that cumulant estimation is easy when $\eta$ is independent of $X$.

Our new procedure is based on the method of moments. Specifically, we will identify a moment function $m$ satisfying $\mathbb{E}_P[m(Z, \theta, h(X)) \mid X] = 0, a.s.$ for all $P$ in some specified distribution set $\mathcal{P}$, where $\theta = \Phi(P)$ is the ground-truth parameter of interest and $h(\cdot)$ is some vector-valued nuisance functions that we need to estimate from data. Moreover, we require that $\theta = \Phi(P)$ is the unique solution to the moment equation. We proceed by plug in estimates of the nuisance $\hat{h}$ derived from a sample $\mathcal{D}_0$, and select $\hat{\theta}$ satisfying the empirical moment equation $\sum_{i \in \mathcal{D}} m(Z, \theta, \hat{h}(X)) = 0$ on a separate sample $\mathcal{D}$. This procedure is widely adopted in the development of DML-type methods [Chernozhukov et al., 2018], and leads to efficient estimates as long as the moment function $m$ is Neyman-orthogonal, meaning that it is insensitive, under expectation, to nuisance estimation errors. The precise definition will be presented in Theorem 4.1. The novelty of our construction lies in a specific recursive procedure that generates moment functions with arbitrarily high levels of insensitivity to nuisance estimation errors.

Consider the model (1) and let $J_1(w, x)$ be any function of $w \in \mathbb{R}$ and $x \in \mathcal{X}$ satisfying

$$J_1(w, x) = \sum_{i=1}^{M_1} a_{i1}(x)\rho_{i1}(w) \quad \text{and} \quad \mathbb{E}\left[J_1(\eta, X) \mid X\right] = 0 \;\; a.s. \tag{7}$$

where each $\mathbb{E}[a_{i1}(X)^2] < \infty$ and each $\rho_{i1}$ is continuous. Without loss of generality, we assume that $\rho_{11}(w) \equiv 1$ (as one can otherwise introduce a dummy summand into the expression (7) with

$a_{11}(x) \equiv 0$). Let $\{J_r(w,x)\}_{r=2}^\infty$ be a series of functions defined by

$$I_r(w,x) = \int_0^w J_{r-1}(w',x)\mathrm{d}w', \quad J_r(w,x) = I_r(w,x) - \mathbb{E}[I_r(\eta,X) \mid X = x], \qquad (8)$$

The following lemma, proved in Appendix E.4, derives the general form of $J_r$ for each $k \in \mathbb{Z}_+$.

**Lemma 4.1** (Explicit formula for $J_r$). *If $M_r = M_1 + r - 1$, then*

$$J_r(w,x) = \sum_{i=1}^{M_r} a_{ir}(x)\rho_{ir}(w) \ \textit{for} \ a_{ir}(x) = \begin{cases} a_{i-1,r-1}(x) & \textit{if } 1 < i \leq M_r \\ -\mathbb{E}[I_r(\eta,X) \mid X = x] & \textit{if } i = 1, \end{cases}$$

$$\rho_{1r}(w) = 1, \ \textit{and} \ \rho_{ir}(w) = \int_0^w \rho_{i-1,r-1}(w')\mathrm{d}w', \quad 2 \leq i \leq M_r. \qquad (9)$$

*In particular, for all $r \geq 2$ we have $\rho_{2r}(w) = w$.*

We would ideally use the moment function

$$m_r(Z,\theta,h(X)) = \big[Y - q(X) - \theta(T - g(X))\big]J_r(T - g(X), X) \qquad (10)$$

to estimate $\theta_0$. However, by Lemma 4.1, $J_r$ depends on the unknown data generating distribution via the functions $a_{ir}(\cdot)$. Fortunately, our next theorem, proved in Appendix E.3, shows that an *estimated* moment function (11) based on an estimate of $J_r$ yields improved treatment effect estimation rates whenever $\theta_0$ is identifiable and $J_r$ is estimated sufficiently well.

**Theorem 4.1** (Structure-agnostic error from estimated moments). *Consider the datasets* $(\mathcal{D}_1, \mathcal{D}_2) = (\{Z_i\}_{i=1}^{n/2}, \{Z_i\}_{i=n/2+1}^n)$ *with each $Z_i \subseteq \mathcal{Z}$. Define the estimated moment function*

$$\hat{m}_r(Z,\theta,h(X);\mathcal{D}_1) = \big[Y - q(X) - \theta(T - g(X))\big]\hat{J}_r(T - g(X), X; \mathcal{D}_1), \qquad (11)$$

*where $h = (g, q)$ and $\hat{J}_r(w,x;\mathcal{D}_1) \triangleq \sum_{i=1}^{M_r} \hat{a}_{ir}(x;\mathcal{D}_1)\rho_{ir}(w)$ for $\hat{a}_{ir} : \mathcal{X} \times \mathcal{Z}^{n/2} \mapsto \mathbb{R}$ and $\rho_{ir}(\cdot)$ recursively defined via (9). Fix any $C_\theta, C_g, C_q; \psi_\xi, \psi_\eta > 0$, $\gamma \in (0,1)$, $s \geq r + 1$, and $\Delta \in \mathbb{R}_+^2$, and let $\mathcal{P} \subseteq \mathcal{P}^\star(C_\theta, C_g, C_q; \psi_\xi, \psi_\eta)$ contain all distributions $P$ satisfying, with probability $1 - \gamma/2$ over $\mathcal{D}_1 \overset{i.i.d.}{\sim} P$, the following four conditions simultaneously:*

$$\big|\mathbb{E}_{Z\sim P}\big[\nabla_\theta \hat{m}_r(Z,\theta_0,h_0(X);\mathcal{D}_1 \mid \mathcal{D}_1)\big]\big| \geq \delta_{\mathsf{id}} \ \textit{(identifiability)} \qquad (12)$$

$$\sup_{h \in \mathcal{B}_{P,s}(h_0,\Delta)} \mathrm{Var}_{Z\sim P}\big(\hat{m}_r(Z,\theta_0,h(X);\mathcal{D}_1) \mid \mathcal{D}_1\big) \leq V_{\mathsf{m}} \ \textit{(finite variance)} \qquad (13)$$

$$\max_{\beta \in \{r,r+1\}} \sup_{h \in \mathcal{B}_{P,s}(h_0,\Delta)} \big\|\hat{J}_r^{(\beta)}(T - g(X), X; \mathcal{D}_1) \mid \mathcal{D}_1\big\|_{L^s(P)} \leq \Lambda_r \ \textit{(finite derivatives)} \qquad (14)$$

$$\sup_{x \in \mathcal{X}} \sup_{0 \leq j \leq r} \big|\mathbb{E}\big[\hat{J}_r^{(j)}(\eta, X; \mathcal{D}_1) \mid X = x, \mathcal{D}_1\big]\big| \leq \epsilon^{(j)} \ \textit{(near orthogonality)} \qquad (15)$$

*where $\hat{J}_r^{(j)}(w,x) \triangleq \frac{\mathrm{d}^j}{\mathrm{d}w^j}\hat{J}_r(w,x)$. Let $\hat{h} = (\hat{g}, \hat{q})$ be a possibly random function independent of $\mathcal{D}$ and $\hat{\theta}$ be the solution of $\frac{1}{n}\sum_{i=n/2+1}^n \hat{m}_r(Z_i,\theta,\hat{h};\mathcal{D}_1) = 0$. Then there exists a constant $C_\gamma > 0$ that only depends on $\gamma$, such that for all $\epsilon \leq \Delta$,*

$$\begin{aligned} \mathfrak{R}_{n,1-\gamma}(\hat{\theta}; \mathcal{P}_{s,\epsilon}(\hat{h}, \Phi^\star)) \leq C_\gamma \delta_{\mathsf{id}}^{-1}\Big[&\sqrt{\tfrac{V_{\mathsf{m}}}{n}} + \sum_{j=0}^{r-1} \tfrac{1}{j!}\max\{\epsilon^{(j+1)}, \epsilon^{(j)}\}\big(\epsilon_1^{j+1} + \epsilon_1^j \epsilon_2\big) \\ &+ \tfrac{1}{(r+1)!}\Big(\big(4(\psi_\xi + C_\theta\psi_\eta)\sqrt{s} + C_\theta\big)\epsilon_1^{r+1} + r\epsilon_1^r \epsilon_2\Big)\Lambda_r\Big], \end{aligned} \qquad (16)$$

In particular, if $\epsilon^{(j)} = \mathcal{O}(\max\{\epsilon_1, \epsilon_2\}^{r-j})$, then the treatment effect error rate (16) has $r$-th order dependencies on the nuisance errors $\epsilon_i, i = 1, 2$ in place of the slower second-order dependencies of DML (see, e.g., Theorems C.1 and C.2). One caveat of applying this bound is that $\delta_{\mathsf{id}}, V_{\mathsf{m}}$, and $\lambda^\star$ all depend on the order of orthogonality $r$ and need to be computed in a case-by-case manner. In Section 5.2, we will construct an explicit moment estimator that satisfies $\epsilon^{(j)} = \mathcal{O}(\max\{\epsilon_1, \epsilon_2\}^{r-j})$ and make the dependency on $r$ explicit.

The identifiability assumption (12) is crucial and is why the construction of Theorem 4.1 does *not* work for Gaussian treatments. Indeed, as we show in Proposition I.2, if the treatment is Gaussian and (15) holds with $\epsilon^{(j)} \to 0$, then $\delta_{\mathsf{id}} \to 0$, so that Theorem 4.1 cannot yield $\sqrt{n}$-consistency.

A natural choice for $J_1(w,x)$ satisfying (7) is $J_1(w,x) \equiv w$, which corresponds to selecting $a_{11} \equiv 0, \rho_{11} \equiv 1, a_{21} \equiv 1$, and $\rho_{21}(w) = w$. In this case, Lemma 4.1 implies that each $J_r(w,x)$ is the following $r$-th order polynomial of $w$:

$$J_r(w,x) = \sum_{i=1}^{r+1} a_{ir}(x)w^{i-1}, \text{ where } a_{il}(x) = \frac{1}{(i-1)!(l+1-i)!} \sum_{\pi \in \Pi_{l+1-i}} (-1)^{|\pi|} \prod_{B \in \pi} \kappa_{|B|}(x). \quad (17)$$

Here, $\Pi_m$ denotes the set of all partitions of $[m]$ [4] and $\kappa_i(x)$ is the $i$-th cumulant of $\eta \mid X = x$. In particular, if $\mu_i(x)$ denotes the $i$-th moment of $\eta \mid X = x$, then $J_2(w,x) = \frac{1}{2}(w^2 - \mu_2(x))$ and $J_3(w,x) = \frac{1}{6}(w^3 - 3\mu_2(x)w - \mu_3(x))$. In Section 5, we will show how to estimate these cumulant-based $J_r$ effectively whenever $\eta$ is non-Gaussian and independent of $X$.

## 5   Agnostic cumulant-based estimation (ACE)

In this section, we apply the general guarantee in Theorem 4.1 to derive structure-agnostic estimators with better rates than DML when $\eta$ is non-Gaussian and independent of $X$.

### 5.1   Structure-agnostic cumulant estimation

The moment function induced by the $J_k(w,x)$ defined in (17) requires estimating the cumulants $\kappa_i$ of the noise variable $\eta$ in the treatment regression model $T = g_0(X) + \eta$. In this subsection, we propose efficient structure-agnostic cumulant estimators assuming that $\eta$ is independent of $X$. We will also see in Remark 5.2 that our approach has potential benefits even when this assumption fails.

Our main result, stated below, indicates that an $r$-th order error rate can be attained for estimating the $r$-th cumulant of $\eta$.

**Theorem 5.1** (Efficient cumulant estimator for noise with finite moments). *Let $C_T > 0$ be a constant, $r \geq 2$ be a positive integer, and $\mathcal{P}$ be the set of all distributions of $(X,T)$ generated from $T = g_0(X) + \eta$, such that $\mathbb{E}[|T|^r \mid X]^{1/r} \leq C_T$ holds a.s.. The target parameter $\theta_0(P) : \mathcal{P} \mapsto \mathbb{R}$ is the $r$-th order cumulant of $\eta = T - g_0(X)$ under $P$. Let $\Phi$ map $P \in \mathcal{P}$ to the nuisance function $g_0$. Let $\hat{g} : \mathcal{X} \mapsto \mathbb{R}$ be a nuisance estimate that satisfies $|\hat{g}(X)| \leq C_g$ and $\hat{\kappa}_r : \{(X_i, T_i)\}_{i=1}^n \mapsto \mathbb{R}$ be the $r$-th order cumulant of the empirical residual distribution $P_n = \frac{1}{n}\sum_{i=1}^n \delta_{T_i - \hat{g}(X_i)}$ where $\delta_z$ is the Dirac measure at $z$. Then for any $\gamma \in (0,1)$ and $s \geq r$, $\hat{\theta} = \hat{\kappa}_r$ satisfies $\mathfrak{R}_{n,1-\gamma}\left(\hat{\theta}; \mathcal{P}_{s,\epsilon}(\hat{g}, \Phi)\right) \leq C_{\gamma,r} n^{-1/2} + (2r\epsilon)^r$ where $C_{\gamma,r} = 10r^{1/2}\gamma^{-1/2}(2C_T)^r(r-1)!$.*

Our next theorem derives a refined bound for sub-Gaussian treatment noise.

**Theorem 5.2** (Efficient cumulant estimator for sub-Gaussian noise). *Let $C_g, \psi_\eta > 0$ be a constant and $\mathcal{P}$ be the set of all distributions of $(X,T)$ generated from $T = g_0(X) + \eta$, such that $|g_0(X)| \leq C_g$, a.s. and $\eta$ is mean-zero, independent of $X$ and $\psi_\eta$-sub-Gaussian. The target parameter $\theta_0(P) : \mathcal{P} \mapsto \mathbb{R}$ is the $r$-th order cumulant of $\eta = T - g_0(X)$ under $P$. Let $\Phi$ map $P \in \mathcal{P}$ to the nuisance function $g$. Let $\hat{g} : \mathcal{X} \mapsto \mathbb{R}$ be a nuisance estimate that satisfies $|\hat{g}(X)| \leq C_g$ and $\hat{\kappa}_r : \{(X_i, T_i)\}_{i=1}^n \mapsto \mathbb{R}$ be the $r$-th order cumulant of the empirical residual distribution $P_n = \frac{1}{n}\sum_{i=1}^n \delta_{T_i - \hat{g}(X_i)}$ where $\delta_z$ is the Dirac measure at $z$. Then for any $\gamma \in (0,1)$ and $s \geq r$, $\hat{\theta} = \hat{\kappa}_r$ satisfies $\mathfrak{R}_{n,1-\gamma}\left(\hat{\theta}; \mathcal{P}_{s,\epsilon}(\hat{g}, \Phi)\right) \leq C_{\gamma,r} n^{-1/2} + (2r\epsilon)^r$ where $C_{\gamma,r} = 3\left[12r(C_g + \psi_\eta)\right]^r l^{1/2}\gamma^{-1/2}$.*

**Remark 5.1** (Comparing the bounds in Theorem 5.1 and Theorem 5.2). *Under the assumptions in Theorem 5.2, $C_T$ in Theorem 5.1 would be $\mathcal{O}(C_g + \sqrt{r}\psi_\eta)$, so that Theorem 5.1 implies a bound which scales as $(cr)^{3r/2}n^{-1/2} + (2r\epsilon)^r$, while the bound in Theorem 5.2 scales as $(c'r)^r n^{-1/2} + (2r\epsilon)^r$, which is strictly tighter.*

**Remark 5.2** (Relaxing the independent noise assumption). *While Theorem 5.2 assumes that the noise variable $\eta$ is independent of $X$, the estimator $\hat{\kappa}_r$ can also be of value when this assumption does not hold. In Appendix F.3, we consider $r = 3$ and derive guarantees for this estimator when $\eta$ is "nearly" independent of $X$.*

---

[4] For example, $\Pi_3 = \{\{\{1,2,3\}\}, \{\{1,2\},\{3\}\}, \{\{1,3\},\{2\}\}, \{\{2,3\},\{1\}\}, \{\{1\},\{2\},\{3\}\}\}$.

The fast rates in Theorems 5.1 and 5.2 hold specifically for estimating the cumulants. When the target estimand is instead a *moment* of $\eta$, our structure-agnostic lower bound in Theorem F.1 requires $\Omega(n^{-1/2} + \epsilon^3)$ minimax error for $r = 3$ and $\Omega(n^{-1/2} + \epsilon^2)$ minimax error for $r \neq 3$. Notably, this cubic-quadratic bottleneck implies that, unlike the cumulant-based approach espoused here, the moment-based approach of Mackey et al. [2018] cannot attain arbitrarily high-order error rates.

The proofs of Theorems 5.1 and 5.2 can be found in Appendix F.2. To the best of our knowledge, these results are novel and may be of independent interest. Next, we will apply these result to construct more efficient structure-agnostic estimators of treatment effects.

## 5.2 Fast rates with independent treatment noise

In this subsection, we introduce agnostic cumulant-based estimation (ACE), a novel treatment effect estimator that leverages the efficient cumulant estimators of Section 5.1.

Throughout, we let $\hat{\kappa}_i$ be the empirical cumulant estimate defined in Theorem 5.2 and define

$$\hat{J}_r(w) = \sum_{i=1}^{r+1} \hat{a}_{ir} w^{i-1} \text{ with } \hat{a}_{1r} = 1 \ \hat{a}_{ir} = \frac{1}{(i-1)!(r+1-i)!} \sum_{\pi \in \Pi_{r+1-i}} (-1)^{|\pi|} \prod_{B \in \pi} \hat{\kappa}_{|B|}, \forall i \geq 2. \quad (18)$$

$\hat{J}_r(\cdot)$ can be viewed as an estimate of the cumulant-based function $J_r(\cdot)$ (17) when $X$ is independent of $\eta$. The key observation is that this $\hat{J}_r(\cdot)$ satisfies (15) with $\epsilon^{(j)} = \mathcal{O}(\epsilon_1^j)$, which follows from the key lemma stated below.

**Lemma 5.1** (Key lemma; higher-order insensitivity condition). *For all* $k \in [r]$, $\mathbb{E}\left[\hat{J}_r^{(k)}(T - g_0(X))\right] = \frac{1}{(r-k)!} \sum_{\pi \in \Pi_{r-k}} \prod_{B \in \pi} \left(\kappa_{|B|} - \hat{\kappa}_{|B|}\right)$.

Notably, each term in the RHS in Lemma 5.1 is a product of cumulant estimation errors. Recall in Theorem 5.2 we show that the estimation error of $\hat{\kappa}_r$ is $\mathcal{O}(\epsilon_1^r)$ when $\|\hat{g} - g\|_{P,s} \leq \epsilon_1$, so that $\prod_{B \in \pi} \left(\kappa_{|B|} - \hat{\kappa}_{|B|}\right) = \mathcal{O}\left(\prod_{B \in \pi} \epsilon^{|B|}\right) = \mathcal{O}(\epsilon_1^{r-k})$. We additionally bound the coefficient hidden in the $\mathcal{O}(\cdot)$ in Lemma G.5. In view of this favorable property, we propose our estimation algorithm, ACE, in Algorithm 1.

---

**Algorithm 1:** Agnostic Cumulant-based Estimation (ACE)

**Input** : Nuisance estimates $\hat{g}$ and $\hat{q}$; observations $\mathcal{D} = \{Z_i = (X_i, T_i, Y_i)\}_{i=1}^n$; order $r$.
**Output:** An estimate $\hat{\theta}$ of the treatment effect $\theta_0$ defined in (1).

1 Split the data into two sets: $\mathcal{D}_1 = \{(X_i, T_i, Y_i)\}_{i=1}^{n/2}, \mathcal{D}_2 = \{(X_i, T_i, Y_i)\}_{i=n/2+1}^n$;
2 **for** $k \leftarrow 1$ *to* $r + 1$ **do**
3 $\quad \mu'_k \leftarrow \frac{2}{n} \sum_{i=1}^{n/2} (Y_i - \hat{g}(X_i))^k$;
4 Define the cumulant-based function $\hat{J}_r(\eta)$ as in (18);
5 Define the moment function $\hat{m}_r(Z, \theta, h(X)) = [Y - q(X) - \theta(T - g(X))] \hat{J}_r(T - g(X))$;
6 **return** $\hat{\theta} \leftarrow$ *solution of* $\frac{2}{n} \sum_{i=n/2+1}^n \hat{m}_r(Z_i, \theta, \hat{h}(X_i)) = 0$

---

The next two theorems, proved in Appendix G.3 and Appendix G.2 respectively, show that ACE can achieve higher-order error rates for treatment effect estimation when $\eta$ is non-Gaussian.

**Theorem 5.3** (ACE estimation error). *Let* $r \in \mathbb{Z}_+$ *and* $\delta_{\mathsf{id}}, C_\theta, C_\mathsf{T}, C_\mathsf{Y} > 0$ *be constants and* $\mathcal{P}$ *be the set of all distributions in* $\mathcal{P}_{2r+2}(C_\theta, C_\mathsf{T}, C_\mathsf{Y})$ *with* $\eta$ *independent of* $X$ *and* $|\kappa_{r+1}| \geq \delta_{\mathsf{id},r}$. *Then, for any* $\gamma \in (1/2, 1)$, *there exists* $C_\gamma > 0$ *such that for all* $\epsilon_1, \epsilon_2 > 0$, *if*

$$r \leq \min\left\{\frac{b_1}{a_1}, \frac{b_2}{a_2 \log(a_2 b_2)}\right\} \quad (19)$$

*where* $b_1 = \log(\gamma n/100)$, $b_2 = 50 \min\{1, C_\theta\} \delta_{\mathsf{id},r} \max\{\epsilon_1, \epsilon_2, (\gamma n)^{-1/2} C_\mathsf{Y}\}^{-1}$, $a_1 = 2 \log(C_\mathsf{T} \epsilon_1^{-1}/2)$, *and* $a_2 = C_\mathsf{T}$, *then then the* $r$-*th order ACE estimator* $\hat{\theta}$ *satisfies*

$$\mathfrak{R}_{n,1-\gamma}(\hat{\theta}; \mathcal{P}_{r,\epsilon}(\hat{h}, \Phi^\star)) \leq C_\gamma r! 4^r \delta_{\mathsf{id},r}^{-1} \left[\epsilon_1^r \epsilon_2 + C_\theta \epsilon_1^{r+1} + 64 C_\mathsf{T}^r (r^2 C_\mathsf{T} + C_\mathsf{Y})(\gamma n)^{-1/2}\right]. \quad (20)$$

**Remark 5.3** (Power of non-Gaussianity). *When $\eta$ is Gaussian, its cumulant $\kappa_{r+1} = 0$ for all $r$, violating the assumption that $|\kappa_{r+1}| \geq \delta_{\mathsf{id},r}$ in Theorem 5.3. Conversely, for non-Gaussian $\eta$, this condition is always satisfied for some $r$ by Levy's Inversion Formula [Durrett, 2019, Theorem 3.3.11], allowing us to obtain higher-order error rates.*

Notably, the constant $C_{\mathsf{T}}$ in (19) may itself grow with $r$. For example, if $\eta = T - g_0(X)$ is sub-Gaussian, we can have $C_{\mathsf{T}} = \Theta(\sqrt{r})$. The theorem below makes this dependence explicit and delivers an even sharper bound in the sub-Gaussian regime.

**Theorem 5.4** (ACE estimation error: sub-Gaussian noise). *Let $\delta_{\mathsf{id}}, C_\theta, C_{\mathsf{g}}, C_{\mathsf{q}}, \psi_\eta, \psi_\xi > 0$ and $r \in \mathbb{Z}_+$ be constants and $\mathcal{P}$ be the set of all distributions in $\mathcal{P}^*(C_\theta, C_{\mathsf{g}}, C_{\mathsf{q}}; \psi_\xi, \psi_\eta)$ with $\eta$ independent of $X$ and $|\kappa_{r+1}| \geq \delta_{\mathsf{id},r}$. Then, for any $\gamma \in (1/2, 1)$, there exists $C_\gamma > 0$ such that $\forall \epsilon_1, \epsilon_2 > 0$, if*

$$r \leq \min\left\{ \frac{b_1}{a_1} - \frac{1}{a_1}\log\frac{b_1}{a_1}, \frac{b_2}{a_2\log(a_2 b_2)} \right\} \tag{21}$$

*where $b_1 = \log(\gamma n/9)$, $b_2 = 200\min\{1, C_\theta\}\delta_{\mathsf{id},r}\max\left\{\epsilon_1, \epsilon_2, (\gamma n)^{-1/2}(\psi_\xi + C_\theta\psi_\eta)\right\}^{-1}$, $a_1 = 2\log(6(C_{\mathsf{g}} + \psi_\eta)\epsilon_1^{-1})$, and $a_2 = 4(C_{\mathsf{g}} + \psi_\eta)$ then the $r$-th order ACE estimator $\hat\theta$ satisfies*

$$\mathfrak{R}_{n,1-\gamma}(\hat\theta; \mathcal{P}_{r,\epsilon}(\hat h, \Phi^\star))$$
$$\leq C_\gamma r! 16^r \delta_{\mathsf{id},r}^{-1}\left[\epsilon_1^r\epsilon_2 + C_\theta\epsilon_1^{r+1} + 64(C_{\mathsf{g}} + \psi_\eta)^r\left(r^2(C_{\mathsf{g}} + \psi_\eta) + \psi_\xi + C_\theta\psi_\eta\right)(\gamma n)^{-1/2}\right]. \tag{22}$$

**Remark 5.4** (Scale of the leading coefficient under uniform noise). *As shown in (22), the estimation error of $r$-th order ACE estimator depends not only on $r, \epsilon_1, \epsilon_2, n$, but also on $\kappa_{r+1}$. This is intuitive as $\kappa_{r+1}$ is a measure of non-Gaussianity. An estimate of $\kappa_{r+1}$ can also be used to estimate the variance of $\hat\theta$; see Section 6 for more details. To understand the role of $\delta_{\mathsf{id},r}$ in the bound, consider the case when $\eta$ follows a uniform distribution on $[-1, 1]$. Then for any $m \in \mathbb{Z}_+$, we have $\kappa_{2m} \sim 4\sqrt{\frac{\pi}{m}}\left(\frac{m}{\pi e}\right)^{2m}$ [Binet, 1839]. Plugging into (22), we have*

$$\mathfrak{R}_{n,1-\gamma}(\hat\theta; \mathcal{P}_{s,\epsilon}(\hat h)) \leq 4r(4\pi e)^r\delta_{\mathsf{id},r}^{-1}\left[\epsilon_1^r\epsilon_2 + C_\theta\epsilon_1^{r+1} + 8(C_{\mathsf{Y}} + (C_\theta+1)C_{\mathsf{T}})(4C_{\mathsf{T}})^r(\gamma n)^{-1/2}\right]. \tag{23}$$

*Hence the leading coefficient is only exponential in $r$, rather than super-exponential.*

When $r = 1$, ACE is identical to DML. When $r = 2, 3$ it recovers the "second-order" orthogonal estimators proposed by Mackey et al. [2018]. Interestingly, for $r = 3$, the rate given by Theorem 5.4 is faster than that of Mackey et al. [2018, Thm. 10], as the latter did not establish third-order orthogonality. When $r \geq 4$, to the best of our knowledge, ACE is novel, and we derive the explicit expressions for $r = 3, 4$ in Appendix G.4.

As a concrete instantiation of Theorem 5.4, consider the setting of high-dimensional linear nuisance,

$$g_0(x) = \langle\alpha_0, x\rangle \text{ and } q_0(x) = \langle\beta_0, x\rangle \text{ for } \alpha_0, \beta_0 \in \mathbb{R}^p, \; s_1 \triangleq \|\alpha_0\|_0, \text{ and } s_2 \triangleq \|\beta_0\|_0, \tag{24}$$

where $(p, s_1, s_2)$ all potentially grow with $n$, and the nuisance functions are estimated using Lasso regression [Hastie et al., 2015, Chap. 11]. In this setting, DML is known to provide order $n^{-1/2}$ estimation error for $\theta_0$ whenever the maximum sparsity level $\max(s_1, s_2) = o(n^{1/2}/\log p)$ [Chernozhukov et al., 2018, Rem. 4.3]. Remarkably, as we prove in Appendix G.5, $r$-th order ACE provides the same guarantee when $\max(s_1, s_2) = o(n^{r/(r+1)}/\log p)$.

## 6 Numerical experiments

To evaluate the empirical effectiveness of our proposed estimators, we simulate a demand estimation scenario using purchase and pricing data. In this setting, $Y$ represents observed demand, the treatment $T$ corresponds to an observed product price, $g_0(X)$ denotes a baseline product price determined by covariates $X$ that influence pricing policy, and the treatment noise $\eta$ represents a random discount offered to customers for demand assessment. Notably, $\eta$ is typically discrete (and thus distinctly non-Gaussian) and independent of $X$.

We replicate the experimental framework of Mackey et al. [2018, Section 5], where $X \sim \mathcal{N}(0, I)$, $\epsilon \sim U([-3, 3])$, and $\eta$ follows a discrete distribution on $\{0.5, 0, -1.5, -3.5\}$ with probabilities

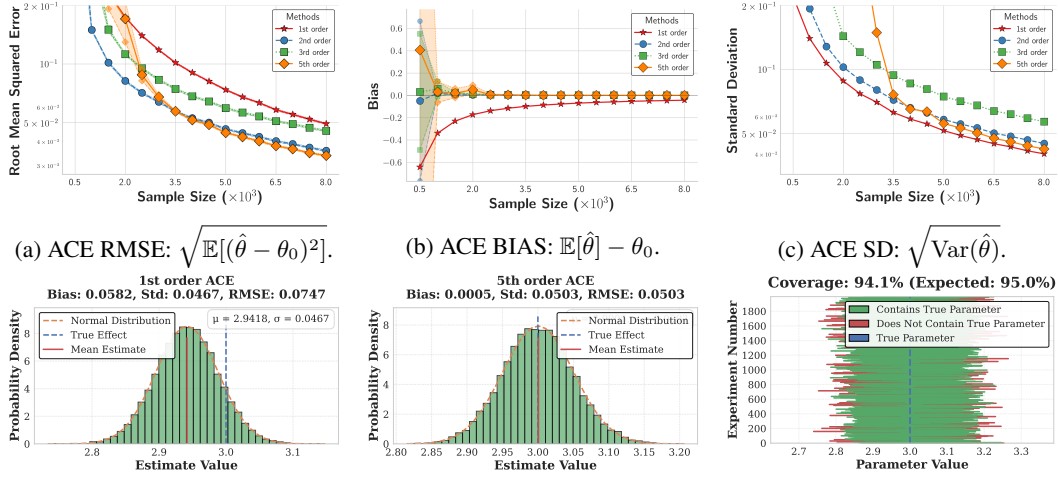

(a) ACE RMSE: $\sqrt{\mathbb{E}[(\hat{\theta} - \theta_0)^2]}$.

(b) ACE BIAS: $\mathbb{E}[\hat{\theta}] - \theta_0$.

(c) ACE SD: $\sqrt{\mathrm{Var}(\hat{\theta})}$.

(d) First-order ACE estimates.

(e) Fifth-order ACE estimates.

(f) Fifth-order confidence intervals.

Figure 1: Comparison of first through fifth-order ACE estimation (Algorithm 1) in the synthetic demand estimation setting of Section 6. Fourth-order ACE is omitted due to substantially larger error. All quality measures and shaded 95% confidence bands are estimated using 20000 independent replicates of the experiment.

$\{0.65, 0.2, 0.1, 0.05\}$, respectively. Each nuisance function is specified as a sparse linear function in $p = 100$ dimensions with $s = 40$ non-zero coefficients.

We examine the $r$-th order ACE estimator introduced in Section 4 across different values of $r$. For $r = 1, 2$, this framework precisely recovers the first-order [Chernozhukov et al., 2018] and second-order [Mackey et al., 2018] orthogonal estimators. First-stage nuisance function estimates are obtained using Lasso regression [Tibshirani, 1996], following Corollary G.3. Complete Python code for replicating all experiments is available at https://github.com/JikaiJin/ACE.

In view of the high-probability bounds in Theorem 5.4, we empirically assess ACE performance for orders $r \leq 5$ across varying sample sizes. A comparison of the total RMSE is provided in Figure 1a, demonstrating that the fifth-order ACE estimator achieves optimal performance. We further decompose RMSE into bias and variance components. Figure 1b compares bias across different orders, with fifth-order ACE exhibiting the smallest bias. Moreover, Figure 1c shows that the first-order ACE estimator achieves the lowest standard deviation, followed by the fifth-order estimator. Figures 1d and 1e present the distribution of estimated values using first- and fifth-order ACE estimators. Both distributions are approximately Gaussian, with the first-order estimator exhibiting substantially larger bias. Based on Theorem 4.1, the variance of $\hat{\theta}$ is bounded by $\delta_{\mathsf{id}}^{-1}(V_{\mathsf{m}}/n)^{1/2}$, where $\delta_{\mathsf{id}}$ provides a lower bound for $\kappa_{r+1}$ in the context of Theorem 5.4. This enables us to construct a direct plug-in variance estimate $\mathcal{E}_{\mathsf{var}}$ as $\mathcal{E}_{\mathsf{var}} = \hat{\kappa}_r^{-1}\sqrt{\frac{V_{\mathsf{m}}}{n}}$ for $V_{\mathsf{m}} = \frac{1}{n}\sum_{i=1}^n \left[(Y_i - \hat{q}(X_i))^2 + \hat{\theta}^2(T_i - \hat{g}(X_i))^2\right]\hat{J}_r(T_i - \hat{g}(X_i))^2$. Lastly, following Corollary E.1, we construct the approximate 95% confidence interval $[\hat{\vartheta} - 1.96\mathcal{E}_{\mathsf{var}}^{1/2}, \hat{\vartheta} + 1.96\mathcal{E}_{\mathsf{var}}^{1/2}]$ for $\theta_0$. Figure 1f demonstrates that approximately 95% of independent experiments yield confidence intervals that contain the true parameter value, confirming the validity of our constructed intervals.

## 7 Conclusion and future directions

In this paper, we provide new insights into how distributional properties could change the statistical limit of structure-agnostic estimation. Focusing on a partial linear outcome model, we show that the Gaussianity of the treatment variable creates a fundamental barrier for improving over DML, while improvements upon DML is possible for non-Gaussian treatment. Moving forward, it would be of interest to exploit distributional properties to design estimators more efficient than DML for heterogeneous treatment effects.

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

# Appendix

## Table of Contents

# A    Preliminaries

## A.1    Semiparametric bounds

Our proofs of lower bounds are based on on the method of fuzzy hypothesis. A key lemma is stated below:

**Lemma A.1.** *(Tsybakov [2008], Theorem 2.15) Let $Z$ be an observation with $\text{supp}(Z) = \mathcal{Z}$, $P \in \mathcal{P}$, $P_1 \subseteq \mathcal{P}$ and $\pi$ be a probability distribution on $\mathcal{P}_1$, which induce the distribution $Q_1(A) = \int Q^{\otimes n}(A) d\pi(Q)$, $\quad \forall A \subset \mathcal{P}$. Suppose that there exists a functional $T : \mathcal{P} \mapsto \mathbb{R}$ which satisfies*

$$T(P) \leq c, \quad \pi(\{Q : T(Q) \geq c + 2s\}) = 1 \tag{25}$$

*for some $s > 0$. If $H^2\left(P^{\otimes n}, Q_1\right) \leq \xi < 2$, then:*

$$\inf_{\hat{T}:\mathcal{Z}^n \mapsto \mathbb{R}} \sup_{P \in \mathcal{P}} P\left[\left|\hat{T} - T(P)\right| \geq s\right] \geq \frac{1 - \sqrt{\xi(1 - \xi/4)}}{2}.$$

We will use the following lemma to bound the Hellinger distance as required in Lemma A.1

**Lemma A.2.** *(Robins et al. [2009, Theorem 2.1], see also Jin and Syrgkanis [2024, Theorem 4]) let $\mathcal{Z} = \cup_{j=1}^m \mathcal{Z}_j$ be a measurable partition of the sample space. Given a vector $\lambda = (\lambda_1, \ldots, \lambda_m)$ in some product measurable space $\Lambda = \Lambda_1 \times \cdots \times \Lambda_m$, let $P$ and $Q_\lambda$ be probability measures on $\mathcal{Z}$ such that the following statements hold:*

- *$P\left(\mathcal{Z}_j\right) = Q_\lambda\left(\mathcal{Z}_j\right) = p_j$ for every $\lambda \in \Lambda$, and*

- *the probability measures $P$ and $Q_\lambda$ restricted to $\mathcal{Z}_j$ depend $\lambda_j$ only.*

*Let $p$ and $q_\lambda$ be the densities of the measures $P$ and $Q_\lambda$ that are jointly measurable in the parameter $\lambda$ and the observation $x$, and $\pi$ be a probability measure on $\Lambda$. Define $b = m \max_j \sup_\lambda \int_{\mathcal{X}_j} \frac{(q_\lambda - p)^2}{p} d\mu$. Suppose that $p = \int q_\lambda d\pi(\lambda)$ and that $n \max\{1, b\} \max_j p_j \leq A$ for all $j$ for some positive constant $A$, then there exists a constant $C$ that depends only on $A$ such that, for any product probability measure $\pi = \pi_1 \otimes \cdots \otimes \pi_m$,*

$$H\left(P^{\otimes n}, \int Q_\lambda^{\otimes n} d\pi(\lambda)\right) \leq \max_j p_j \cdot Cn^2 b^2.$$

## A.2    Useful properties of sub-Gaussian distributions

In this subsection, we recall a few useful properties of sub-Gaussian distributions. Recall that for a variable $Z$, its sub-Gaussian norm is defined as

$$\|Z\|_{\psi_2} = \inf\left\{c > 0 : \mathbb{E}\exp(Z^2/c^2) \leq 2\right\}. \tag{26}$$

**Proposition A.1** (Moment bounds for sub-Gaussian variables, see e.g. , Vershynin [2018] Proposition 2.5.2). *Suppose that $Z$ is a sub-Gaussian random variable with Orlicz norm $\sigma = \|Z\|_{\psi_2}$. Then for every integer $k \geq 1$*

$$\left(\mathbb{E}|Z|^k\right)^{1/k} \leq C\sigma\sqrt{k}, \qquad \text{equivalently} \qquad \mathbb{E}|Z|^k \leq (C\sigma\sqrt{k})^k,$$

*where $C > 0$ is an absolute constant; one may take $C = 2$.*

The following bound of cumulants is due to Saulis and Statulevicius [2012], Lemma 1.5.

**Proposition A.2** (Cumulant bounds for sub-Gaussian variables). *Let $Z$ be a centred sub-Gaussian random variable with Orlicz norm $\sigma = \|Z\|_{\psi_2}$, i.e. $\mathbb{E}\left[e^{tZ}\right] \leq \exp(\sigma^2 t^2/2)$ for all $t \in \mathbb{R}$. Denote by $\kappa_r(Z)$ its $r$-th cumulant, $r \in \mathbb{N}$. Then*

$$|\kappa_r(Z)| \leq (r-1)!\,(4\sigma^2)^{r/2}, \qquad r \geq 2.$$

*In particular, the sequence $\{|\kappa_r(Z)|^{1/r}\}_{r \geq 2}$ grows at most like $2\sigma\sqrt{r}$, which is sharp up to the constant 2 (no smaller absolute constant works for all sub-Gaussians).*

# B  Proof and discussion of Theorem 3.1: Structure-agnostic lower bound for binary treatment

In this section, we present the proof of Theorem 3.1.

We define the following data generating distribution:

$$
\begin{aligned}
\lambda_i &\sim \text{Uniform}(\{-1, +1\}), \quad i = 1, 2, \cdots, M \\
X &\sim \text{Uniform}(\mathcal{X}) \\
T \mid X = x &\sim \text{Bernoulli}(\hat{g}(x)) \\
Y \mid T = t, X = x &\sim \text{Bernoulli}(\hat{f}(x) + \hat{\theta}t),
\end{aligned}
\tag{27}
$$

where $\hat{\theta} = \frac{1}{2}c_q$ and $\hat{f}(x) = \hat{q}(x) - \hat{\theta}\hat{g}(x)$. Since $c_q \leq \hat{q}(x) \leq 1 - c_q$ and $0 \leq \hat{g}(x) \leq 1$ by assumption, it is easy to see that $\hat{f}(x) + \theta t \in [0, 1], t \in \{0, 1\}$. Hence (27) defines a valid data generating distribution. We denote the joint distribution of $(X, T, Y)$ as $\hat{P}$.

Since $\gamma \in \left(\frac{1}{2}, 1\right)$, there always exists some $\zeta \in (0, 2)$ such that $\frac{1 - \sqrt{\zeta(1 - \zeta/4)}}{2} = \gamma$. Let $m \geq \frac{Cn^2}{\hat{\theta}^4\zeta}$ be a positive integer and $B_i, i = 1, 2, \cdots, 2m$ be a partition of the covariate space $\mathcal{X} = [0, 1]^K$ such that each set has a Lebesgue measure of $\frac{1}{2m}$. We also define

$$
\begin{aligned}
\lambda_i &\sim \text{Uniform}(\{-1, +1\}), \quad i = 1, 2, \cdots, M \\
X &\sim P_X = \text{Uniform}(\mathcal{X}) \\
T \mid X = x &\sim \text{Bernoulli}(g_\lambda(x)) \\
Y \mid T = t, X = x &\sim \text{Bernoulli}(f_\lambda(x) + \theta't),
\end{aligned}
\tag{28}
$$

where

$$
\begin{aligned}
\theta' &= (1 - \tilde{\epsilon}_1^2)^{-1}(\hat{\theta} + \tilde{\epsilon}_1\tilde{\epsilon}_2) \\
g_\lambda(x) &= \hat{g}(x) + \tilde{\epsilon}_1\sqrt{\hat{g}(x)(1 - \hat{g}(x))}\Delta(\lambda, x) \\
q_\lambda(x) &= \hat{q}(x) - \tilde{\epsilon}_2\sqrt{\hat{g}(x)(1 - \hat{g}(x))}\Delta(\lambda, x) \\
f_\lambda(x) &= q_\lambda(x) - \theta'g_\lambda(x)
\end{aligned}
\tag{29}
$$

and

$$
\Delta(\lambda, x) = \sum_{j=1}^{M/2} \lambda_j \left(\mathbb{1}\left\{x \in B_{2j-1}\right\} - \mathbb{1}\left\{x \in B_{2j}\right\}\right).
$$

Let $P_\lambda$ be the joint distribution of $(X, T, Y)$ induced by (28), $\mu$ be the uniform measure over $\mathcal{X} \times \mathcal{T} \times \mathcal{Y}$ and $p_\lambda = \frac{dP_\lambda}{d\mu}$. From (28) we can derive the expressions of $p_\lambda(x, t, y)$ as follows:

$$
p_\lambda(x, t, y) = g_\lambda(x)^t \left(1 - g_\lambda(x)\right)^{1-t} \left(f_\lambda(x) + \theta't\right)^y \left(1 - f_\lambda(x) - \theta't\right)^{1-y}.
\tag{30}
$$

Specifically, we have

$$
\begin{aligned}
p_\lambda(x,1,1) &= g_\lambda(x)\left(f_\lambda(x)+\theta'\right) \\
&= g_\lambda(x)\left(q_\lambda(x)+\theta'\left(1-g_\lambda(x)\right)\right) \\
&= \left[\hat{g}(x)\hat{q}(x)-\tilde{\epsilon}_1\tilde{\epsilon}_2\hat{g}(x)\left(1-\hat{g}(x)\right)+\theta'\left(\hat{g}(x)\left(1-\hat{g}(x)\right)-\tilde{\epsilon}_1^2\hat{g}(x)\left(1-\hat{g}(x)\right)\right)\right] \\
&\quad + \left(\tilde{\epsilon}_1\hat{q}(x)-\tilde{\epsilon}_2\hat{g}(x)+\tilde{\epsilon}_1-2\tilde{\epsilon}_1\hat{g}(x)\right)\sqrt{\hat{g}(x)(1-\hat{g}(x))}\Delta(\lambda,x) \\
&= \hat{g}(x)\hat{q}(x)+\hat{\theta}\hat{g}(x)\left(1-\hat{g}(x)\right) \hspace{4cm} \text{(31a)} \\
&\quad + \left(\tilde{\epsilon}_1\hat{q}(x)-\tilde{\epsilon}_2\hat{g}(x)+\tilde{\epsilon}_1-2\tilde{\epsilon}_1\hat{g}(x)\right)\sqrt{\hat{g}(x)(1-\hat{g}(x))}\Delta(\lambda,x) \\
&= \hat{p}(x,1,1)+\left(\tilde{\epsilon}_1\hat{q}(x)-\tilde{\epsilon}_2\hat{g}(x)+\tilde{\epsilon}_1-2\tilde{\epsilon}_1\hat{g}(x)\right)\sqrt{\hat{g}(x)(1-\hat{g}(x))}\Delta(\lambda,x) \quad \text{(31b)}
\end{aligned}
$$
$$
\begin{aligned}
p_\lambda(x,1,0) &= g_\lambda(x)-p_\lambda(x,1,1) \\
&= \hat{p}(x,1,0)+\left(\tilde{\epsilon}_1(1-\hat{q}(x))+\tilde{\epsilon}_2\hat{g}(x)-\tilde{\epsilon}_1+2\tilde{\epsilon}_1\hat{g}(x)\right)\sqrt{\hat{g}(x)(1-\hat{g}(x))}\Delta(\lambda,x)
\end{aligned}
$$
$$\text{(31c)}$$

$$
\begin{aligned}
p_\lambda(x,0,1) &= (1-g_\lambda(x))f_\lambda(x) \\
&= (1-g_\lambda(x))q_\lambda(x)-\theta' g_\lambda(x)(1-g_\lambda(x)) \\
&= q_\lambda(x)-p_\lambda(x,1,1) \\
&= \hat{p}(x,0,1)-\left(\tilde{\epsilon}_1\hat{q}(x)+\tilde{\epsilon}_2(1-\hat{g}(x))+\tilde{\epsilon}_1-2\tilde{\epsilon}_1\hat{g}(x)\right)\sqrt{\hat{g}(x)(1-\hat{g}(x))}\Delta(\lambda,x)
\end{aligned}
$$
$$\text{(31d)}$$

$$
\begin{aligned}
p_\lambda(x,0,0) &= 1-g_\lambda(x)-p_\lambda(x,0,1) \\
&= \hat{p}(x,0,0)+\left(\tilde{\epsilon}_1(1-\hat{q}(x))-\tilde{\epsilon}_2(1-\hat{g}(x))-\tilde{\epsilon}_1+2\tilde{\epsilon}_1\hat{g}(x)\right)\sqrt{\hat{g}(x)(1-\hat{g}(x))}\Delta(\lambda,x).
\end{aligned}
$$
$$\text{(31e)}$$

Crucially, all the $p_\lambda-\hat{p}$'s are linear functions of $\Delta(\lambda,x)$.

**Lemma B.1** ($L^2$-norm bounds). *Let* $\tilde{\epsilon}_1=A^{-1/2}\delta^{-1/2}\epsilon_1, \tilde{\epsilon}_2=A^{-1/2}\delta^{-1/2}\epsilon_2$, *then if* $\epsilon_1,\epsilon_2\leq\frac{\delta}{2}$, *it holds that*

$$
0\leq g_\lambda,q_\lambda\leq 1, \quad \|g_\lambda-\hat{g}\|_{L^2(P_X)}\leq\epsilon_1 \quad and \quad \|q_\lambda-\hat{q}\|_{L^2(P_X)}\leq\epsilon_2. \tag{32}
$$

*As a result,* $P_\lambda\in\mathcal{P}_{2,\epsilon}(\hat{h})$.

**Proof** : By definition, we have

$$
\|g_\lambda-\hat{g}\|_{L^2(P_X)}=\tilde{\epsilon}_1\mathbb{E}_{P_X}[\hat{g}(X)(1-\hat{g}(X))]=\epsilon_1.
$$

Similarly $\|q_\lambda-\hat{q}\|_{L^2(P_X)}=\epsilon_2$. Hence it directly follows that $P_\lambda\in\mathcal{P}_{2,\epsilon}(\hat{h})$. By our assumption, $\hat{g}(x)(1-\hat{g}(x))\geq A^{-1}\delta$, so that $A^{-1}\hat{g}(x)\geq\delta\geq\tilde{\epsilon}_1^2$ and

$$
g_\lambda(x)\geq\hat{g}(x)-\tilde{\epsilon}_1\sqrt{\hat{g}(x)(1-\hat{g}(x))}\geq\sqrt{\hat{g}(x)}(\sqrt{\hat{g}(x)}-\tilde{\epsilon}_1)\geq 0.
$$

By a similar argument, one can show that $g_\lambda(x)<1$ and also $q_\lambda(x)\in(0,1)$, concluding the proof.
$\square$

To apply Lemma A.1, we now use Lemma A.2 to bounding the Hellinger distance between $\hat{P}^{\otimes n}$ and $\int P_\lambda^{\otimes n}\mathrm{d}\pi(\lambda)$. We recall the following lemma:

**Lemma B.2** (Joint density lower bound). *If* $\epsilon_1,\epsilon_2\leq\frac{\delta}{8}$, *we have* $p_\lambda(x,t,y)\geq\frac{1}{8}\delta^2, \forall(x,t,y)\in\mathcal{X}\times\mathcal{T}\times\mathcal{Y}$.

**Proof** : Note that $\hat{f}(x)=\hat{q}(x)-\frac{\delta}{2}\hat{g}(x)\in\left[\frac{\delta}{2},1-\frac{\delta}{2}\right]$ and

$$
|f_\lambda(x)-\hat{f}(x)|\leq\epsilon_2+\theta'\epsilon_1\leq\frac{\delta}{4},
$$

so we have that $f_\lambda(x)\in\left[\frac{\delta}{4},1-\frac{\delta}{4}\right]$. Similarly, one can show that $f_\lambda(x)+\theta'\in\left[\frac{\delta}{4},1-\frac{\delta}{4}\right]$. By Lemma B.1, $g_\lambda(x)\in\left[\frac{\delta}{2},1-\frac{\delta}{2}\right]$, so the conclusion directly follows from (30). $\square$

**Lemma B.3** (Hellinger distance bound). *For any* $\zeta>0$, *as long as* $M\geq$, *it holds that*

$$
H\left(\hat{P}^{\otimes n},\int P_\lambda^{\otimes n}\mathrm{d}\pi(\lambda)\right)\leq\delta.
$$

***Proof*** : Let $\mathcal{Z} = \mathcal{X} \times \mathcal{T} \times \mathcal{Y}$, where $\mathcal{T} = \mathcal{Y} = \{-1, +1\}$. We apply Lemma A.2 with $\Lambda_i = \{-1, +1\}$, $\mathcal{Z}_j = (B_{2j-1} \cup B_{2j}) \times \{-1, +1\} \times \{-1, +1\}$, $P = \hat{P}, Q_\lambda = P_\lambda$ and $\pi$ being the uniform distribution on $\{-1, +1\}^m$. Firstly, since $\lambda_i \sim \text{Uniform}(\{-1, +1\})$, we have $\mathbb{E}_\pi[\Delta(\lambda, x)] = 0$ for any fixed $x \in \mathcal{X}$, so that (31) implies that $\mathbb{E}_\pi P_\lambda = \hat{P}$.

By our choice of $B_j$, we have $\hat{P}(\mathcal{Z}_j) = P_\lambda(\mathcal{Z}_j) = \frac{1}{m}$ for all $j$, so we have that $p_j = \frac{1}{m}$. Notice that

$$
\begin{aligned}
b &\leq m \max_j \mu(\mathcal{Z}_j) \cdot \sup_\lambda \sup_{(x,t,y) \in \mathcal{Z}} \frac{(p_\lambda(x,t,y) - \hat{p}(x,t,y))^2}{\hat{p}(x,t,y)} \\
&= \sup_\lambda \sup_{(x,t,y) \in \mathcal{Z}} \frac{(p_\lambda(x,t,y) - \hat{p}(x,t,y))^2}{\hat{p}(x,t,y)},
\end{aligned}
$$

where we recall that $\mu(\mathcal{Z}_j) = \frac{1}{m}$ since $\mu$ is the uniform distribution on $\mathcal{Z}$. When $t = y = 1$, we have

$$
\frac{(p_\lambda(x,t,y) - \hat{p}(x,t,y))^2}{\hat{p}(x,t,y)} \leq \frac{\hat{g}(x)(1 - \hat{g}(x))}{\hat{g}(x)(\hat{q}(x) + \hat{\theta}(1 - \hat{g}(x)))} \leq \hat{\theta}^{-1},
$$

where the last step holds since $\hat{q}(x) + 2\hat{\theta}(1 - \hat{g}(x)) \geq 0$ by our choice of $\hat{\theta}$ and $\hat{q}(x) \geq 0$. Similarly, we can deduce the same bound for $(t, y) \in \{(0, 1), (1, 0), (1, 1)\}$. Hence we have that $b \leq \hat{\theta}^{-1}$. We can choose $A = \hat{\theta}^{-1}$ and $n \max\{1, b\} \max_j p_j \leq \hat{\theta}^{-1} nm^{-1} \leq A$ is satisfied. Therefore, by Lemma A.2 we can deduce that

$$
H\left(\hat{P}^{\otimes n}, \int_\lambda P_\lambda^{\otimes n} \mathrm{d}\pi(\lambda)\right) \leq Cm^{-1}n^2 b^2 \leq C\hat{\theta}^{-2}m^{-1}n^2 \leq \zeta,
$$

where the last step holds since $m \geq \frac{Cn^2}{\hat{\theta}^2 \zeta}$. $\qquad\square$

Now we are ready to apply Lemma A.1. We choose $Z = (X, T, Y), P = \hat{P}, \mathcal{P}_1 = \{P_\lambda : \lambda \in \{-1, +1\}\}, \pi$ be the uniform distribution on $\mathcal{P}_1$, and $T$ that maps each observation distribution $P$ generated by (1) to the corresponding $\theta$. Then we have that

$$
T(P) = \hat{\theta}, \quad \pi\left(\{Q : T(Q) = \hat{\theta} + 2s\}\right) = 1,
$$

where $s = \frac{1}{2}(\theta'\tilde{\epsilon}_1^2 + \tilde{\epsilon}_1\tilde{\epsilon}_2) = (4A\delta)^{-1}(c_q\epsilon_1^2 + \epsilon_1\epsilon_2)$. Moreover, our choice of $m$ and Lemma B.3 together implies that $H^2\left(P^{\otimes n}, \int Q^{\otimes n} \mathrm{d}\pi(Q)\right) \leq \zeta$. Therefore, Lemma A.1 implies that for any estimator $\hat{T}$, it holds that

$$
\sup_{P \in \mathcal{P}} P\left[\left|\hat{T} - T(P)\right| \geq s\right] \geq \frac{1 - \sqrt{\zeta(1 - \zeta/4)}}{2} = \gamma.
$$

Equivalently, we have

$$
\mathfrak{M}_{1-\gamma}\left(\mathcal{P}_{2,\epsilon}(\hat{h})\right) \geq \frac{1}{4}A^{-1}\delta^{-1}(c_q\epsilon_1^2 + \epsilon_1\epsilon_2). \tag{33}
$$

We now proceed to prove the $n^{-1/2}$ component of the lower bound. Define

$$
\tilde{q}(x) = \hat{q}(x) + \epsilon\sqrt{\hat{g}(x)(1 - \hat{g}(x))}, \quad \tilde{\theta} = \hat{\theta} + \epsilon \tag{34}
$$

and let $\tilde{P}$ be the distribution generated by

$$
x \sim P_X, \quad T \mid X = x \sim \text{Bernoulli}(\hat{g}(x)), \quad Y \mid X = x, T = t \sim \text{Bernoulli}(\tilde{q}(x) - \hat{g}(x) + \tilde{\theta}t)
$$

and $\tilde{p}(x, t, y)$ be the density. Then we have that

$$
\begin{aligned}
\tilde{p}(x, 1, 1) &= \hat{g}(x)\left(\tilde{q}(x) + \tilde{\theta}(1 - \hat{g}(x))\right) \\
&= \hat{g}(x)\left[\hat{q}(x) + \hat{\theta}(1 - \hat{g}(x)) + \epsilon\sqrt{\hat{g}(x)(1 - \hat{g}(x))} + \epsilon(1 - \hat{g}(x))\right] \\
&= \hat{p}(x, 1, 1) + \epsilon\hat{g}(x)\sqrt{1 - \hat{g}(x)}\left(\sqrt{\hat{g}(x)} + \sqrt{1 - \hat{g}(x)}\right),
\end{aligned}
$$

so that

$$
\frac{(\tilde{p}(x, 1, 1) - \hat{p}(x, 1, 1))^2}{\hat{p}(x, 1, 1)} \leq \epsilon^2 \frac{\hat{g}(x)(1 - \hat{g}(x))}{\hat{q}(x) + \hat{\theta}(1 - \hat{g}(x))} \leq c_q^{-1}\epsilon^2 \hat{g}(x)(1 - \hat{g}(x)).
$$

We also have that

$$\tilde{p}(x,1,0) = \hat{g}(x)\big[1 - \tilde{q}(x) - \tilde{\theta}(1-\hat{g}(x))\big]$$

$$= \hat{g}(x)\big[1 - \hat{q}(x) - \hat{\theta}(1-\hat{g}(x)) - \epsilon\sqrt{\hat{g}(x)(1-\hat{g}(x))} - \epsilon(1-\hat{g}(x))\big]$$

$$= \hat{p}(x,1,0) - \epsilon\hat{g}(x)\sqrt{1-\hat{g}(x)}\big(\sqrt{\hat{g}(x)} + \sqrt{1-\hat{g}(x)}\big),$$

so that

$$\frac{(\tilde{p}(x,1,0) - \hat{p}(x,1,0))^2}{\hat{p}(x,1,1)} \leq \epsilon^2 \frac{\hat{g}(x)(1-\hat{g}(x))}{1 - \hat{q}(x) - \hat{\theta}(1-\hat{g}(x))} \leq 4c_q^{-1}\epsilon^2\hat{g}(x)(1-\hat{g}(x)),$$

where we use $\hat{\theta} = \frac{1}{2}c_q$ and $1 - \hat{q}(x) \geq c_q$ in the last step. Similarly, one can show that

$$\frac{(\tilde{p}(x,0,0) - \hat{p}(x,0,0))^2}{\hat{p}(x,0,0)}, \frac{(\tilde{p}(x,0,1) - \hat{p}(x,0,1))^2}{\hat{p}(x,0,1)} \leq 4c_q^{-1}\epsilon^2\hat{g}(x)(1-\hat{g}(x)).$$

Combining all the inequalities above, we have

$$\chi^2(\tilde{P}, \hat{P}) \leq 10c_q^{-1}\epsilon^2\mathbb{E}[\hat{g}(X)(1-\hat{g}(X))] = 10c_q^{-1}\epsilon^2\delta.$$

By choosing $\epsilon = 0.1\zeta c_q^{1/2}\delta^{-1/2}n^{-1/2}$, it holds that

$$H\big(\tilde{P}^{\otimes n}, \hat{P}^{\otimes n}\big) \leq nH(\tilde{P}, \hat{P}) \leq n\chi^2(\tilde{P}, \hat{P}) \leq \zeta,$$

so [Lemma A.1](#) directly implies that

Therefore, [Lemma A.1](#) implies that for any estimator $\hat{T}$, it holds that

$$\sup_{P\in\mathcal{P}} P\left[\left|\hat{T} - T(P)\right| \geq \epsilon/2\right] \geq \frac{1 - \sqrt{\zeta(1-\zeta/4)}}{2} = \gamma.$$

Equivalently, we have

$$\mathfrak{M}_{1-\gamma}\left(\mathcal{P}_{2,\epsilon}(\hat{h})\right) \geq \epsilon/2 = 0.05\zeta c_q^{1/2}\delta^{-1/2}n^{-1/2}. \tag{35}$$

Combining (33) and (35), we obtain the desired result.

## B.1 Some remarks on the constants

Notice that the assumptions that we make for deriving the upper and lower bounds are not exactly the same. Here we discuss some important aspects of their differences.

**Remark B.1** (Assumptions on the uniform overlap)**.** *Compared with [Theorem C.1](#), we make the additional assumption that $\hat{g}(x)(1 - \hat{g}(x)) \geq A^{-1}\delta$, which we will refer to as uniform overlap. Equivalently, this assumption states that there exists some $\delta_1 = \Theta(A^{-1}\delta)$ such that $\delta_1 \leq \hat{g}(x) \leq 1 - \delta_1$. This assumption is also made in previous work [[Jin and Syrgkanis, 2024](#)], albeit for a different causal estimand. If $\delta_1$ is treated as a universal constant, then we can choose $A = \delta_1^{-1}$ and the expected overlap assumption is satisfied. In this case, our lower bound has tight dependency on both $\epsilon_i, i = 1, 2$ and $\delta$. However, the error rate of DML does not match the lower bound in $\delta$ if $\hat{g}$ is not uniformly extreme. The minimax optimal rate is unknown in this regime.*

**Remark B.2** (Assumptions on $\hat{q}(\cdot)$)**.** *Compared with [Theorem C.1](#), another additional assumption that we make is that $c_q \leq \hat{q}(x) \leq 1 - c_q$ in [Theorem 3.1](#). This assumption is also needed in the previous lower bound established in [Balakrishnan et al. [2023]](#) for $\mathbb{E}[\mathrm{Cov}(Y, T \mid X)]$. Interestingly, this assumption is not needed for upper bound. The $c_q\epsilon_1^2$ term in our lower bound (5) corresponds to the $C_\theta\epsilon_1^2$ term in our upper bound, where we recall that $C_\theta$ is an upper bound on the ground-truth $\theta_0$. In fact, the overlap assumption on $\hat{q}(x)$ also implicitly imposes a constraint on the magnitude of the ground-truth parameter $\theta_0$; more discussions can be found in [Appendix B.2](#).*

## B.2 Discussion of the constant $c_q$

For any pair of functions $(g, q)$ that take values in $[0, 1]$, we define their *cross ratio* to be

$$\Psi(g, q) = \max\left\{\min_{x\in\mathcal{X}}\min\left\{\frac{q(x)}{g(x)}, \frac{1-q(x)}{1-g(x)}\right\}, \min_{x\in\mathcal{X}}\min\left\{\frac{1-q(x)}{g(x)}, \frac{q(x)}{1-g(x)}\right\}\right\}.$$

First, note that [Theorem 3.1](#) can be slightly strengthened as follows:

**Theorem B.1** (Strengthened binary lower bound). *Let $c_q, \delta \in (0, \frac{1}{4})$ and $\mathcal{P}$ be the set of all possible $P$'s generated by (1), such that the variables $T, Y$ are binary and that the marginal distribution of $P$ on $\mathcal{X}$ is $P_X$. Let $\Phi$ be a mapping that maps a distribution $P \in \mathcal{P}$ to the nuisance functions $h_0 = (g_0, q_0) \in \mathcal{H} = \mathrm{range}(\Phi)$. Then for any $\gamma \in (1/2, 1)$, there exists a constant $c_\gamma > 0$ such that for any $\epsilon_i \le \delta/2, i = 1, 2$ and any estimates $\hat{h} = (\hat{g}, \hat{q})$ with $\mathbb{E}_{P_X}[\hat{g}(X)(1 - \hat{g}(X))] = 2\delta$ and $\hat{g}(x)(1 - \hat{g}(x)) \ge A^{-1}\delta, \forall x \in \mathcal{X}$, we have*

$$\mathfrak{M}_{n,1-\gamma}\left(\mathcal{P}_{2,\epsilon}(\hat{h})\right) \ge c_\gamma A^{-1}\delta^{-1}(c_\theta \epsilon_1^2 + \epsilon_1 \epsilon_2), \tag{36}$$

*where $\gamma$ is a universal constant that only depends on $\gamma$, and*

$$c_\theta = \sup\left\{\Psi(g,q) \mid \|g - \hat{g}\|_{L^2(P_X)} \le \epsilon_1/2, \|q - \hat{q}\|_{L^2(P_X)} \le \epsilon_2/2\right\}.$$

**Proof** : Without loss of generality, we assume that

$$\Psi(g,q) = \min_{x \in \mathcal{X}} \min\left\{\frac{1 - q(x)}{g(x)}, \frac{q(x)}{1 - g(x)}\right\}.$$

For any $h = (g, q) \in \mathcal{P}_{2,\epsilon/2}(\hat{h})$, note that

$$\mathcal{P}_{2,\epsilon/2}(h) \subseteq \mathcal{P}_{2,\epsilon}(\hat{h}),$$

so that

$$\mathfrak{M}_{n,1-\gamma}\left(\mathcal{P}_{2,\epsilon}(\hat{h})\right) \ge \mathfrak{M}_{n,1-\gamma}\left(\mathcal{P}_{2,\epsilon/2}(h)\right).$$

Now it suffices to show that

$$\mathfrak{M}_{n,1-\gamma}\left(\mathcal{P}_{2,\epsilon/2}(h)\right) \ge c_\gamma A^{-1}\delta^{-1}(\Psi(g,q)\epsilon_1^2 + \epsilon_1 \epsilon_2). \tag{37}$$

Notice that this lower bound can be derived with exactly the same argument as we employed in the previous section, since the only place that we use the assumption $c_q \le \hat{q}(x) \le 1 - c_q$ is that

$$\hat{q}(x) - 2\hat{\theta}\hat{g}(x), \hat{q}(x) + 2\hat{\theta}(1 - \hat{g}(x)) \in [0, 1].$$

Now we replace $\hat{g}, \hat{q}$ with $g, q$ respectively, and choosing $\hat{\theta} = c_\theta/2$ ensures that the above relationship holds. Therefore we obtain the desired lower bound. $\square$

The upper bound side can also be strengthened by replacing $C_\theta$ with

$$C'_\theta = \sup\left\{\Psi(g,q) \mid \|g - \hat{g}\|_{L^2(P_X)} \le \epsilon_1, \|q - \hat{q}\|_{L^2(P_X)} \le \epsilon_2\right\}.$$

The main insight is that the nuisance estimates already tells us that $|\theta| \le C'_\theta$.

**Theorem B.2** (Strengthened binary upper bound). *Let $\delta > 0$ and $\mathcal{P}$ be the set of all distributions $P$ of $(X, T, Y)$ generated from (1) that satisfies $\mathbb{E}_P[(T - g_0(X))^2] \ge \delta$ and $T, Y$ are binary. Let $\Phi$ be a mapping that maps each $P \in \mathcal{P}$ to $(g_0, q_0) \in \mathcal{H} = \mathrm{range}(\mathcal{H})$. Then there exists a constant $n_0 = n_0(\delta)$ such that when $n \ge n_0$, for any estimates $\hat{h} = (\hat{g}, \hat{q})$ and any $\gamma \in (0, 1)$, the DML estimator $\vartheta_{\mathrm{DML}}$ derived from the moment function*

$$m(Z, \theta_0, h(X)) = [Y - q(X) - \theta_0(T - g(X))](T - g(X)), \quad h = (g, q) \tag{38}$$

*satisfies*

$$\mathfrak{R}_{n,1-\gamma}(\hat{\vartheta}_{\mathrm{DML}}; \mathcal{P}_{2,\epsilon}(\hat{h})) \le A_\gamma \left[\delta^{-1}(C'_\theta \epsilon_1^2 + \epsilon_1 \epsilon_2) + \left(\delta^{-1}(C'_\theta \epsilon_1 + \epsilon_2) + \delta^{-1/2}\right)n^{-1/2}\right]$$

*for any $\gamma \in (0, 1)$, where $A_\gamma$ is a constant that only depends on $\gamma$.*

**Proof** : We prove the theorem by applying Theorem C.1. First, since $T$ and $Y$ are binary, we can take $C = 1$. It remains to show that for any $P \in \mathcal{P}_{2,\epsilon}(\hat{h})$, the corresponding $\theta_0$ is bounded by $C'_\theta$. Indeed, let $g_0(x) = \mathbb{E}_P[T \mid X = x]$ and $q_0(x) = \mathbb{E}_P[T \mid X = x]$, then $\|g_0 - \hat{g}\|_{L^2(P_X)} \le \epsilon_1$ and $\|q_0 - \hat{q}\|_{L^2(P_X)} \le \epsilon_2$. Moreover, note that

$$q_0(x) - \theta g_0(x), q_0(x) + \theta(1 - g_0(x)) \in [0, 1], \quad \forall x \in \mathcal{X},$$

so we have that

$$\theta_0 \le \Psi(g_0, q_0) \le C'_\theta,$$

concluding the proof. $\square$

# C Proofs of upper bounds for DML

In this section, we present the formal statements and proofs of Theorem C.1 and Theorem C.2. Both results are already known in the literature, and we present here the explicit structure-agnostic rates for completeness.

**Theorem C.1** (Structure-agnostic rate of DML). *Let $\delta, C_\theta, C_T, C_Y > 0$, $\mathcal{P}_0 = \{P_0 \in \mathcal{P}_0^\star(C_\theta, C_T, C_Y) : \mathbb{E}_{P_0}[(T - g_0(X))^2] \geq \delta\}$, and $\Phi = \Phi^\star$. Then for any estimates $\hat{h} = (\hat{g}, \hat{q})$ and any $\gamma \in (0,1)$ such that $|\hat{g}(X)| \leq C_T$ a.s., the DML estimator $\vartheta_{\mathrm{DML}}$ derived from the moment function (38) satisfies*

$$\mathfrak{R}_{n,1-\gamma}(\hat{\vartheta}_{\mathrm{DML}}; \mathcal{P}_{2,\epsilon}(\hat{h})) \leq 4\delta^{-1}(C_\theta \epsilon_g^2 + \epsilon_g \epsilon_q) + C_\gamma \big[\delta^{-1}(C_T + C_Y)\epsilon_g + C_Y \delta^{-1/2}\big]n^{-1/2}$$

*for any $\gamma \in (0,1)$ and $\delta \geq 15 C_T^{1/2} n^{-1/2}$, where $C_\gamma$ is a constant that only depends on $\gamma$.*

The proof of this result follows the standard arguments in the DML literature and can be found in Appendix C.1. Existing theoretical guarantees largely focus on establishing sufficient conditions for achieving $\mathcal{O}(n^{-1/2})$ rate and establishing confidence intervals [Chernozhukov et al., 2018, 2022], while here we make the dependency of the error on $\epsilon_i, i = 1, 2$ and $\delta$ explicit, which would be of interest when the rate is slower than $n^{-1/2}$. The existence a constant $\delta > 0$ that satisfies the assumption in Theorem C.1 is commonly referred to as the overlap assumption and is widely adopted in the DML literature. The estimation error still be large if $\delta$ is small compared to $\epsilon_g, \epsilon_q$. When the assumption on $\delta$ is violated, *i.e.*, $\delta = \mathcal{O}(n^{-1/2})$, the second term in the upper bound becomes $\Omega(\epsilon_g)$, so that DML is not better than the naive estimator $\frac{1}{n} \sum_{i=1}^n (\hat{g}(1, X_i) - \hat{g}(0, X_i))$.

On the other hand, since $\sigma$ is known, the following upper bound can be easily established for a modified version of DML, using the moment function

$$m(Z, \theta_0, h_0(X)) = (Y - q_0(X))(T - g_0(X)) - \sigma^2 \theta. \tag{39}$$

**Theorem C.2** (Structure-agnostic upper bound with known treatment noise). *Let $\Phi, \mathcal{H}, \hat{h}$ be defined as in Theorem 3.2, and $s_1, s_2 > 0$ satisfy $s_1^{-1} + s_2^{-1} \leq 1$, then the estimator $\tilde{\vartheta}_{\mathrm{DML}}$ derived from the moment function (39) satisfies*

$$\mathfrak{R}_{1-\gamma}(\hat{\vartheta}_{\mathrm{DML}}; \mathcal{P}_{s,\epsilon}(\hat{h})) \leq C_\gamma \left[\sigma^{-2}\epsilon_1 \epsilon_2 + (C_\theta \sigma^{-2}\epsilon_1 + \sigma^{-1})n^{-1/2}\right]$$

*for any $\gamma \in (0,1)$, where $C_\gamma$ is a constant that only depends on $\gamma$.*

Theorem C.2 can be derived in a similar way as Theorem C.1; the proof can be found in Appendix C.2. Since it considers a simplified setting where the treatment variance is known in $X$, the $\epsilon_1^2$ term in Theorem C.1 induced by estimating the treatment variance vanishes in the current upper bound.

When the nuisance error of $q_0$ is small, *i.e.*, $\epsilon_2 \leq C_\theta$, the upper bound matches the lower bound derived in Theorem 3.2 up to logarithmic factors. Moreover, if the estimation error $\epsilon_g$ of $g_0$ is polynomial in $n$, then they differ by only a constant factor, implying that DML is minimax optimal.

## C.1 Proof of Theorem C.1: Structure-agnostic rate of DML

Given data $\{(X_i, T_i, Y_i)\}_{i=1}^n$, the DML estimator is defined by

$$\hat{\theta}_{\mathrm{DML}} := \left(n^{-1} \sum_{i=1}^n (T_i - \hat{g}(X_i))^2\right)^{-1} \left(n^{-1} \sum_{i=1}^n (Y_i - \hat{q}(X_i))(T_i - \hat{g}(X_i))\right).$$

Let $\eta_i = T_i - g_0(X_i), \epsilon_i = Y_i - q_0(X_i) - \theta_0 \eta_i, \Delta_g = \hat{g} - g$ and $\Delta_q = \hat{q} - q$, then

$$\hat{\theta}_{\mathrm{DML}} = \left(n^{-1} \sum_{i=1}^n (\Delta_g(X_i) - \eta_i)^2\right)^{-1} \left(n^{-1} \sum_{i=1}^n (\Delta_q(X_i) - \epsilon_i - \theta_0 \eta_i)(\Delta_g(X_i) - \eta_i)\right).$$

For any $\gamma \in (0,1)$, by assumption there exists some constant $N_\gamma$ such that for any $n \geq N_\gamma$, we have

$$\delta^2 \geq 100 C \gamma^{-1} n^{-1} \tag{40}$$

Since $\{\Delta_g(X_i)\eta_i\}_{i=1}^n$ are i.i.d random variables with $\mathbb{E}[\Delta_g(X_i)\eta_i] = \mathbb{E}[\Delta_g(X_i)\mathbb{E}[\eta_i \mid X_i]] = 0$ and

$$\mathbb{E}[\Delta_g(X_i)^2\eta_i^2] = \mathbb{E}[\Delta_g(X_i)^2\mathbb{E}[\eta_i^2 \mid X_i]] \leq 4C_\mathsf{T}^2\mathbb{E}[\Delta_g(X_i)^2] \leq 4C_\mathsf{T}^2\epsilon_g^2$$

we have

$$\mathbb{P}\left[\left|n^{-1/2}\sum_{i=1}^n \Delta_g(X_i)\eta_i\right| \leq 2AC_\mathsf{T}\epsilon_g\right] \geq 1 - A^{-2}$$

by Chebyshev's inequality, where $A = 0.1\gamma^{-1/2}$. Similarly, we also have

$$\mathbb{P}\left[\left|n^{-1/2}\sum_{i=1}^n \Delta_q(X_i)\eta_i\right| \leq 2AC_\mathsf{T}\epsilon_q\right] \geq 1-A^{-2}, \mathbb{P}\left[\left|n^{-1/2}\sum_{i=1}^n \Delta_g(X_i)\epsilon_i\right| \leq 2AC_\mathsf{T}\epsilon_g\right] \geq 1-A^{-2}.$$

From

$$\mathbb{E}[\epsilon_i^2\eta_i^2] \leq 4C_\mathsf{Y}^2\mathbb{E}[\eta^2]$$

we deduce that

$$\mathbb{P}\left[\left|n^{-1/2}\sum_{i=1}^n \epsilon_i\eta_i\right| \leq 2AC_\mathsf{Y}\mathbb{E}[\eta^2]^{1/2}\right] \geq 1 - A^{-2}.$$

Since $\mathbb{E}[|\Delta_g(X)\Delta_q(X)|] \leq \epsilon_g\epsilon_q$ by Cauchy-Schwarz inequality, Chebyshev's inequality again implies that

$$\mathbb{P}\left[\left|n^{-1}\sum_{i=1}^n \Delta_g(X)\Delta_q(X)\right| \leq \epsilon_g\epsilon_q + An^{-1/2}\mathbb{E}[\Delta_g(X)^2\Delta_q(X)^2]^{1/2}\right] \geq 1 - A^{-2}.$$

with a similar reasoning, we have

$$\mathbb{P}\left[\left|n^{-1}\sum_{i=1}^n \Delta_g(X)^2\right| \leq \epsilon_g^2 + An^{-1/2}\mathbb{E}[\Delta_g(X)^4]^{1/2}\right] \geq 1 - A^{-2}.$$

Lastly, since $\mathbb{E}[\eta_i^2] \geq \delta$ by assumption, we also have that

$$\mathbb{P}\left[\left|n^{-1}\sum_{i=1}^n \eta_i^2 - \mathbb{E}[\eta^2]\right| \leq \frac{1}{2}\mathbb{E}[\eta^2]\right] \geq 1 - 4n^{-1}\frac{\mathbb{E}[(\eta^2 - \mathbb{E}[\eta^2])^2]}{\mathbb{E}[\eta^2]^2} \geq 1 - 8C_\mathsf{T}n^{-1}\delta^{-2} \geq 1 - 0.04\gamma,$$

so that

$$\mathbb{P}\left[n^{-1}\sum_{i=1}^n \eta_i^2 \geq \frac{1}{2}\mathbb{E}[\eta^2]\right] \geq 1 - 0.04\gamma.$$

Let $\mathcal{E}$ be the event that all the above high-probability bounds hold, then

$$\mathbb{P}[\mathcal{E}] \geq 1 - 5A^{-2} - 0.04\gamma > 1 - \gamma.$$

Under $\mathcal{E}$, we have

$$n^{-1}\sum_{i=1}^n (\Delta_g(X_i) - \eta_i)^2 \geq \frac{1}{2}\mathbb{E}[\eta^2] - 2AC_\mathsf{T}\epsilon_g n^{-1/2} \geq \frac{1}{4}\mathbb{E}[\eta^2],$$

since $\mathbb{E}[\eta^2] \geq \delta$ by assumption and (40) implies that $\delta \geq 4ACn^{-1/2}$. Moreover,

$$\left|n^{-1}\sum_{i=1}^n (\Delta_q(X_i) - \epsilon_i - \theta_0\Delta_g(X_i))(\Delta_g(X_i) - \eta_i)\right|$$

$$\leq |\theta_0|\epsilon_g^2 + \epsilon_g\epsilon_q + An^{-1/2}\left(\mathbb{E}[\Delta_g(X)^2\Delta_q(X)^2]^{1/2} + |\theta_0|\mathbb{E}[\Delta_g(X)^4]^{1/2} + 2C_\mathsf{Y}\mathbb{E}[\eta^2]^{1/2}\right).$$

It is easy to see that $\mathbb{E}[\Delta_g(X)^2\Delta_q(X)^2] \leq 4C_\mathsf{Y}^2\epsilon_g^2$ and $\mathbb{E}[\Delta_g(X)^4] \leq 4C_\mathsf{T}^2\epsilon_g^2$. As a result, we can deduce that

$$|\hat{\theta}_\mathrm{DML} - \theta_0|$$

$$= \left|\left(n^{-1}\sum_{i=1}^n (\Delta_g(X_i) - \eta_i)^2\right)^{-1}\left(n^{-1}\sum_{i=1}^n (\Delta_q(X_i) - \epsilon_i - \theta_0\Delta_g(X_i))(\Delta_g(X_i) - \eta_i)\right)\right|$$

$$\leq 4\delta^{-1}(C_\theta\epsilon_g^2 + \epsilon_g\epsilon_q) + 8A\left[\delta^{-1}(C_\mathsf{T} + C_\mathsf{Y})\epsilon_g + C_\mathsf{Y}\delta^{-1/2}\right]n^{-1/2},$$

concluding the proof.

## C.2 Proof of Theorem C.2: Structure-agnostic upper bound with known treatment noise

Since the variance of $\eta \mid X$ is assumed to be $\sigma^2$ where $\sigma$ is a known constant, we have

$$\hat{\theta}_{\text{DML}} = \sigma^{-2} \left( n^{-1} \sum_{i=1}^{n} (\Delta_q(X_i) - \epsilon_i - \theta_0 \eta_i)(\Delta_g(X_i) - \eta_i) \right),$$

so that

$$\hat{\theta}_{\text{DML}} - \theta_0$$
$$= \sigma^{-2} \left( n^{-1} \sum_{i=1}^{n} (\Delta_q(X_i) - \epsilon_i)(\Delta_g(X_i) - \eta_i) - \theta_0 \Delta_g(X_i) \eta_i + \theta_0 \left( n^{-1} \sum_{i=1}^{n} \eta_i^2 - \sigma^2 \right) \right).$$

Note that the high-probability bounds for each term in the above expression can be obtained with similar arguments as in the previous subsection, with $C = \Theta(\sigma)$ and $\delta = \sigma^2$. Hence it is straightforward to deduce that

$$|\hat{\theta}_{\text{DML}} - \theta_0| \lesssim A\sigma^{-2}\epsilon_g\epsilon_q + AC\big(C_\theta\sigma^{-2}\epsilon_g + \sigma^{-1}\big)n^{-1/2}.$$

## D Proof of Theorem 3.2: The Gaussian treatment barrier

Our proof is based on a constrained risk inequality for testing composite hypothesis developed in Cai and Low [2011].

**Lemma D.1.** *[Cai and Low, 2011, Corollary 1] Let $X$ be an observation with distribution $P \in \mathcal{P}$ and $\mathcal{P}_i, i = 0, 1$ be two subsets of $\mathcal{P}$ satisfying $\mathcal{P}_1 \cup \mathcal{P}_2 = \mathcal{P}$, and $\mu_i$ be some distribution supported on $\mathcal{P}$. Define*

$$m_i = \int T(P)\mu_i(\mathrm{d}P), \quad v_i^2 = \int \big(T(P) - m_i\big)^2 \mu_i(\mathrm{d}P) \tag{41}$$

*to be the mean and variance of a functional $T : \mathcal{P} \mapsto \mathbb{R}$, $F_i$ be the distribution of $X$ with prior $\mu_i$ and $f_i$ be its density with respect to some common dominating measure $\mu$. Then for any estimator $\hat{T}(X)$ we have that*

$$\sup_{P \in \mathcal{P}} \mathbb{E}_P\big[(\hat{T}(X) - T(P))^2\big] \geq \frac{\big(|m_1 - m_0| - v_0 I\big)^2}{(I+2)^2}$$

*as long as $|m_1 - m_0| - v_0 I \geq 0$, where $I = \left( \mathbb{E}_{f_0}\left[ \left( \frac{f_1(X)}{f_0(X)} - 1 \right)^2 \right] \right)^{1/2}$ is the $\chi^2$-distance between $F_0$ and $F_1$.*

To apply this inequality, we construct the null and alternative hypotheses as mixtures of data distributions with matching moments. While moment matching techniques are widely adopted in establishing minimax lower bounds, the structural nature of our causal model (1) brings additional challenges to our construction. Unlike most existing works where only moments of a single variable need to be matched, here we need to match moments that contain two variables: we seek for distributions $\nu_0, \nu_1$ over $\mathcal{P}_{s,\epsilon}(\hat{h})$ with corresponding mixtures $\bar{P}_0, \bar{P}_1$ respectively, such that both

$$\mathbb{E}_{\bar{P}_0}\big[(Y - \mathbb{E}_{\bar{P}_0}[Y \mid X])(T - \mathbb{E}_{\bar{P}_0}[T \mid X])^k\big] = \mathbb{E}_{\bar{P}_1}\big[(Y - \mathbb{E}_{\bar{P}_1}[Y \mid X])(T - \mathbb{E}_{\bar{P}_1}[T \mid X])^k\big]$$

and

$$\mathbb{E}_{\bar{P}_0}\big[(T - \mathbb{E}_{\bar{P}_0}[T \mid X])^k\big] = \mathbb{E}_{\bar{P}_1}\big[(T - \mathbb{E}_{\bar{P}_1}[T \mid X])^k\big]$$

hold for $k = 1, 2, \cdots, k_n$. This would imply that $\chi^2(\bar{P}_0 || \bar{P}_1)$ is small, which further implies that $\chi^2(\bar{P}_0^{\otimes n} || \bar{P}_1^{\otimes n})$ is also small, where $\bar{P}_i^{\otimes n} := \int P^{\otimes n} \mathrm{d}\nu_i, i = 1, 2$ and $P^{\otimes}$ is the $n$-fold product distribution.

To apply Lemma D.1, we need to show that there exists a sufficient gap between $m_0$ and $m_1$, which correspond to the expected value of $\theta$ under $\nu_0$ and $\nu_1$ respectively. Our key insight is that, for Gaussian treatment, there is no need to match $\mathbb{E}\left[(Y - \mathbb{E}[Y \mid X])(T - \mathbb{E}[T \mid X])\right]$ since this term always vanishes. This fact is due to a recursive property of the Hermite polynomial $H_k(x) = $

$(-1)^k \varphi^{(k)}(z)/\varphi(z)$ (where $\varphi(\cdot)$ is the Gaussian density); we will elaborate on this connection in Lemma D.6. As a result, we can construct mixtures of distributions that are close in terms of $\chi^2$-distance (Corollary D.1) but their average values of the $\mathbb{E}\left[(Y - \mathbb{E}[Y \mid X])(T - \mathbb{E}[T \mid X])\right]$ term are well-separated. Given the structure-agnostic oracle, this separation can be as large as $\tilde{\Omega}(\epsilon_g \epsilon_q)$, and it further induces a separation between $m_0$ and $m_1$ at the same scale, yielding the desired lower bound.

In the following, we present the full proof of this theorem.

The following lemma turns out to be a useful tool for moment matching in establishing our lower bounds.

**Lemma D.2** ($L_\infty$-distance to univariate polynomial bases). *Let $\mathfrak{P}_k$ be a linear space of polynomials on $[-1, 1]$ in the form of $\sum_{i=1}^m a_i \lambda^{u_i}$ where $u_i \in \{0, 1, \cdots, k\} \setminus \{1\}$, and $\delta_k$ be the $L_\infty$-distance of $a(\lambda) = \lambda$ to $\mathfrak{P}_k$. Then $\delta_k \geq \frac{1}{2k^3}$.*

**Proof :** Let $b(\lambda) \in \mathfrak{P}_k$ be a polynomial that satisfies $\|a - b\|_{L_\infty} = \delta_k$. Define $r = a - b$, then $r$ satisfies $\|r\|_{L_\infty} = \delta_k$ and $r'(0) = 1$.

Since $\deg(r) \leq k$, the Lagrange interpolation formula implies that

$$r(x) = \sum_{i=1}^{2k} r(x_i) \frac{\prod_{j \neq i}(x - x_j)}{\prod_{j \neq i}(x_i - x_j)}$$

for any $x_i \in [-1, 1], 1 \leq i \leq 2k$. Taking the derivative of both sides, we obtain

$$r'(x) = \sum_{i=1}^{2k} r(x_i) \frac{\sum_{l \neq i} \prod_{j \neq i,l}(x - x_j)}{\prod_{j \neq i}(x_i - x_j)}.$$

In particular, we choose

$$x_i = \begin{cases} -\dfrac{k + 1 - i}{k} & 1 \leq i \leq k \\ \dfrac{i - k}{k} & k + 1 \leq i \leq 2k, \end{cases}$$

then it holds that

$$\left| \frac{\prod_{j \neq i,l} x_j}{\prod_{j \neq i}(x_i - x_j)} \right| = \frac{(k!)^2}{ilk^{2k-2}} \frac{ik^{2k-1}}{(k-i)!(k+i)!} = \frac{k}{l} \frac{(k!)^2}{(k-i)!(k+i)!} \leq k.$$

As a result, we have

$$1 = |r'(0)| \leq \sum_{i=1}^{2k} \delta_k \sum_{l \neq i} \left| \frac{\prod_{j \neq i,l} x_j}{\prod_{j \neq i}(x_i - x_j)} \right| \leq 2k^3 \delta_k.$$

Hence $\delta_k \geq \frac{1}{2k^3}$ as desired. $\square$

**Lemma D.3** ($L_\infty$-distance to bivariate polynomial bases). *Let $\mathfrak{P}_{k,1}$ be a linear space of polynomials on $[-1, 1]^2$ of the form $\sum_{i=1}^m a_i \lambda^{u_i} \rho^{v_i}$ where $(u_i, v_i) \in \{0, 1, \cdots, k\} \times \{0, 1\} \setminus \{(1, 1)\}$, and $\delta_{k,1}$ be the $L_\infty$-distance from $a_1(\lambda, \rho) = \lambda\rho$ to $\mathfrak{P}_{k,1}$. Then $\delta_{k,1} \geq \frac{1}{2}\delta_k$.*

**Proof :** Assume the contrary holds, *i.e.* $\delta_{k,1} < \frac{1}{2}\delta_k$, then there exists some $b_1(\lambda, \rho) \in \mathfrak{P}_{k,1}$ such that $\|b_1 - a_1\|_{L_\infty} < \frac{1}{2}\delta_k$. By definition, there exists polynomials $r_1 \in \mathfrak{P}_k$ and $s_1$ such that $b_1(\lambda, \rho) = \rho r_1(\lambda) + s_1(\lambda)$. In particular, setting $\rho = 1$ and $\rho = -1$ implies that $\|r_1 + s_1 - \lambda\|_{L_\infty} < \frac{1}{2}\delta_k$ and $\|r_1 - s_1 - \lambda\|_{L_\infty} < \frac{1}{2}\delta_k$. The triangle inequality implies that $\|r_1 - \lambda\|_{L_\infty} < \delta_k$, which is a contradiction to the definition of $\delta_k$. Thus the conclusion follows. $\square$

**Lemma D.4** (Separation of measures under matching properties). *There exists two probability measures $\nu_0$ and $\nu_1$ on $[-1, 1]^2$ such that*

$$\int a(\lambda, \rho)\nu_0(\mathrm{d}\lambda\mathrm{d}\rho) = \int a(\lambda, \rho)\nu_1(\mathrm{d}\lambda\mathrm{d}\rho), \quad \forall a \in \mathfrak{P}_{k,1} \tag{42}$$

*and*

$$\int \lambda\rho\nu_0(\mathrm{d}\lambda\mathrm{d}\rho) - \int \lambda\rho\nu_1(\mathrm{d}\lambda\mathrm{d}\rho) \geq \frac{1}{4k^3}. \tag{43}$$

**Proof** : The proof is similar to that of [Cai and Low, 2011, Lemma 1]. Let $C([-1,1]^2)$ be the space of continuous functions on $[-1,1]^2$ equipped with the $L_\infty$ norm and $\mathcal{F}$ be linear space spanned by $a_1(\lambda, \rho) = \lambda\rho$ and $\mathfrak{P}_{k,1}$. Define a linear functional $T$ that maps any $f = ca_1 + g \in \mathcal{F}$ (where $c \in \mathbb{R}$ and $g \in \mathfrak{P}_k$) to $c\delta_{k,1}$, where $\delta_{k,1}$ is defined in the previous lemma. Let $g_1 \in \mathfrak{P}_k$ be the best $L_\infty$-approximation of $a_1$ in $\mathfrak{P}_k$, then $\|a_1 - g_1\|_{L_\infty} = \delta_{k,1}$ and $T(a_1 - g_1) = \delta_{k,1}$, so $\|T\| \geq 1$. On the other hand, for any $f = ca_1 + g \in \mathcal{F}$, we have $\|f\|_\infty \geq |c|\delta_{k,1}$ since otherwise the $L_\infty$ distance between $a_1$ and $-c^{-1}g$ would be smaller than $\delta_{k,1}$, which is a contradiction. Thus $|T(f)| = |c|\delta_{k,1} \leq \|f\|_{L_\infty}$, which implies that $\|T\| \leq 1$.

Therefore, we must have $\|T\| = 1$. By Hahn-Banach theorem, $T$ can be extended to a linear functional on $C([-1,1]^2)$ with unit norm, which we still denote by $T$. The Riesz representer theorem then implies that there exists a signed measure $\mu$ with unit total variation such that $T(f) = \int f\mathrm{d}\mu, \forall f \in C([-1,1]^2)$. In particular, we have

$$\int a(\lambda, \rho)\mathrm{d}\mu = 0, \forall a \in \mathfrak{P}_{k,1} \quad \text{and} \quad \int \lambda\rho\mathrm{d}\mu = \delta_{k,1}.$$

Finally, by the Hahn decomposition theorem, there exists (positive) measures $\nu_0, \nu_1$ such that $\mu = \nu_0 - \nu_1$. Then it is easy to see that such $\nu_0$ and $\nu_1$ satisfy the desired properties, concluding the proof. $\qquad\square$

In the following, we provide the full proof of Theorem 3.2. For any $A > 0$ and $q \in (0,1)$, we define a *two-piece Bernoulli distribution*, denoted by $\mathrm{B}_2(q; A)$, as a distribution with PDF

$$p(x) = \begin{cases} A^{-1}q & x \in [0, A] \\ A^{-1}(1-q) & x \in [-A, 0) \\ 0 & \text{otherwise.} \end{cases}$$

It is easy to see that such a distribution has mean $\left(q - \frac{1}{2}\right)A$.

To begin with, note that we can assume that $|\hat{g}(X)| \leq C_\mathsf{T} - \epsilon_1/2$ and $|\hat{q}(X)| \leq C_\mathsf{q} - \epsilon_2/2$ without loss of generality. Indeed, since $|\hat{g}(X)| \leq C_\mathsf{T}$ and $|\hat{q}(X)| \leq C_\mathsf{q}$, there exists $\tilde{h} = (\tilde{g}(\cdot), \tilde{q}(\cdot))$ satisfying $\|\hat{q} - \tilde{q}\|_{L^\infty} \leq \epsilon_1/2, \|\hat{g} - \tilde{g}\|_{L^\infty} \leq \epsilon_2/2$ and $|\tilde{g}(X)| \leq C_\mathsf{T} - \epsilon_1/2, |\tilde{q}(X)| \leq C_\mathsf{q} - \epsilon_2/2$. Then, $\mathcal{P}_{s,\epsilon/2}(\tilde{h}) \subseteq \mathcal{P}_{s,\epsilon}(\hat{h})$, and any lower bound for $\mathcal{P}_{s,\epsilon/2}(\tilde{h})$ also applies to $\mathcal{P}_{s,\epsilon}(\hat{h})$, implying the desired lower bound up to constants. In the following, we will replace $\hat{h}$ with $\tilde{h}$ work with the uncertainty set $\mathcal{P}_{s,\epsilon/2}(\hat{h})$.

Now, let $A = C_\mathsf{q}, Q = C_\mathsf{q} - \epsilon_2/2$ and $G = C_\mathsf{T} - \epsilon_1/2$. Let $k_n > 0$ be some even integer that will be specified later, and $\nu_0, \nu_1$ be the corresponding distributions in Lemma D.4. For any $\lambda, \rho \in \mathbb{R}$, we define the following data generating process:

$$
\begin{aligned}
&X \sim P_X = \mathrm{Uniform}(\mathcal{X}) \\
&T \sim \mathcal{N}\left(g_\lambda(X), \sigma^2\right) \\
&Y \sim \begin{cases} \mathrm{B}_2(q; 4A), q = \dfrac{1}{4A}\left(2A + \theta_{\lambda,\rho}T + f_{\lambda,\rho}(X)\right) & \text{if } |\theta_{\lambda,\rho}T + f_{\lambda,\rho}(X)| \leq A \\ \mathcal{N}\left(\theta_{\lambda,\rho}T + f_{\lambda,\rho}(X), 1\right) & \text{otherwise} \end{cases}
\end{aligned}
\tag{44}
$$

where

$$
\begin{aligned}
g_\lambda(x) &= \hat{g}(x) + \tilde{\epsilon}_1\sigma\lambda \\
q_\rho(x) &= \hat{q}(x) - \tilde{\epsilon}_2\rho \\
\theta_{\lambda,\rho} &= \tilde{\epsilon}_1\tilde{\epsilon}_2\sigma^{-1}\lambda\rho \\
f_{\lambda,\rho}(x) &= q_\rho(x) - \theta_{\lambda,\rho}g_\lambda(x).
\end{aligned}
$$

By choosing $\tilde{\epsilon}_1 = \sigma^{-1}\epsilon_1/4$ and $\tilde{\epsilon}_2 = \epsilon_2/4$, we can ensure that for any $h_\lambda = (g_\lambda, q_\lambda)$ it holds that $h_\lambda \in \mathcal{P}_{s,\epsilon/2}(\hat{h}), \forall s \in [1, +\infty]^2$.

We use $P_{\lambda,\rho}$ to denote the joint distribution of $(X,T,Y)$ in (44), and $p_{\lambda,\rho}$ be its density. Then we have that

$$
p_{\lambda,\rho}(x,t,y) = \begin{cases}
\dfrac{1}{4A^2}\varphi\left(\dfrac{t-g_\lambda(x)}{\sigma}\right)\cdot(2A+\theta_{\lambda,\rho}t+f_{\lambda,\rho}(x)) & \text{if } |\theta_{\lambda,\rho}t+f_{\lambda,\rho}(x)|\le A \text{ and } y\in[0,2A]\\[2mm]
\dfrac{1}{4A^2}\varphi\left(\dfrac{t-g_\lambda(x)}{\sigma}\right)\cdot(2A-\theta_{\lambda,\rho}t-f_{\lambda,\rho}(x)) & \text{if } |\theta_{\lambda,\rho}t+f_{\lambda,\rho}(x)|\le A \text{ and } y\in[-2A,0)\\[2mm]
0 & \text{if } |\theta_{\lambda,\rho}t+f_{\lambda,\rho}(x)|\le A \text{ and } |y|>A\\[2mm]
\varphi\left(\dfrac{t-g_\lambda(x)}{\sigma}\right)\varphi(y-\theta_{\lambda,\rho}t-f_{\lambda,\rho}(x)) & \text{if } |\theta_{\lambda,\rho}t+f_{\lambda,\rho}(x)|>A.
\end{cases}
$$
$$(45)$$

The following lemma derives an equivalent expression for $\mathbb{E}[Y\mid X=x,T=t]=\theta_{\lambda,\rho}t+f_{\lambda,\rho}(x)$:

**Lemma D.5** (Expression for conditional mean outcome). *For any $x,t$ we have*

$$
\theta_{\lambda,\rho}t+f_{\lambda,\rho}(x)=\hat q(x)+\sigma^{-1}\tilde\epsilon_1\tilde\epsilon_2\lambda\rho(t-\hat g(x))-\tilde\epsilon_2\rho-\sigma^{-2}\tilde\epsilon_1^2\tilde\epsilon_2\lambda^2\rho.
$$

Note that the last event in (45) happens with small probability. Indeed, we can define a *good* event

$$
\mathcal{E}_{\tilde\epsilon_1,\tilde\epsilon_2}=\left\{|T|\le\frac{A-Q-\tilde\epsilon_2}{\tilde\epsilon_1\tilde\epsilon_2\max\{1,\sigma\}^2}-G-1\right\}.
$$
$$(46)$$

An important property of the above definition is that the bound goes to infinity since we assumed that $\epsilon_1=o(1)$ and $\epsilon_2=o(1)$, so that it would happen with high probability.

The following result summarizes the good properties enjoyed by $\mathcal{E}_{\tilde\epsilon_1,\tilde\epsilon_2}$:

**Proposition D.1** (Properties of good events). *We have*

*(1). $\lim_{\tilde\epsilon_1,\tilde\epsilon_2\to0}\inf_{|\lambda|,|\rho|\le1}P_{\lambda,\rho}[\mathcal{E}_{\tilde\epsilon_1,\tilde\epsilon_2}]=1$;*

*(2). If $t\in\mathcal{E}_{\tilde\epsilon_1,\tilde\epsilon_2}$, then for any $\lambda,\rho\in[-1,1]$ and any $x$, we have $|\theta_{\lambda,\rho}t+f_{\lambda,\rho}(x)|\le A$.*

**Proof :** By definition, $P_{\lambda,\rho}[\mathcal{E}_{\tilde\epsilon_1,\tilde\epsilon_2}]=P_{T\sim\mathcal{N}(g_\lambda(x),\sigma^2)}\left[|T|\le\frac{A-Q-\tilde\epsilon_2}{\tilde\epsilon_1\tilde\epsilon_2\max\{1,\sigma\}^2}-G-1\right]$. Since $|g_\lambda(x)|\le G+o(1)$, (1) directly follows. To prove (2), it suffices to note that

$$
|\theta_{\lambda,\rho}T+f_{\lambda,\rho}(X)|=\left|\hat q(X)+\sigma^{-1}\tilde\epsilon_1\tilde\epsilon_2\lambda\rho(T-\hat g(X))-\tilde\epsilon_2\rho-\sigma^{-2}\tilde\epsilon_1^2\tilde\epsilon_2\lambda^2\rho\right|
$$
$$
\le\tilde\epsilon_1\tilde\epsilon_2\max\{1,\sigma\}^2\left(|T|+G+\tilde\epsilon_1\right)+Q+\tilde\epsilon_2
$$

where the first equation is due to Lemma D.5. $\qquad\square$

Let $H_k$ be the $k$-th order Hermite polynomial and $\bar P_0$ and $\bar P_1$ be the mixture of $P_{\lambda,\rho}$ with priors $\nu_0$ and $\nu_1$ respectively, and $\hat p_0$ and $\bar p_1$ be their densities. Our next step would be to bound the $\chi^2$-divergence between $\bar P_0$ and $\bar P_1$. To do this, we need to analyze the densities $\bar p_i(x,t,y),i=0,1$ for two cases $(x,t,y)\in\mathcal{E}_{\tilde\epsilon_1,\tilde\epsilon_2}$ and $(x,t,y)\notin\mathcal{E}_{\tilde\epsilon_1,\tilde\epsilon_2}$ separately. Our next two lemmas handle the first case.

**Lemma D.6** (Taylor expansions of perturbed densities). *Suppose that $t\in\mathcal{E}_{\tilde\epsilon_1,\tilde\epsilon_2}$, then we have*

$$
p_{\lambda,\rho}(x,t,y)=\begin{cases}
\dfrac{1}{4A^2}\varphi(z)\displaystyle\sum_{k=0}^{+\infty}\dfrac{\tilde\epsilon_1^k\lambda^k}{k!\sigma^k}\left(2A+\hat q(x)+(k-1)\tilde\epsilon_2\rho\right)H_k(z) & \text{if } y\in[0,A]\\[3mm]
\dfrac{1}{4A^2}\varphi(z)\displaystyle\sum_{k=0}^{+\infty}\dfrac{\tilde\epsilon_1^k\lambda^k}{k!\sigma^k}\left(2A-\hat q(x)-(k-1)\tilde\epsilon_2\rho\right)H_k(z) & \text{if } y\in[-A,0)
\end{cases}
$$

*where $z=\frac{t-\hat g(x)}{\sigma}$.*

**Proof :** We only prove the statement for $y\in[-A,0)$; the other case $y\in[0,A]$ can be handled similarly. Since

$$
\varphi\left(\frac{t-g_\lambda(x)}{\sigma}\right)=\sum_{k=0}^{+\infty}H_k(z)\,\varphi(z)\,\frac{(-1)^k\tilde\epsilon_1^k\lambda^k}{k!\sigma^k}
$$

and
$$2A - \theta_{\lambda,\rho} t - f_{\lambda,\rho}(x) = 2A - \hat{q}(x) - \tilde{\epsilon}_2 \rho \left[1 - \sigma^{-1}\tilde{\epsilon}_1 \lambda(t - \hat{g}(x)) + \sigma^{-2}\tilde{\epsilon}_1^2 \lambda^2\right]$$
by Lemma D.5, we can deduce that

$$
\begin{aligned}
p_{\lambda,\rho}(x,t,y) &= \frac{1}{4A^2}\varphi(z)\left(\sum_{k=0}^{+\infty} H_k(z)\frac{\tilde{\epsilon}_1^k \lambda^k}{k!\sigma^k}\right)\left[2A - \hat{q}(x) - \tilde{\epsilon}_2\rho\left(1 - \sigma^{-1}\tilde{\epsilon}_1\lambda(t - \hat{g}(x)) + \sigma^{-2}\tilde{\epsilon}_1^2\lambda^2\right)\right] \\
&= \frac{1}{4A^2}\varphi(z)\sum_{k=0}^{+\infty}\frac{\tilde{\epsilon}_1^k \lambda^k}{k!\sigma^k}\left[(2A - \hat{q}(x))H_k(z) - \tilde{\epsilon}_2\rho\left(H_k(z) - kzH_{k-1}(z) + (k-1)kH_{k-2}(z)\right)\right] \\
&= \frac{1}{4A^2}\varphi(z)\sum_{k=0}^{+\infty}\frac{\tilde{\epsilon}_1^k \lambda^k}{k!\sigma^k}\left(2A - \hat{q}(x) - (k-1)\tilde{\epsilon}_2\rho\right)H_k(z),
\end{aligned}
$$

where in the final step holds since $H_k(z) = zH_{k-1}(z) - (k-1)H_{k-2}(z)$. The conclusion follows.
$\square$

**Lemma D.7** (Bounding $\chi^2$-distance under good event). *We have*

$$\int \frac{(\bar{p}_0(x,t,y) - \bar{p}_1(x,t,y))^2}{\bar{p}_0(x,t,y)}\mathbb{1}\left\{t \in \mathcal{E}_{\tilde{\epsilon}_1,\tilde{\epsilon}_2}\right\}\mathrm{d}x\mathrm{d}t\mathrm{d}y \le 200\sigma\left(\frac{e\tilde{\epsilon}_1^2}{k_n - 1}\right)^{k_n - 1}.$$

***Proof :*** Let $(x,t,y)$ satisfies $t \in \mathcal{E}_{\tilde{\epsilon}_1,\tilde{\epsilon}_2}$ and $y \in [-A, 0)$, then Lemma D.6 implies that

$$
\begin{aligned}
\bar{p}_0(x,t,y) - \bar{p}_1(x,t,y) &= \frac{1}{4A^2}\varphi(z)\Bigg[(2A - \hat{q}(x))\sum_{k=0}^{+\infty} H_k(z)\frac{\tilde{\epsilon}_1^k}{k!}\int_{[-1,1]}\lambda^k \mathrm{d}(\nu_0 - \nu_1) \\
&\qquad - \tilde{\epsilon}_2\sum_{k=0}^{\infty} H_k(z)\frac{(k-1)\tilde{\epsilon}_1^k}{k!}\int_{[-1,1]}\lambda^k \rho \mathrm{d}(\nu_0 - \nu_1)\Bigg] \\
&= \frac{1}{4A^2}\varphi(z)\Bigg[(2A - \hat{q}(x))\sum_{k=k_n+1}^{+\infty} H_k(z)\frac{\tilde{\epsilon}_1^k}{k!}\int_{[-1,1]}\lambda^k \mathrm{d}(\nu_0 - \nu_1) \quad (47) \\
&\qquad - \tilde{\epsilon}_2\sum_{k=k_n+1}^{\infty} H_k(z)\frac{(k-1)\tilde{\epsilon}_1^k}{k!}\int_{[-1,1]}\lambda^k \rho \mathrm{d}(\nu_0 - \nu_1)\Bigg] \\
&= \frac{1}{4A^2}\varphi(z)\sum_{k=k_n+1}^{+\infty}\frac{1}{(k-1)!}H_k(z)c_k(x),
\end{aligned}
$$

where $z = \frac{t - \hat{g}(x)}{\sigma}$ and

$$c_k(x) = k^{-1}(2A - \hat{q}(x))\tilde{\epsilon}_1^k\int_{[-1,1]}\lambda^k \mathrm{d}(\nu_0 - \nu_1) - \tilde{\epsilon}_2\tilde{\epsilon}_1^k\int_{[-1,1]}\lambda^k \rho \mathrm{d}(\nu_0 - \nu_1). \qquad (48)$$

On the other hand, since $x \mapsto e^{-x}$ is convex, by Jensen's inequality we have

$$
\begin{aligned}
\bar{p}_0(x,t,y) &= \int p_{\lambda,\rho}(x,t,y)\mathrm{d}\nu_0(\lambda,\rho) \\
&\ge \frac{1}{4A}\int \varphi(z - \tilde{\epsilon}_1\lambda)\mathrm{d}\nu_0(\lambda,\rho) \\
&\ge \frac{1}{4A}\exp\left(-\frac{1}{2}\int(z - \tilde{\epsilon}_1\lambda)^2 \mathrm{d}\nu_0(\lambda,\rho)\right) \qquad (49) \\
&\ge \frac{1}{4A}\exp\left(-\frac{1}{2}z^2 - \frac{1}{2}\tilde{\epsilon}_1^2\right) \ge \frac{1}{8A}\varphi(z)
\end{aligned}
$$

for sufficiently large $n$, since $\tilde{\epsilon}_1, \tilde{\epsilon}_2 = o(1)$. Hence,

$$\frac{(\bar{p}_0(x,t,y) - \bar{p}_1(x,t,y))^2}{\bar{p}_0(x,t,y)} \le \frac{1}{2A^3}\varphi(z)\left(\sum_{k=k_n+1}^{+\infty}\frac{1}{(k-1)!}H_k(z)c_k(x)\right)^2.$$

Fixing $x, y$ and integrating both sides of (47) with respect to $t$, we can deduce that

$$
\int \frac{(\bar{p}_0(x,t,y) - \bar{p}_1(x,t,y))^2}{\bar{p}_0(x,t,y)} \mathbb{1}\{t \in \mathcal{E}_{\tilde{\epsilon}_1, \tilde{\epsilon}_2}\} \mathrm{d}t
$$

$$
\leq \frac{1}{2A^3} \int \varphi(z) \left( \sum_{k=k_n+1}^{+\infty} \frac{1}{(k-1)!} H_k(z) c_k(x) \right)^2 \mathbb{1}\{\sigma z + \hat{g}(x) \in \mathcal{E}_{\tilde{\epsilon}_1, \tilde{\epsilon}_2}\} \cdot \sigma \mathrm{d}z
$$

$$
\leq \frac{\sigma}{2A^3} \int \varphi(z) \left( \sum_{k=k_n+1}^{+\infty} \frac{1}{(k-1)!} H_k(z) c_k(x) \right)^2 \mathrm{d}z
$$

$$
= \frac{\sigma}{2A^3} \left[ \sum_{k=k_n+1}^{+\infty} \frac{c_k^2(x)}{((k-1)!)^2} \int \varphi(z) H_k^2(z) \mathrm{d}z + 2 \sum_{j > i \geq k_n+1} \frac{c_i(x) c_j(x)}{(i-1)!(j-1)!} \int \varphi(z) H_i(z) H_j(z) \mathrm{d}z \right]
$$

$$
= \frac{\sigma}{2A^3} \sum_{k=k_n+1}^{+\infty} \frac{k c_k^2(x)}{(k-1)!} \leq \frac{\sigma}{A^3} \sum_{k=k_n+1}^{+\infty} \frac{c_k^2(x)}{(k-2)!},
$$

(50)

where the last step follows from the orthogonality of Hermite polynomials:

$$
\int \varphi(z) H_k^2(z) = k! \quad \text{and} \quad \int \varphi(z) H_i(z) H_j(z) \mathrm{d}z = 0, \forall i \neq j.
$$

Moreover, (48) implies that

$$
|c_k(x)| \leq 6Ak^{-1}\tilde{\epsilon}_1^k + 2\tilde{\epsilon}_1^k \tilde{\epsilon}_2 \leq 8A\tilde{\epsilon}_1^k.
$$

Plugging into (50), we can deduce that

$$
\int \frac{(\bar{p}_0(x,t,y) - \bar{p}_1(x,t,y))^2}{\bar{p}_0(x,t,y)} \mathbb{1}\{t \in \mathcal{E}_{\tilde{\epsilon}_1, \tilde{\epsilon}_2}\} \mathrm{d}t \leq 64\sigma A^{-1} \sum_{k=k_n-1}^{+\infty} \frac{1}{k!} \tilde{\epsilon}_1^{2k} \leq 100\sigma A^{-1} \left( \frac{e\tilde{\epsilon}_1^2}{k_n-1} \right)^{k_n-1}.
$$

For $t \in [0, A]$, the above inequality can be established in a similar fashion. As a result, we have

$$
\int \frac{(\bar{p}_0(x,t,y) - \bar{p}_1(x,t,y))^2}{\bar{p}_0(x,t,y)} \mathrm{d}x \mathrm{d}t \mathrm{d}y \leq 100\sigma A^{-1} \left( \frac{e\tilde{\epsilon}_1^2}{k_n-1} \right)^{k_n-1} \int \mathrm{d}x \mathrm{d}y \leq 200\sigma \left( \frac{e\tilde{\epsilon}_1^2}{k_n-1} \right)^{k_n-1},
$$

as desired. $\qquad \square$

Our next lemma, on the other hand, develops bounds for densities outside $\mathcal{E}_{\tilde{\epsilon}_1, \tilde{\epsilon}_2}$. It essentially shows that this part makes a negligible contribution to the overall $\chi^2$-divergence.

**Lemma D.8** (Bounding $\chi^2$-distance under bad event). *For any $t \notin \mathcal{E}_{\tilde{\epsilon}_1, \tilde{\epsilon}_2}$, we have*

$$
\frac{(\bar{p}_0(x,t,y) - \bar{p}_1(x,t,y))^2}{\bar{p}_0(x,t,y)} \leq 16 \exp\left( -\frac{1}{12\sigma^2} (t - \hat{g}(x))^2 \right).
$$

**Proof :** For any $t \notin \mathcal{E}_{\tilde{\epsilon}_1, \tilde{\epsilon}_2}$ and any $x \in \mathcal{X}$, define

$$
\Lambda_{x,t,y}^{(0)} := \left\{ (\lambda, \rho) \in [-1,1]^2 \mid |\theta_{\lambda,\rho} t + f_{\lambda,\rho}(x)| \leq A \right\}
$$

and

$$
\Lambda_{x,t,y}^{(1)} := [-1,1]^2 \setminus \Lambda_{x,t,y}^{(0)}.
$$

For any $(\lambda, \rho), (\hat{\lambda}, \hat{\rho}) \in [-1, 1]^2$ such that $(\lambda, \rho), (\hat{\lambda}_i, \hat{\rho}'_i) \in \Lambda^{(1)}_{x,t,y}$, we have

$$- \log p_{\lambda,\rho}(x,t,y) \tag{51a}$$

$$= \frac{1}{2\sigma^2}\big(t - g_\lambda(x)\big)^2 + \frac{1}{2}\Big[y - \hat{q}(x) - \sigma^{-1}\tilde{\epsilon}_1\tilde{\epsilon}_2\lambda_i\rho_i(t - \hat{g}(x)) + \tilde{\epsilon}_2\rho + \sigma^{-2}\tilde{\epsilon}_1^2\tilde{\epsilon}_2\lambda_i^2\rho_i\Big]^2 \tag{51b}$$

$$\geq \frac{1}{2\sigma^2}\big(t - g_\lambda(x)\big)^2 + \frac{1}{4}\Big[y - \hat{q}(x) - \sigma^{-1}\tilde{\epsilon}_1\tilde{\epsilon}_2\hat{\lambda}_i\hat{\rho}_i(t - \hat{g}(x)) + \tilde{\epsilon}_2\hat{\rho}_i + \sigma^{-2}\tilde{\epsilon}_1^2\tilde{\epsilon}_2\hat{\lambda}_i^2\hat{\rho}_i\Big]^2$$
$$- \frac{1}{2}(\tilde{\epsilon}_1 + \tilde{\epsilon}_2)^2 - \frac{1}{2\sigma^2}\tilde{\epsilon}_1^2\tilde{\epsilon}_2^2\big(t - \hat{g}(x)\big)^2 \tag{51c}$$

$$\geq \frac{1}{3\sigma^2}\big(t - \hat{g}(x)\big)^2 + \frac{1}{4}\Big[y - \hat{q}(x) - \sigma^{-1}\tilde{\epsilon}_1\tilde{\epsilon}_2\hat{\lambda}_i\hat{\rho}_i(t - \hat{g}(x)) + \tilde{\epsilon}_2\hat{\rho}_i + \sigma^{-2}\tilde{\epsilon}_1^2\tilde{\epsilon}_2\hat{\lambda}_i^2\hat{\rho}_i\Big]^2$$
$$- \frac{1}{2}(\tilde{\epsilon}_1 + \tilde{\epsilon}_2)^2 - \frac{1}{2\sigma^2}\tilde{\epsilon}_1^2\tilde{\epsilon}_2^2\big(t - \hat{g}(x)\big)^2 - \frac{4\tilde{\epsilon}_1^2}{\sigma^2} \tag{51d}$$

$$\geq -\frac{1}{2}\log p_{\hat{\lambda},\hat{\rho}}(x,t,y) + \frac{1}{24\sigma^2}\big(t - \hat{g}(x)\big)^2, \tag{51e}$$

where (51b) follows from (45), (51c) and (51d) follow from the inequality $u^2 \geq \rho(u + v)^2 - \frac{\rho}{1-\rho}v^2, \forall \rho \in (0,1)$, and (51e) holds because $\tilde{\epsilon}_1, \tilde{\epsilon}_2 = o(1)$ and $|t - \hat{g}(x)| = \Omega(1)$ by (46). With a similar reasoning, we can also deduce from (51b) that

$$- \log p_{\lambda,\rho}(x,t,y) \geq \frac{1}{4\sigma^2}\big(t - \hat{g}(x)\big)^2, \quad \forall(\lambda, \rho) \text{ s.t. } (\lambda, \rho) \in \Lambda^{(1)}_{x,t,y}. \tag{52}$$

Define

$$I_{i,j} = \int_{(\lambda,\rho)\in\Lambda^{(i)}_{x,t,y}} p_\lambda(x,t,y)\mathrm{d}\nu_j(\lambda,\rho), \quad i,j \in \{0,1\},$$

where we drop the dependency on $(x,t,y)$ for convenience. Then from (51) we can deduce that

$$I^2_{1,1} \leq \max_{(\lambda,\rho)\in\Lambda^{(1)}_{x,t,y}} p_{\lambda,\rho}(x,t,y)^2 \leq \exp\left(-\frac{1}{12\sigma^2}\big(t - \hat{g}(x)\big)^2\right) \min_{(\lambda,\rho)\in\Lambda^{(1)}_{x,t,y}} p_{\lambda,\rho}(x,t,y)$$

$$\leq \exp\left(-\frac{1}{12\sigma^2}\big(t - \hat{g}(x)\big)^2\right) I_{1,0}.$$

Thus, combining the above inequality (52) we have

$$\frac{(I_{1,0} - I_{1,1})^2}{I_{1,0}} \leq 2\left(I_{1,0} + I^{-1}_{1,0}I^2_{1,1}\right) \leq 4\exp\left(-\frac{1}{12\sigma^2}\big(t - \hat{g}(x)\big)^2\right). \tag{53}$$

On the other hand, from (45) it is easy to see that for any $(\lambda, \rho) \in \Lambda^{(0)}_{x,t,y}$ we have

$$\frac{1}{4A}\varphi(z) \leq p_\lambda(x,t,y) \leq \frac{3}{4A}\varphi(z)$$

where $z = \frac{t - g_\lambda(x)}{\sigma}$, so $\frac{1}{3}I_{0,0} \leq I_{0,1} \leq 3I_{0,0}$ and

$$\frac{(I_{0,0} - I_{0,1})^2}{I_{0,0}} \leq 4I_{0,0} \leq 4A^{-1}\exp\left(-\frac{1}{4\sigma^2}\big(t - \hat{g}(x)\big)^2\right), \tag{54}$$

where the last inequality is due to the bound $I_{0,0} \leq A^{-1}\exp\left(-\frac{1}{4\sigma^2}\big(t - \hat{g}(x)\big)^2\right)$, which directly follows from its definition (45) and the argument used in (51e).

Combining (53) and (54), we can deduce that

$$\frac{(\bar{p}_0(x,t,y) - \bar{p}_1(x,t,y))^2}{\bar{p}_0(x,t,y)} = \frac{(I_{0,0} - I_{0,1} + I_{1,0} - I_{1,1})^2}{I_{0,0} + I_{1,0}}$$

$$\leq 2\left(\frac{(I_{0,0} - I_{0,1})^2}{I_{0,0}} + \frac{(I_{1,0} - I_{1,1})^2}{I_{1,0}}\right)$$

$$\leq 16\exp\left(-\frac{1}{12\sigma^2}\big(t - \hat{g}(x)\big)^2\right),$$

as desired. □

Combining the results of Lemma D.7 and Lemma D.8, we obtain the following:

**Corollary D.1** (Bounding the whole $\chi^2$-distance between the null and alternative distributions). *Let $\bar{P}_0$ and $\bar{P}_1$ be the mixture distributions as defined before. Then we have that*

$$\chi^2(\bar{P}_0||\bar{P}_1) \le 200\sigma \left(\frac{e\tilde{\epsilon}_1^2}{k_n - 1}\right)^{k_n - 1} + 96A^{-1}\sigma^2\tilde{\epsilon}_1\tilde{\epsilon}_2 \exp\left(-\frac{A^2}{24\tilde{\epsilon}_1^2\tilde{\epsilon}_2^2}\right). \tag{55}$$

*In particular, if $\tilde{\epsilon}_1\tilde{\epsilon}_2 = o\left(\log^{-1/2} n\right)$ and $k_n \ge -\frac{\log n}{\log \tilde{\epsilon}_1}$, then $\chi^2(\bar{P}_0||\bar{P}_1) = o\left((nk_n^3)^{-1}\right)$ and $\chi^2\left(\bar{P}_0^{\otimes n}||\bar{P}_1^{\otimes n}\right) = o\left(k_n^{-3}\right)$.*

*Proof* : By definition we have

$$\chi^2(\bar{P}_0||\bar{P}_1)$$
$$= \int_{\mathcal{E}_{\tilde{\epsilon}_1,\tilde{\epsilon}_2}} \frac{(\bar{p}_0(x,t,y) - \bar{p}_1(x,t,y))^2}{\bar{p}_0(x,t,y)^2} \mathrm{d}\bar{P}_0 + \int_{\mathcal{E}_{\tilde{\epsilon}_1,\tilde{\epsilon}_2}^c} \frac{(\bar{p}_0(x,t,y) - \bar{p}_1(x,t,y))^2}{\bar{p}_0(x,t,y)^2} \mathrm{d}\bar{P}_0$$

$$\le 200\sigma \left(\frac{e\tilde{\epsilon}_1^2}{k_n - 1}\right)^{k_n - 1} + 16 \int_{\mathcal{E}_{\tilde{\epsilon}_1,\tilde{\epsilon}_2}^c} \exp\left(-\frac{1}{12\sigma^2}\left(t - \hat{g}(x)\right)^2\right) \mathrm{d}x\mathrm{d}t\mathrm{d}y$$

$$\le 200\sigma \left(\frac{e\tilde{\epsilon}_1^2}{k_n - 1}\right)^{k_n - 1} + 16 \int \exp\left(-\frac{1}{12\sigma^2}\left(t - \hat{g}(x)\right)^2\right) \mathbb{1}\left\{|t - \hat{g}(x)| \ge \frac{A}{2\tilde{\epsilon}_1\tilde{\epsilon}_2}\right\} \mathrm{d}x\mathrm{d}t$$

$$= 200\sigma \left(\frac{e\tilde{\epsilon}_1^2}{k_n - 1}\right)^{k_n - 1} + 16\sqrt{6}\sigma \int \exp\left(-\frac{1}{2}\left(s - \hat{g}(x)\right)^2\right) \mathbb{1}\left\{|s - \hat{g}(x)| \ge \frac{A}{2\sqrt{6}\sigma\tilde{\epsilon}_1\tilde{\epsilon}_2}\right\} \mathrm{d}x\mathrm{d}t$$

$$\le 200\sigma \left(\frac{e\tilde{\epsilon}_1^2}{k_n - 1}\right)^{k_n - 1} + 96A^{-1}\sigma^2\tilde{\epsilon}_1\tilde{\epsilon}_2 \exp\left(-\frac{A^2}{24\tilde{\epsilon}_1^2\tilde{\epsilon}_2^2}\right).$$

If $\tilde{\epsilon}_1\tilde{\epsilon}_2 = o\left(\log^{1/2} n\right)$ and $k_n$ is the smallest integer satisfying $k_n \ge -\frac{\log n}{\log \tilde{\epsilon}_1}$, it is easy to see that both terms in (55) are $o\left((nk_n^3)^{-1}\right)$, so $\chi^2(\bar{P}_0||\bar{P}_1) = o\left((nk_n^3)^{-1}\right)$. It follows that

$$\chi^2\left(\bar{P}_0^{\otimes n}||\bar{P}_1^{\otimes n}\right) = \int \frac{\left(\prod_{i=1}^n \bar{p}_0(x_i,t_i,y_i) - \prod_{i=1}^n \bar{p}_1(x_i,t_i,y_i)\right)^2}{\prod_{i=1}^n \bar{p}_0(x_i,t_i,y_i)} \mathrm{d}x_1 \cdots \mathrm{d}x_n \mathrm{d}t_1 \cdots \mathrm{d}t_n \mathrm{d}y_1 \cdots \mathrm{d}y_n$$

$$= \int \frac{\left(\prod_{i=1}^n \bar{p}_1(x_i,t_i,y_i)\right)^2}{\prod_{i=1}^n \bar{p}_0(x_i,t_i,y_i)} \mathrm{d}x_1 \cdots \mathrm{d}x_n \mathrm{d}t_1 \cdots \mathrm{d}t_n \mathrm{d}y_1 \cdots \mathrm{d}y_n - 1$$

$$= \prod_{i=1}^n \int \frac{\bar{p}_1(x_i,t_i,y_i)^2}{\bar{p}_0(x_i,t_i,y_i)} \mathrm{d}x_i\mathrm{d}t_i\mathrm{d}y_i - 1$$

$$\le \left(1 + o\left((nk_n^3)^{-1}\right)\right)^n - 1 = o\left(k_n^{-3}\right),$$

which concludes the proof. $\qquad\square$

Finally, we can apply Lemma D.1 to deduce our lower bound. We define the following functional $T$: for any observation distribution $P_{\lambda,\rho}^{\otimes n}$ of $\{(x_i, t_i, y_i)\}_{i=1}^n$ generated from a model in (44), $T(P)$ equals the corresponding parameter value $\theta_{\lambda,\rho}$. Let $\nu_0, \nu_1$ be the distributions that satisfy the property in Lemma D.4 corresponding to the $k_n$ in Corollary D.1; we can also view $\nu_0$ and $\nu_1$ as distributions on the $P_{\lambda,\rho}^{\otimes n}$'s. Note that $\theta_{\lambda,\rho} = \sigma^{-1}\tilde{\epsilon}_1\tilde{\epsilon}_2\lambda\rho$ by (44), we know from Lemma D.4 that the mean difference between $\nu_0$ and $\nu_1$ is

$$m_1 - m_0 = \int T\left(P_{\lambda,\rho}^{\otimes n}\right) \mathrm{d}(\nu_0 - \nu_1)(\lambda,\rho) = \sigma^{-1}\tilde{\epsilon}_1\tilde{\epsilon}_2 \int \lambda\rho\mathrm{d}(\nu_0 - \nu_1)(\lambda,\rho) \ge \frac{1}{4\sigma k_n^3}\tilde{\epsilon}_1\tilde{\epsilon}_2.$$

On the other hand, we clearly have $v_0 \le 2\sigma^{-1}\tilde{\epsilon}_1\tilde{\epsilon}_2$, and Corollary D.1 implies that $I = \chi^2\left(\bar{P}_0^{\otimes n}||\bar{P}_1^{\otimes n}\right) = o\left(k_n^{-3}\right)$. So for sufficiently large $n$, we have $m_1 - m_0 - v_0 I \ge \frac{1}{8\sigma k_n^3}\tilde{\epsilon}_1\tilde{\epsilon}_2$. By Lemma D.1, the minimax mean-square error for any estimator $\hat{T}$ is at least

$$\Omega\left(k_n^{-3}\tilde{\epsilon}_1\tilde{\epsilon}_2\right) = \Omega\left(-\left(\frac{\log n}{\log \tilde{\epsilon}_1}\right)^{-3}\tilde{\epsilon}_1\tilde{\epsilon}_2\right) = \Omega\left(-\left(\frac{\log n}{\log \epsilon_{n,g}}\right)^{-3}\sigma^{-2}\epsilon_1\epsilon_2\right).$$

In other words, we have

$$\mathfrak{M}_{n,1-\gamma}\left(\mathcal{P}_{s,\epsilon}(\hat{h})\right) \geq -c_\gamma \left(\frac{\log n}{\log \epsilon_1}\right)^{-3} \sigma^{-2}\epsilon_1\epsilon_2. \tag{56}$$

It remains to prove the $n^{-1/2}$ component of the lower bound. Our proof relies on the following lemma that derives the $\chi^2$-divergence between two Gaussian mixtures.

**Lemma D.9** ($\chi^2$-distance for a specific Gaussian model). *Let $P_i, i = 0,1$ be the distribution of $(X, T, Y)$ generated from*

$$X \sim P_X, \quad T \mid X \sim \mathcal{N}(g_i(X), \sigma^2), \quad Y|X,T \sim \mathcal{N}(q_i(X) + (T - g_i(X))\theta_i, 1), \tag{57}$$

*such that $\sqrt{2}\sigma|\theta_1 - \theta_0| < 1$. Then*

$$\chi^2(P_1, P_0)$$
$$= \left[1 - 2\sigma^2(\theta_1 - \theta_0)^2\right]^{-1/2} \int \exp\left(\frac{\left[q_1(x) - q_0(x) + (\theta_1 - 2\theta_0)(g_1(x) - g_0(x))\right]^2}{1 - 2\sigma^2(\theta_1 - \theta_0)^2}\right) dx - 1.$$

**Proof** : It is easy to see that the density of $P_i$ can be written as

$$p_i(x,t,y) = \frac{1}{2\pi\sigma}p_X(x)\exp\left(-\frac{1}{2\sigma^2}(t - g_i(x))^2 - \frac{1}{2}\left(y - q_i(x) - (t - g_i(x))\theta_i\right)^2\right),$$

thus

$$-2\log p_1(x,t,y) + \log p_0(x,t,y)$$
$$= -\log\left(\frac{p_X(x)}{2\pi\sigma}\right) + \frac{1}{2}y^2 - \left[2\big(q_1(x) + (t - g_1(x))\theta_1\big) - \big(q_0(x) + (t - g_0(x))\theta_0\big)\right]y$$
$$+ \frac{1}{2\sigma^2}\left[2(t - g_1(x))^2 - (t - g_0(x))^2\right]$$
$$+ \frac{1}{2}\left[2(q_1(x) + (t - g_1(x))\theta_1)^2 - (q_0(x) + (t - g_0(x))\theta_0)^2\right]$$
$$= -\log\left(\frac{p_X(x)}{2\pi\sigma}\right) + \frac{1}{2}\left[y - \big(2(q_1(x) + (t - g_1(x))\theta_1) + (q_0(x) + (t - g_0(x))\theta_0)\big)\right]^2$$
$$+ \frac{1}{2\sigma^2}\left[2(t - g_1(x))^2 - (t - g_0(x))^2\right] - \left[\big(q_1(x) + (t - g_1(x))\theta_1\big) - \big(q_0(x) + (t - g_0(x))\theta_0\big)\right]^2$$
$$= -\log\left(\frac{p_X(x)}{2\pi\sigma}\right) + \underbrace{\frac{1}{2}\left[y - \big(2(q_1(x) + (t - g_1(x))\theta_1) + (q_0(x) + (t - g_0(x))\theta_0)\big)\right]^2}_{:=a(y,t,x)}$$
$$+ \underbrace{\left[\frac{1}{2\sigma^2} - (\theta_1 - \theta_0)^2\right](t - c(x))^2}_{:=b(t,x)} - \underbrace{\frac{\left[q_1(x) - q_0(x) + (\theta_1 - 2\theta_0)(g_1(x) - g_0(x))\right]^2}{1 - 2\sigma^2(\theta_1 - \theta_0)^2}}_{:=d(x)}$$

where $c(x)$ is some irrelevant function of $x$. By taking integration, we can deduce that

$$\chi^2(P_1, P_0) = \int \frac{p_1^2(x,t,y)}{p_0(x,t,y)}dxdtdy - 1$$
$$= \frac{1}{2\pi\sigma}\int p_X(x)\exp(-a(y,t,x) - b(t,x) + d(x))dxdtdy - 1$$
$$= \frac{1}{2\pi\sigma}\int p_X(x)\exp(d(x))dx \int \exp(-b(t,x))dt \int \exp(-a(y,t,x))dy$$
$$= \left[1 - 2\sigma^2(\theta_1 - \theta_0)^2\right]^{-1/2}\int \exp\left(d(x)\right)dx - 1$$

as desired. $\qquad\square$

The next corollary highlights the special case of Lemma D.9 that we will use in our proof:

**Corollary D.2** (Bounding the $\chi^2$-distance). *In the setting of Lemma D.9, if $g_0 = g_1, q_0 = q_1$ and $\sigma|\theta_1 - \theta_0| \leq 0.1$, then $\chi^2(P_1, P_0) \leq 2\sigma^2(\theta_1 - \theta_0)^2$.*

***Proof*** : By Lemma D.9, we have $\chi^2(P_1, P_0) \leq \left[1 - 2\sigma^2(\theta_1 - \theta_0)^2\right]^{-1/2} - 1$, and $\sigma|\theta_1 - \theta_0| \leq 0.1$ implies that

$$\left[1 - 2\sigma^2(\theta_1 - \theta_0)^2\right]^{-1/2} \leq 1 + 2\sigma^2(\theta_1 - \theta_0)^2,$$

concluding the proof. $\qquad\square$

We now define

$$\hat{\theta} = 0, \quad \tilde{\theta} = \xi^{1/2}\sigma^{-1}n^{-1/2}/2$$

and let $\hat{P}$ and $\tilde{P}$ be distributions of $(X, T, Y)$ generated from (57) with $(g, q, \theta) = (\hat{g}, \hat{q}, \hat{\theta})$ and $(\hat{g}, \hat{q}, \tilde{\theta})$ respectively. Then Corollary D.2 implies that $\chi^2(\tilde{P}, \hat{P}) \leq \xi n^{-1}/2$ and thus

$$H(\tilde{P}^{\otimes n}, \hat{P}^{\otimes n}) \leq nH(\tilde{P}, \hat{P}) \leq n\chi^2(\tilde{P}, \hat{P}) \leq \xi/2.$$

Therefore, Lemma A.1 implies that for any estimator $\hat{T}$, it holds that

$$\sup_{P \in \mathcal{P}} P\left[\left|\hat{T} - T(P)\right| \geq \xi^{1/2}\sigma^{-1}n^{-1/2}/4\right] \geq \frac{1 - \sqrt{\xi(1 - \xi/4)}}{2} = \gamma.$$

Equivalently, we have

$$\mathfrak{M}_{1-\gamma}\left(\mathcal{P}_{2,\epsilon}(\hat{h})\right) \geq \xi^{1/2}\sigma^{-1}n^{-1/2}/4. \tag{58}$$

Combining (56) and (58), we obtain the desired result.

# E   General upper bounds under Neyman Orthogonality

## E.1   Estimation error of general moment estimators

In this section, we establish upper bounds for general orthogonal estimators beyond DML.

To state our first result, we require several assumptions, as stated below. These assumptions largely follows [Mackey et al., 2018]. We first define the Neyman orthogonality property of moment functions.

**Definition E.1** (Orthogonality of moment function). *A moment function $m(Z, \theta_0, h_0(X)) : \mathbb{R}^K \times \mathbb{R} \times \mathbb{R}^\ell \mapsto \mathbb{R}^d$ is said to be $(S_0, S_1)$-orthogonal for some sets $S_1 \subseteq S_0 \subseteq \mathbb{Z}_{\geq 0}^\ell$, if for any $\alpha \in S_0$, we have $\mathbb{E}_P\left[D^\alpha m(Z, \theta_0, h_0(X)) \mid X\right] = 0$ a.s., and for any $\alpha' \in S_1$, we have $D^\alpha m(Z, \theta_0, \gamma) = 0$ a.s., where $D^\alpha m(Z, \theta_0, \gamma) := \nabla_{\gamma_1}^{\alpha_1} \nabla_{\gamma_2}^{\alpha_2} \cdots \nabla_{\gamma_\ell}^{\alpha_\ell} m(Z, \theta_0, \gamma), \forall \gamma \in \mathbb{R}^\ell$.*

This property is the key to constructing efficient structure-agnostic estimators.

**Assumption E.1** (Main assumptions). *Let $S_1 \subseteq S_0$ be non-empty sets and $k \in \mathbb{Z}_+$, then the following conditions hold:*

*(1). The moment $m$ is $(S_0, S_1)$-orthogonal.*

*(2). $\mathbb{E}_P[m(Z, \theta_0, h_0(X))] \neq 0$ for all $\theta \neq \theta_0$.*

*(3). $\left|\mathbb{E}_P[\nabla_\theta m(Z, \theta_0, h_0(X))]\right| \geq \delta_{\mathsf{id}}$ and $\mathrm{Var}_P(m(Z, \theta_0, h_0(X))) \leq V_{\mathsf{m}}$.*

*(4). $D^\alpha m$ exists and is continuous for all $\|\alpha\|_1 \leq k + 1$.*

The specific choices of $S_0, S_1$ and $k$ will be explicitly stated in all our results. In Assumption E.1, (1) requires orthogonality of the moment function, (2) guarantees that $\theta_0$ is the unique solution to the moment equation, (3) guarantees identifiability of $\theta_0$, and lastly, (4) requires sufficient regularity of the moment function. Finally, we assume the following regularity conditions:

**Assumption E.2** (Additional regularity assumptions). *Define $\mathcal{B}_{h_0, r} = \left\{h \in \mathcal{H} : \max_{\|\alpha\|_1 \leq k+1} \mathbb{E}\left[\prod_{i=1}^\ell |h_i(X) - h_{0,i}(X)|^{2\alpha_i}\right] \leq r\right\}$. Then there exists $r > 0$ such that*

*(1)*. $\mathbb{E}\left[\sup_{|\theta-\theta_0|\leq r}\|\nabla_\theta m_\theta(Z,\theta,h_0(X))\|\right] < +\infty$;

*(2)*. *For any compact set* $A \subseteq \Theta$, *it holds that* $\sup_{\theta\in A, h\in\mathcal{B}_{h_0,r}} \mathbb{E}\left[\|\nabla_\gamma m(Z,\theta,h(X))\|^2\right] <$
$+\infty$ *and* $\mathbb{E}\left[\sup_{\theta\in A, h\in\mathcal{B}_{h_0,r}} |m(Z,\theta,h(X))|\right] < +\infty$;

*(3)*. $\sup_{h\in\mathcal{B}_{h_0,r}} \mathbb{E}\left[\sup_{|\theta-\theta_0|\leq r}\|\nabla_{\theta,\gamma} m(Z,\theta,h(X))\|^2\right] < +\infty$;

*(4)*. $\lambda_\star(\theta_0,h_0) := \max_{\|\alpha\|_1\leq k+1} \sup_{h\in\mathcal{B}_{h_0,r}} \|D^\alpha m(Z,\theta,h(X))\|_{2p} < +\infty$.

Let $\mathcal{I}_{k,\ell}$ the the set of all indices $\alpha \in \mathbb{Z}_{\geq 0}^\ell$ such that $\|\alpha\|_1 \leq k$ and $\mathcal{I}_{k,\ell,0} = \mathcal{I}_{k,\ell}\backslash\mathcal{I}_{k-1,\ell}$. The following theorem shows that orthogonal moments as in Definition E.1 directly yields efficient structure-agnostic estimators of $\theta_0$.

**Theorem E.1** (Structure-agnostic guarantee for general orthogonal estimators). *Let* $S_1 \subseteq S_0 \subseteq \mathbb{Z}_{\geq 0}^\ell, k \in \mathbb{Z}_+$, *and* $p, q \in [1, +\infty]$ *be such that* $p^{-1} + q^{-1} = 1$. *Let* $\mathcal{P}$ *be a set of distributions of* $(X, T, Y)$ *generated from* (1), $\epsilon_i > 0, i = 1, 2, \cdots, \ell$ *and* $s \geq q\max_{\alpha\in S}\sum_{i=1}^l \alpha_i$. *Further let* $\Phi$ *be an arbitrary mapping that maps* $P \in \mathcal{P}$ *to some function* $h_0 : \mathbb{R}^\ell \mapsto \mathbb{R}$ *in some vector space* $\mathcal{F}$. *Consider the estimate* $\hat{\theta}_{\text{OML}}$ *obtained by solving the moment equations*

$$\frac{1}{n}\sum_{i=1}^n m\left(Z_i, \theta, \hat{h}\right) = 0. \tag{59}$$

*Suppose that the moment function* $m : \mathbb{R}^K \times \mathbb{R} \times \mathbb{R}^\ell \mapsto \mathbb{R}^d$ *satisfies Assumption E.1 with* $S_0, S_1, k$ *specified above and additional regularity conditions (stated in Assumption E.2) for all* $P \in \mathcal{P}$. *then for any* $\gamma \in (0, 1)$, *there exists a constant* $C_\gamma > 0$ *such that*

$$\mathfrak{R}_{n,1-\gamma}(\hat{\theta}_{\text{OML}}; \mathcal{P}_{s,\epsilon}(\hat{h})) \leq C_\gamma \delta_{\text{id}}^{-1} \times \left(\sqrt{\frac{V_{\mathsf{m}}}{n}} + \lambda_\star \sum_{\alpha\in(\mathcal{I}_{k,\ell}\backslash S_0)\cup(\mathcal{I}_{k+1,\ell,0}\backslash S_1)} \frac{1}{\|\alpha\|_1!}\prod_{i=1}^\ell \epsilon_i^{\alpha_i}\right) \tag{60}$$

*with probability* $\geq 1 - \gamma$, *where*

$$\lambda_\star = \sup_{P\in\mathcal{P}} \max_{\alpha\in(\mathcal{I}_{k,\ell}\backslash S_0)\cup(\mathcal{I}_{k+1,\ell,0}\backslash S_1)} \|D^\alpha m(Z,\theta_0,h_0(X))\|_{L^p(P)}.$$

Additionally, when the nuisance error rates are sufficiently fast, we have the following asymptotic normality guarantee for $\hat{\theta}$:

**Corollary E.1** (Asymptotic normality). *Suppose that* $\prod_{i=1}^\ell \epsilon_i^{\alpha_i} = o(n^{-1/2})$ *for all* $\alpha \in (\mathcal{I}_{k,\ell} \backslash S_0) \cup (\mathcal{I}_{k+1,\ell,0} \backslash S_1)$, *then* $\sqrt{n}(\hat{\theta} - \theta_0) \xrightarrow{d} \mathcal{N}(0, \delta_{\text{id}}^{-2}V_{\mathsf{m}})$.

The proof can be found in Appendix E.3.

## E.2 Proofs of Theorem E.1 and Corollary E.1

The proof is based on the standard arguments for bouding estimation errors of orthogonal estimators; see *e.g.* [Mackey et al., 2018, Section A]. The only major difference is that our bound is structure-agnostic while their goal is to establish $\mathcal{O}(n^{-1/2})$ convergence rate under assumptions on nuisance errors. For conciseness, we will not repeat the arguments that have already been covered in their paper.

To begin with, their eq.(10) shows that

$$\sqrt{n}(\hat{\theta} - \theta_0)\mathbb{1}\{\det J(\hat{h}) \neq 0\} = J(\hat{h})^{-1}\mathbb{1}\{\det J(\hat{h}) \neq 0\}\frac{1}{\sqrt{n}}\sum_{i=1}^n m(Z_i, \theta_0, \hat{h}(X_i)),$$

where $J(\hat{h}) = \frac{1}{n}\sum_{i=1}^n m'_\theta(Z_i, \tilde{\theta}, \hat{h}(X_i))$ for some $\theta = \lambda\theta_0 + (1-\lambda)\hat{\theta}, \lambda \in [0, 1]$ and $J = \mathbb{E}[m(Z, \theta_0, h_0(X))]$. They also show that $J(\hat{h})^{-1}\mathbb{I}[\det J(\hat{h}) \neq 0] \xrightarrow{p} J^{-1}$. Hence

$$\sqrt{n}(\hat{\theta} - \theta_0) = J^{-1}\underbrace{\frac{1}{\sqrt{n}}\sum_{i=1}^n m(Z_i, \theta_0, \hat{h}(X_i))}_{=:B} + o_P(1).$$

We then consider the decomposition of $B$ following [Mackey et al., 2018, eq.(11)]:

$$
\begin{aligned}
B &= \underbrace{\frac{1}{\sqrt{n}} \sum_{i=1}^{n} m\left(Z_i, \theta_0, h_0\left(X_i\right)\right)}_{=:B_1} \\
&+ \underbrace{\frac{1}{\sqrt{n}} \sum_{i=1}^{n} \sum_{\alpha \in \mathcal{I}_{k,\ell} \cap S_0} \frac{1}{\|\alpha\|_1!} D^\alpha m\left(Z_i, \theta_0, h_0\left(X_i\right)\right)\left(\hat{h}\left(X_i\right) - h_0\left(X_i\right)\right)^\alpha}_{=:B_2} \\
&+ \underbrace{\frac{1}{\sqrt{n}} \sum_{i=1}^{n} \sum_{\alpha \in \mathcal{I}_{k,\ell} \backslash S_0} \frac{1}{\|\alpha\|_1!} D^\alpha m\left(Z_i, \theta_0, h_0\left(X_i\right)\right)\left(\hat{h}\left(X_i\right) - h_0\left(X_i\right)\right)^\alpha}_{=:B_3} \\
&+ \underbrace{\frac{1}{\sqrt{n}} \sum_{i=1}^{n} \sum_{\alpha \in \mathcal{I}_{k+1,\ell,0} \cap S_1} \frac{1}{(k+1)!} D^\alpha m_1\left(Z_i, \theta_0, \tilde{h}\left(X_i\right)\right)\left(\hat{h}\left(X_i\right) - h_0\left(X_i\right)\right)^\alpha}_{=:B_4} \\
&+ \underbrace{\frac{1}{\sqrt{n}} \sum_{i=1}^{n} \sum_{\alpha \in \mathcal{I}_{k+1,\ell,0} \backslash S_1} \frac{1}{(k+1)!} D^\alpha m_1\left(Z_i, \theta_0, \tilde{h}\left(X_i\right)\right)\left(\hat{h}\left(X_i\right) - h_0\left(X_i\right)\right)^\alpha}_{=:B_5}
\end{aligned}
\tag{61}
$$

where we recall that $\mathcal{I}_{k,\ell} = \{\alpha \in \mathbb{Z}_{\geq 0}^\ell : \|\alpha\|_1 \leq k\}$. First, it is easy to see that with high probability, it holds that

$$
B_1 \lesssim \operatorname{Var}\left(m(Z, \theta_0, h_0(X))\right)^{1/2}.
$$

Second, by our assumption on the error $\hat{h} - h_0$, we have

$$
\begin{aligned}
\mathbb{E}\left[|B_3|\right] &\leq \sum_{\alpha \in \mathcal{I}_{k,\ell} \backslash S_0} \frac{\sqrt{n}}{\|\alpha\|_1!} \mathbb{E}\left[\left|D^\alpha m\left(Z, \theta_0, h_0\left(X\right)\right)\left(\hat{h}\left(X\right) - h_0\left(X\right)\right)^\alpha\right|\right] \\
&\leq \sum_{\alpha \in \mathcal{I}_{k,\ell} \backslash S_0} \frac{\sqrt{n}}{\|\alpha\|_1!} \mathbb{E}\left[\left|D^\alpha m\left(Z, \theta_0, h_0\left(X\right)\right)\right|^p\right]^{1/p} \mathbb{E}\left[\left|\hat{h}\left(X\right) - h_0\left(X\right)\right|^{\alpha q}\right]^{1/q} \\
&\leq \lambda_\star \sum_{\alpha \in \mathcal{I}_{k,\ell} \backslash S_0} \frac{\sqrt{n}}{\|\alpha\|_1!} \prod_{i=1}^\ell \epsilon_i^{\alpha_i} \leq \sqrt{n} \lambda_\star \sum_{\alpha \in \mathcal{I}_{k,\ell} \backslash S} \prod_{i=1}^\ell \epsilon_i^{\alpha_i},
\end{aligned}
\tag{62}
$$

where the last step follows from Holder's inequality:

$$
\begin{aligned}
\left\|\prod_{i=1}^\ell \left|\hat{h}_i(X) - h_{0i}(X)\right|^{\alpha_i}\right\|_{P_X, q} &\leq \prod_{i=1}^\ell \left\|\left|\hat{h}_i(X) - h_{0i}(X)\right|^{\alpha_i}\right\|_{P_X, s_i/\alpha_i} \\
&= \prod_{i=1}^\ell \left\|\hat{h}_i(X) - h_{0i}(X)\right\|_{P_X, s_i}^{\alpha_i} \leq \prod_{i=1}^\ell \epsilon_i^{\alpha_i}.
\end{aligned}
$$

Similarly, we have

$$
\mathbb{E}[|B_5|] \leq \frac{\sqrt{n}\lambda_\star}{(k+1)!} \sum_{\alpha \in \mathcal{I}_{k+1,\ell,0} \backslash S_1} \prod_{i=1}^\ell \epsilon_i^{\alpha_i}.
$$

Finally, the arguments in [Mackey et al., 2018, Section A.2] imply that $B_2, B_4 = o_P(1)$. Combining everything above, we conclude the proof of Theorem E.1.

Under the assumptions in Corollary E.1, it holds that $B_3, B_5 = o(n^{-1/2})$. As a result, the same arguments in [Mackey et al., 2018, Section A.2] would imply the desired asymptotic normality result in Corollary E.1.

### E.3 Proof of Theorem 4.1: Structure-agnostic error from estimated moments

The proof follows a similar argument as the proof of the previous theorem. Consider any probability distribution $P \in \mathcal{P}$. We define $\mathcal{E}$ as the "good" event that the dataset $\mathcal{D}_1$ satisfies the conditions (12), (13), (14) and (15). By assumption, we known that $\mathbb{P}[\mathcal{E}] \geq 1 - \gamma/2$. Our subsequent analysis consider a *fixed* choice of $\mathcal{D}_1$ that fails into $\mathcal{E}$. Note that the moment function $\hat{m}_r(\cdot)$ is partially linear in $q$, so for any index $\alpha = (\alpha_1, \alpha_2)$, if $\alpha_2 \geq 2$, then $D^\alpha \hat{m}_r = 0$. Now let's calculate the derivative for $\alpha_2 \in \{0, 1\}$.

When $\alpha_2 = 0$, we have

$$D^\alpha \hat{m}_r(Z, \theta, h; \mathcal{D}_1)$$
$$= \left[ Y - q(X) - \theta(T - g(X)) \right] \hat{J}_r^{(\alpha_1)}(T - g(X), X; \mathcal{D}_1) + \theta \hat{J}_r^{(\alpha_1 - 1)}(T - g(X), X; \mathcal{D}_1),$$

so that under $\mathcal{E}$, we have

$$\left| \mathbb{E}\left[ D^\alpha \hat{m}_r(Z, \theta, h; \mathcal{D}_1) \mid X \right] \right| = \left| \theta \mathbb{E}\left[ \hat{J}_r^{(\alpha_1 - 1)}(T - g(X), X; \mathcal{D}_1) \right] \right| \leq C_\theta \epsilon^{\alpha_1 - 1}$$

for all $\alpha_1 \leq j + 1$. When $\alpha_2 = 1$, we have

$$D^\alpha \hat{m}_r(Z, \theta, q, g) = -\hat{J}_r^{(\alpha_1)}(T - g(X), X; \mathcal{D}_1),$$

so that

$$\left| \mathbb{E}\left[ D^\alpha \hat{m}_r(Z, \theta, q, g) \mid X \right] \right| = \left| \mathbb{E}\left[ \hat{J}_r^{(\alpha_1)}(T - g(X), X; \mathcal{D}_1) \mid X \right] \right| \leq \epsilon^{\alpha_1}$$

for all $\alpha_1 \leq j$. The above derivations also imply that

$$\|D^{(r+1,0)} \hat{m}_r(Z, \theta, h; \mathcal{D}_1)\|_{L^{s/2}(P)}$$
$$\leq \|Y - q(X) - \theta(T - g(X))\|_{L^s(P)} \left\| \hat{J}_r^{(r+1)}(T - g(X), X; \mathcal{D}_1) \right\|_{L^s(P)}$$
$$\quad + C_\theta \left\| \hat{J}_r^{(r)}(T - g(X), X; \mathcal{D}_1) \right\|_{L^s(P)}$$
$$\leq \left( \|Y - q(X)\|_{L^s(P)} + C_\theta \|T - g(X)\|_{L^s(P)} \right) \left\| \hat{J}_r^{(r+1)}(T - g(X), X; \mathcal{D}_1) \right\|_{L^s(P)}$$
$$\quad + C_\theta \left\| \hat{J}_r^{(r)}(T - g(X), X; \mathcal{D}_1) \right\|_{L^s(P)}$$
$$\leq 4(\psi_\xi + C_\theta C_\eta) \sqrt{s} \left\| \hat{J}_r^{(r+1)}(T - g(X), X; \mathcal{D}_1) \right\|_{L^s(P)} + \psi_\theta \left\| \hat{J}_r^{(r)}(T - g(X), X; \mathcal{D}_1) \right\|_{L^s(P)}$$
$$\leq \left[ 4(\psi_\xi + C_\theta C_\eta) \sqrt{s} + C_\theta \right] \Lambda_r,$$
$$\tag{63}$$

and

$$\|D^{(r,1)} \hat{m}_r(Z, \theta, h; \mathcal{D}_1)\|_{L^{s/2}(P)} \leq \left\| \hat{J}_r^{(r)}(T - g(X), X; \mathcal{D}_1) \right\|_{L^{s/2}(P)} \leq \Lambda_r, \tag{64}$$

where we use the assumed property that

$$\max_{\beta \in \{r, r+1\}} \sup_{h \in \mathcal{B}_{P,s}(h_0, \Delta)} \left\| \hat{J}_r^{(\beta)}(T - g(X), X; \mathcal{D}_1) \right\|_{L^\infty(P)} \leq \Lambda_r < +\infty.$$

Let $J(\mathcal{D}_1) = \mathbb{E}_{Z \sim P}[m(Z, \theta_0, h_0(X); \mathcal{D}_1) \mid \mathcal{D}_1]$. Similar to the proof of the previous theorem, we consider the decomposition

$$\hat{\theta} - \theta_0 = J^{-1} \frac{2}{n} \sum_{i=n/2+1}^{n} m(Z_i, \theta_0, \hat{h}(X_i); \mathcal{D}_1) + o_P(n^{-1/2})$$

By Chebyshev's inequality, with probability $\geq 1 - \gamma/2$, we have

$$\left| \frac{2}{n} \sum_{i=n/2+1}^{n} m(Z_i, \theta_0, \hat{h}(X_i); \mathcal{D}_1) - \mathbb{E}\left[ m(Z, \theta_0, \hat{h}(X); \mathcal{D}_1) \mid \mathcal{D}_1 \right] \right| \leq 2\gamma^{-1} V_m n^{-1/2},$$

which implies that

$$|\hat{\theta} - \theta_0| \leq \delta_{\mathsf{id}}^{-1} \left( \mathbb{E}\left[ m(Z, \theta_0, \hat{h}(X); \mathcal{D}_1) \mid \mathcal{D}_1 \right] + 2\gamma^{-1} V_m n^{-1/2} \right) + o_P(n^{-1/2}).$$

Finally, by Taylor's formula and the orthogonality condition,

$$\big|\mathbb{E}\big[m(Z,\theta_0,\hat{h}(X);\mathcal{D}_1)\big]\big|$$

$$= \big|\mathbb{E}\big[m(Z,\theta_0,\hat{h}(X);\mathcal{D}_1)\big] - \mathbb{E}\big[m(Z,\theta_0,h_0(X);\mathcal{D}_1)\big]\big|$$

$$= \sum_{j=1}^{r} \sum_{\alpha \in \{(j,0),(j-1,1)\}} \frac{1}{\|\alpha\|_1!} \big|\mathbb{E}\big[D^\alpha m(Z,\theta_0,h_0(X);\mathcal{D}_1)\,(\hat{h}(X)-h_0(X))^\alpha\big]\big|$$

$$\qquad + \sum_{\alpha \in \{(r+1,0),(r,1)\}} \frac{1}{\|\alpha\|_1!} \big|\mathbb{E}\big[D^\alpha m(Z,\theta_0,\tilde{h}(X);\mathcal{D}_1)\,(\hat{h}(X)-h_0(X))^\alpha\big]\big|$$

(where $\tilde{h} = h_0 + t(\hat{h} - h_0),\ t \in [0,1]$)

$$\leq \sum_{j=1}^{r} \sum_{\alpha \in \{(j,0),(j-1,1)\}} \frac{1}{\|\alpha\|_1!} \mathbb{E}\Big[\big|\mathbb{E}\big[D^\alpha m(Z,\theta_0,h_0(X);\mathcal{D}_1) \mid X\big](\hat{h}(X)-h_0(X))^\alpha\big|\Big]$$

$$\qquad + \sum_{\alpha \in \{(k+1,0),(k,1)\}} \frac{1}{\|\alpha\|_1!} \big\|D^\alpha m(Z,\theta_0,\tilde{h}(X);\mathcal{D}_1)\big\|_{L^{s/2}(P)}\, \epsilon_1^{\alpha_1}\,\epsilon_2^{\alpha_2}$$

$$\leq \sum_{j=1}^{r} \Big(\frac{1}{j!}\,\epsilon^{(j)}\epsilon_1^j + \frac{1}{(j-1)!}\,\epsilon^{(j-1)}\epsilon_1^{j-1}\epsilon_2\Big) + \frac{1}{(k+1)!}\Big[\big(4(\psi_\xi + C_\theta\psi_\eta)\sqrt{s} + C_\theta\big)\epsilon_1^{r+1} + r\,\epsilon_1^r\epsilon_2\Big]\Lambda_r$$

$$\leq \sum_{j=0}^{r-1} \frac{1}{j!}\,\max\{\epsilon^{(j+1)},\epsilon^{(j)}\}\big(\epsilon_1^{j+1} + \epsilon_1^j\epsilon_2\big) + \frac{1}{(r+1)!}\Big[\big(4(\psi_\xi + C_\theta\psi_\eta)\sqrt{s} + C_\theta\big)\epsilon_1^{r+1} + r\,\epsilon_1^r\epsilon_2\Big]\Lambda_r,$$

(65)

where the third step uses Holder's inequality and $s \geq 2r + 2$. This yields the desired bound. The total probability for this bound to hold is $(1-\gamma)^2 \geq 1 - 2\gamma$, as desired.

### E.4   Proof of Lemma 4.1: Explicit formula for $J_r$

We prove this lemma by induction on $r$. When $r = 1$, the conclusion automatically holds by the assumed expression of $J_1(w,x)$. Now suppose that $r \geq 2$, and the conclusion holds for $r-1$, then by definition,

$$I_r(w,x) = \int_0^w J_{r-1}(w',x)\mathrm{d}w' = \sum_{i=1}^{M_{r-1}} a_{ir}(x) \int_0^w \rho_{i,r-1}(w')\mathrm{d}w = \sum_{i=1}^{M_{r-1}} a_{ir}(x)\rho_{i+1,r}(x)$$

and the conclusion follows.

## F   Technical details in Section 5.1

In this section, we provide additional results and details that complement Section 5.1.

We first consider the problem of estimating $\mu_r := \mathbb{E}[\eta^r]$, where $r \geq 2$ is some positive integer. It turns out that even under the independence noise assumption, there exists a fundamental bottleneck for estimating $\mu_r$, as stated in the following theorem. Interestingly, this statistical limit is different for $r = 3$ and all other values of $r$.

**Theorem F.1** (Structure-agnostic limit for estimating residual moments). *Let $C_T > 0$ be a constant and $\mathcal{P}_0$ be the set of all distributions of $(X,T)$ generated from $T = g_0(X) + \eta$, such that $|T| \leq C_T, a.s.$ and $\eta$ is some mean-zero noise variable independent of $X$. Let $\Phi$ be a mapping that maps $P_0 \in \mathcal{P}_0$ to the nuisance function $g_0$, and the target parameter is defined by $\theta(P_0) = \mathbb{E}[(T - g_0(X))^r]$. Then for any $\gamma \in (1/2,1)$ and $r \in \mathbb{Z}_+$, there exists a constant $c_{\gamma,r} > 0$ such that $\mathfrak{M}_{n,1-\gamma}\big(\mathcal{P}_{\infty,\epsilon}(\hat{g})\big) \geq c_\gamma\big(\epsilon^{\alpha_r} + n^{-1/2}\big)$, where $\alpha_r = 3$ if $r = 3$ and $\alpha_r = 2$ otherwise. Moreover, these rates can be attained by $\theta = \theta_r$, where $\{\theta_k\}_{k=1}^{+\infty}$ is recursively defined as $\theta_1 = 0,\quad \theta_k = \mu_k' - k\theta_{k-1}\mu_1'\quad (k \geq 2)$, where $\mu_k' = \frac{1}{n}\sum_{i=1}^{n}(T_i - \hat{g}(X_i))^k$.*

The remaining part of this section is devoted to proving Theorem F.1 and Theorem 5.2.

## F.1 Proof of Theorem F.1: Structure-agnostic limit for estimating residual moments

The proof is based on the method of fuzzy hypothesis, as introduced in Lemma A.1. Let $X \sim P_X$ be uniformly distributed on $[0, 1]$ and $\eta$ be a random variable independent of $X$, with density

$$p_\eta(z) = \begin{cases} \exp(-z^4) & \text{if } z \leq 0 \\ \exp(-1.5z^4 + az^3) & \text{if } z > 0, \end{cases}$$

where $a > 0$ is chosen such that $\mathbb{E}[\eta] = 0$. Let $M$ and $\lambda = (\lambda_1, \lambda_2, \cdots, \lambda_M)$ where $\lambda_i$'s i.i.d. random variables such that $\lambda_i = 2$ with probability $\frac{1}{3}$ and $= -1$ with probability $\frac{2}{3}$. Let $B_1, B_2, \cdots, B_M$ be a partition of $\mathcal{X} = [0, 1]$ such that $P_X(B_i) = \frac{1}{M}, \forall i \in [M]$. Define $P_\lambda$ to be the joint distribution of $(X, T)$ generated from

$$X \sim P_X, \quad T = g_\lambda(X) + \eta,$$

where $g_\lambda(x) = \hat{g}(x) + \frac{1}{2}\epsilon_g \Delta(\lambda, x)$, and $\hat{P} = \int P_\lambda d\pi(\lambda)$. It is easy to see that $P_\lambda \in \mathcal{Q}_0$.

Let $\hat{p}$ and $p_\lambda$ be the density of $\hat{P}$ and $\hat{P}_\lambda$ respectively, then the above definitions and Taylor's formula together imply that

$$p_\lambda(x, t) = p_X(x)p_\eta(t - g_\lambda(x)) = p_\eta(t - \hat{g}(x) - \epsilon_g\Delta(\lambda, x)/2)$$

$$= \sum_{i=0}^{+\infty} \frac{1}{(-2)^i i!} \Delta(\lambda, x)^i \epsilon_g^i p_\eta^{(i)}(t - \hat{g}(x)).$$

For any given $X = x$, $\mathbb{E}_\pi[\Delta(\lambda, x)^i] = \frac{2}{3}(2^{i-1} + (-1)^i)$ is independent of $x$, thus $\mathbb{E}_\pi[p_\lambda(x, t)]$ only depends on $x, t$ through $t - \hat{g}(x)$. As a result, we can define a random variable $\hat{\eta}$ which is independent of $X$ and has density $p_{\hat{\eta}}(t - \hat{g}(x)) = \mathbb{E}_\pi[p_\lambda(x, t)]$. The data generating process

$$X \sim P_X, \quad T = \hat{g}(X) + \hat{\eta}$$

thus induces a density $p_X(x)p_{\hat{\eta}}(t - \hat{g}(x)) = \mathbb{E}_\pi[p_\lambda(x, t)] = \hat{p}(x, t)$.

We choose $P = \hat{P}$, $Q_\lambda = P_\lambda$ and $\mathcal{Z}_j = B_j \times \mathcal{T}$ in Lemma A.2, the corresponding $p_j = \frac{1}{M}$. For any $t \in \mathcal{T}$, $x \in \mathcal{X}$ and $\lambda \in \text{supp}(\pi)$, we have

$$\frac{p_\eta(t - \hat{g}(x) - \epsilon_g\Delta(\lambda, x)/2)^2}{p_{\hat{\eta}}(t - \hat{g}(x))} \leq \frac{p_\eta(t - \hat{g}(x) - \epsilon_g)^2}{p_\eta(t - \hat{g}(x) + \epsilon_g/2)}$$

$$\leq \exp\left(-2(t - \hat{g}(x) - \epsilon_g)^4 + 1.5(t - \hat{g}(x) + \epsilon_g/2)^4 + a|t - \hat{g}(x) + \epsilon_g/2|^3\right)$$

$$\leq \underbrace{\exp\left(-(t - \hat{g}(x))^4/2 + 11|t - \hat{g}(x)|^3\epsilon_g + 9|t - \hat{g}(x)|\epsilon_g^3/2 + a|t - \hat{g}(x) + \epsilon_g/2|^3\right)}_{\Gamma(t - \hat{g}(x))}.$$

since $\hat{g}$ is assumed to be uniformly bounded, it is easy to see that $x \mapsto \int_{\mathbb{R}} \Gamma(t - \hat{g}(x))dt$ is uniformly bounded as well. Therefore, in the setting of Lemma A.2, we have

$$b = M \max_j \sup_\lambda \int_{\mathcal{Z}_j} \frac{(p_\lambda - \hat{p})^2}{p} d\mu$$

$$\leq M \max_j \sup_\lambda \int_{B_j} dx \int_{\mathbb{R}} \left(\hat{p}(x, t) + \frac{p_\lambda(x, t)^2}{\hat{p}(x, t)}\right) dt$$

$$\leq \max_{x \in \mathcal{X}} \sup_\lambda \int_{\mathbb{R}} \left(\hat{p}(x, t) + \frac{p_\lambda(x, t)^2}{\hat{p}(x, t)}\right) dt$$

$$\leq \max_{x \in \mathcal{X}} \sup_\lambda \int_{\mathbb{R}} \left(\hat{p}(x, t) + \Gamma(t - \hat{g}(x))\right) dt$$

$$< +\infty$$

is bounded by some universal constant, which we denote by $\bar{b}$. Let $C$ be the constant in Lemma A.2 that corresponds to $A = 1$, and choose $M \geq \max\left\{n \max\{1, \bar{b}\}, C\delta^{-1}n^2\bar{b}^2\right\}$, then we have that

$$H\left(\hat{P}^{\otimes n}, \int P_\lambda^{\otimes n} d\pi(\lambda)\right) \leq \delta.$$

The final step is to verify the separation condition (25). Specifically, we choose $\mathcal{P} = \mathcal{Q}_0$ and define $T(P)$ to be $-\mu_r$ for any $P \in \mathcal{P}$. We abuse notation and use $\mu_r(P)$ to denote the value of $\mu_r$ corresponds to $P \in \mathcal{P}$. Then we have that

$$\mu_r(\hat{P}) = \int z^r p_{\hat{\eta}}(z)\mathrm{d}z = \mu_r(P_\lambda) + \sum_{i=1}^{3} \frac{2^{i-1}+(-1)^i}{3(-2)^{i-1}i!}\epsilon_g^i \int z^r p_\eta^{(i)}(z)\mathrm{d}z + \mathcal{O}(\epsilon_g^4).$$

Note that for $i \leq r$ we have

$$\int z^r p_\eta^{(i)}(z)\mathrm{d}z = (-1)^i \frac{r!}{(r-i)!} \int z^{r-i} p_\eta(z)\mathrm{d}z.$$

In particular, we consider the case where $i = 2$. If $r \neq 3$ then the above equation is nonzero, implying that

$$\mu_r(\hat{P}) = \mu_r(P_\lambda) + \Theta(\epsilon_g^2).$$

Therefore, Lemma A.1 can be applied with $s = \Theta(\epsilon_g^2)$, which yields the desired result.

If $r = 3$, then $\int z^{r-2}p_\eta(z)\mathrm{d}z = \mathbb{E}[\eta] = 0$, so that

$$\mu_r(\hat{P}) = \mu_r(P_\lambda) + \Theta(\epsilon_g^3),$$

and the conclusion can be similarly derived.

### F.2 Proofs of Theorem 5.1 and Theorem 5.2

In this subsection, we present the proofs of Theorem 5.1 and Theorem 5.2. The proof techniques are largely similar. We choose to start with the proof of Theorem 5.2, which is more involved.[5]

**Proof of Theorem 5.2** For any $P \in \mathcal{P}$, let $\bar{\mu}_k' = \mathbb{E}_P[(T - \hat{g}(X))^k]$, then it is easy to see that

$$|\bar{\mu}_k'| \leq 2^{k-1}\left(\mathbb{E}\left[(T - g_0(X))^k\right] + \mathbb{E}\left[(g_0(X) - \hat{g}(X))^k\right]\right) \leq 2^{2k}\left(k^{k/2}\psi_\eta^k + C_g^k\right). \quad (66)$$

By Chebyshev's inequality, we have

$$\mathbb{P}[|\mu_k' - \bar{\mu}_k'| > \delta_k] \leq \frac{1}{\delta^2 n}\mathrm{Var}\left((T - \hat{g}(X))^k\right) \leq \frac{1}{\delta^2 n}\mathbb{E}\left[(T - \hat{g}(X))^{2k}\right]$$

$$= \frac{1}{\delta^2 n}\mathbb{E}\left[\left(\eta + g_0(X) - \hat{g}(X)\right)^{2k}\right] = \frac{2^{4k}(C_g^{2k} + k^k\psi_\eta^{2k})}{\delta_k^2 n},$$

where the last step uses $|g|, |\hat{g}| \leq C_g$ and $\|\eta\|_{\psi_2} \leq \psi_\eta$. We choose $\delta_k = r^{1/2}(\gamma n)^{-1/2}2^{2k}(C_g^{2k} + k^k\psi_\eta^{2k})^{1/2}$, then it is easy to see that with probability $\geq 1 - \gamma$, $|\mu_k' - \bar{\mu}_k'| \leq \delta_k$ for all $k \in [r]$. Let $\mathcal{E}$ be the event that all these inequalities hold. The following lemma bounds the difference between $\theta$ and its population version (with $\mu_l'$ replaced by $\bar{\mu}_l'$), defined as

$$\bar{\theta}_r = (-1)^{r+1}\begin{vmatrix} \bar{\mu}_1' & 1 & 0 & 0 & 0 & 0 & \ldots & 0 \\ \bar{\mu}_2' & \bar{\mu}_1' & 1 & 0 & 0 & 0 & \ldots & 0 \\ \bar{\mu}_3' & \bar{\mu}_2' & \binom{2}{1}\bar{\mu}_1' & 1 & 0 & 0 & \ldots & 0 \\ \vdots & \vdots & \vdots & \vdots & \vdots & \ddots & \ddots & \vdots \\ \bar{\mu}_r' & \bar{\mu}_{r-1}' & \ldots & \ldots & \ldots & \ldots & \ldots & \binom{r-1}{r-2}\bar{\mu}_1' \end{vmatrix},$$

**Lemma F.1** (Moment–to–cumulant type bounds). *Let the sequences* $\{\theta_k\}_{k\geq 1}$, $\{\bar{\theta}_k\}_{k\geq 1}$, $\{\mu_k'\}_{k\geq 1}$ *and* $\{\bar{\mu}_k'\}_{k\geq 1}$ *satisfy the recursions*

$$\theta_1 = \mu_k', \quad \bar{\theta}_1 = \bar{\mu}_1' \quad (67)$$

$$\theta_k = \mu_k' - \sum_{j=1}^{k-1}\binom{k-1}{j-1}\mu_{k-j}'\theta_j, \quad (68)$$

$$\bar{\theta}_k = \bar{\mu}_k' - \sum_{j=1}^{k-1}\binom{k-1}{j-1}\bar{\mu}_{k-j}'\bar{\theta}_j. \quad (69)$$

---

[5]Although Theorem 5.1 consider a more general class of noise, the rate is strictly looser in the setting of Theorem 5.2, as discussed in Remark 5.1.

*Assume there exist constants $C_{\mathrm{g}}, \psi_{\eta} > 0$ and $l, \gamma, n \in (0, \infty)$ such that for every $k \geq 1$*

$$|\bar{\mu}'_k| \leq 2^{2k}\Big(C_{\mathrm{g}}^k + k^{k/2}\psi_{\eta}^k\Big), \tag{70}$$

$$|\mu'_k - \bar{\mu}'_k| \leq r^{1/2}(\gamma n)^{-1/2}\, 2^{2k}\Big(C_{\mathrm{g}}^{2k} + k^k\psi_{\eta}^{2k}\Big)^{1/2}. \tag{71}$$

*Then, for all $k \geq 1$,*

$$|\bar{\theta}_k| \leq \big[8k(C_{\mathrm{g}} + \psi_{\eta})\big]^k, \tag{72}$$

$$|\bar{\theta}_k - \theta_k| \leq 3\big[12k(C_{\mathrm{g}} + \psi_{\eta})\big]^k r^{1/2}(\gamma n)^{-1/2}. \tag{73}$$

***Proof*** : Throughout the argument write

$$A_k := 2^{2k}(C_{\mathrm{g}} + \psi_{\eta})^k k^{k/2}, \qquad D_k := r^{1/2}(\gamma n)^{-1/2}\, A_k.$$

(70)–(71) imply that $|\bar{\mu}'_k| \leq A_k$ and $|\mu'_k - \bar{\mu}'_k| \leq D_k$. By Triangle inequality, (68) implies that

$$|\theta_k| \leq A_k + \sum_{j=1}^{k-1} \binom{k-1}{j-1} A_{k-j}|\theta_j|.$$

Let $\bar{\theta}_k = A_k\rho_k$, then this becomes

$$|\rho_k| \leq 1 + \sum_{j=1}^{k-1} \binom{k-1}{j-1} \frac{A_j A_{k-j}}{A_k}|\rho_j|. \tag{74}$$

We now prove that

$$\binom{k-1}{j-1} \frac{A_j A_{k-j}}{A_k} \leq \frac{k^{k/2}}{j^{j/2}(k-j)^{(k-j)/2}}. \tag{75}$$

Indeed, since

$$\sqrt{2\pi n}\left(\frac{n}{e}\right)^n < n! < \sqrt{2\pi n}\left(\frac{n}{e}\right)^n e^{\frac{1}{12n}}, \quad \forall n \geq 1,$$

we can deduce that

$$\binom{k-1}{j-1} \leq \binom{k}{j} < \frac{\sqrt{2\pi k}(k/e)^k \exp(1/(12k))}{\sqrt{2\pi j}(j/e)^j \cdot \sqrt{2\pi(k-j)}((k-j)/e)^{k-j}} \leq \frac{k^k}{j^j(k-j)^{k-j}}.$$

Hence,

$$\binom{k-1}{j-1} \frac{A_j A_{k-j}}{A_k} \leq \frac{k^k}{j^j(k-j)^{k-j}} \cdot \frac{j^{j/2}(k-j)^{(k-j)/2}}{k^{k/2}} \leq \frac{k^{k/2}}{j^{j/2}(k-j)^{(k-j)/2}},$$

proving (75). Plugging into (74) and rearranging, we obtain

$$k^{-k/2}|\rho_k| \leq k^{-k/2} + \sum_{j=1}^{k-1}(k-j)^{-(k-j)/2}j^{-j/2}|\rho_j| \leq k^{-k/2} + \sum_{j=1}^{k-1}j^{-j/2}|\rho_j|.$$

$$|\rho_k| \leq 3^{k-1}(k-1)!, \quad \forall k \geq 1. \tag{76}$$

Define

$$S_k := \sum_{i=1}^{k} i^{-i/2}|\rho_i|, \quad k \geq 1.$$

Then we have that

$$S_k \leq 2S_{k-1} + k^{-k/2}$$
$$S_k + k^{-k/2} \leq 2\big(S_{k-1} + (k-1)^{-(k-1)/2}\big). \tag{77}$$

Moreover,

$$S_1 = |\rho_1| = \frac{|\bar{\mu}'_1|}{4(C_{\mathrm{g}} + \psi_{\eta})} \leq \frac{\epsilon_1}{4\psi_{\eta}} \leq \frac{1}{4},$$

so it follows from (77) that $S_k \leq 2^k$. Finally, we have

$$|\bar{\theta}_k| \leq A_k|\rho_k| \leq A_k \cdot 2^k k^{k/2} \leq \big[8k(C_{\mathsf{g}} + \psi_{\mathfrak{n}})\big]^k.$$

We now turn to bound $\bar{\theta}_k - \theta_k$. Set $\Delta_k := \bar{\theta}_k - \theta_k$. Subtracting (68) from (69) gives

$$\Delta_k = (\bar{\mu}'_k - \mu'_k) - \sum_{j=1}^{k-1} \binom{k-1}{j-1}\Big[(\bar{\mu}'_{k-j} - \mu'_{k-j})\bar{\theta}_j + \mu'_{k-j}\Delta_j\Big].$$

Taking absolute values and invoking the bounds already proved yields

$$
\begin{aligned}
|\Delta_k| &\leq D_k + \sum_{j=1}^{k-1} \binom{k-1}{j-1}\Big(D_{k-j}|\bar{\theta}_j| + (|\bar{\mu}'_{k-j}| + D_{k-j})|\Delta_j|\Big) \\
&\leq D_k + \sum_{j=1}^{k-1} \binom{k-1}{j-1}\Big[r^{1/2}(\gamma n)^{-1/2}A_{k-j} \cdot 2^j j^{j/2}A_j + (|\bar{\mu}'_{k-j}| + D_{k-j})|\Delta_j|\Big] \\
&\leq D_k + r^{1/2}(\gamma n)^{-1/2}k^{k/2}A_k \sum_{j=1}^{k-1}(k-j)^{-(k-j)/2} + \sum_{j=1}^{k-1}\binom{k-1}{j-1}(A_{k-j} + D_{k-j})|\Delta_j| \\
&\leq D_k + 4r^{1/2}(\gamma n)^{-1/2}k^{k/2}A_k + 2\sum_{j=1}^{k-1}\binom{k-1}{j-1}A_{k-j}|\Delta_j| \\
&\leq 8r^{1/2}(\gamma n)^{-1/2}k^{k/2}A_k + 2\sum_{j=1}^{k-1}\binom{k-1}{j-1}A_{k-j}|\Delta_j|.
\end{aligned}
\tag{78}
$$

Let $\Delta_k = A_k\delta_k$, then (75) and (78) implies that

$$
\begin{aligned}
|\delta_k| &\leq 8r^{1/2}(\gamma n)^{-1/2}k^{k/2} + 2\sum_{j=1}^{k-1}\binom{k-1}{j-1}\frac{A_{k-j}A_j}{A_k}|\delta_j| \\
&\leq 8r^{1/2}(\gamma n)^{-1/2}k^{k/2} + 2k^{k/2}\sum_{j=1}^{k-1}(k-j)^{-(k-j)/2}j^{-j/2}|\delta_j|
\end{aligned}
\tag{79}
$$

Define

$$T_k := \sum_{i=1}^{k} i^{-i/2}|\delta_i|, \quad k \geq 1.$$

Then we have that

$$
\begin{aligned}
T_k &\leq T_{k-1} + 8r^{1/2}(\gamma n)^{-1/2} + 2\sum_{j=1}^{k-1}(k-j)^{-(k-j)/2}j^{-j/2}|\delta_j| \\
&\leq 8r^{1/2}(\gamma n)^{-1/2} + 3T_{k-1}
\end{aligned}
$$

which further implies

$$T_k + 4r^{1/2}(\gamma n)^{-1/2} \leq 3\big(T_{k-1} + 4r^{1/2}(\gamma n)^{-1/2}\big). \tag{80}$$

Moreover,

$$T_1 = |\delta_1| = \frac{|\mu_k - \mu'_k|}{4(C_{\mathsf{g}} + \psi_{\mathfrak{n}})} \leq \frac{1}{4}r^{1/2}(\gamma n)^{-1/2},$$

so it follows from (80) that $T_r \leq 3^{r+1}r^{1/2}(\gamma n)^{-1/2}$. Therefore, $|\Delta_r| = A_r|\delta_r| \leq r^{r/2}T_r A_r \leq r^{r/2}3^{r+1}r^{1/2}(\gamma n)^{-1/2}A_r$, concluding the proof. $\qquad\square$

In view of the previous lemma, we only need to bound the difference between $\bar{\theta}_l$ and $\kappa_l$. Note the following well-known property of cumulants $\{\kappa_l\}_{l=1}^{+\infty}$:

$$\log \mathbb{E}_{P_0}[e^{t\eta}] = \sum_{l=1}^{+\infty} \kappa_l \frac{t^l}{l!}.$$

Let $\hat{\eta} = T - \hat{g}(X)$ and $D_X = g_0(X) - \hat{g}(X)$, then $\hat{\eta} = \eta + D_X$ and $\eta, D_X$ are independent by our assumption. By definition, $\bar{\theta}_l$ is the $l$-th order cumulant of $T - \hat{g}(X)$. Hence,

$$\sum_{l=1}^{+\infty} \bar{\theta}_l \frac{t^l}{l!} = \log \mathbb{E}[e^{t\hat{\eta}}] = \log \mathbb{E}[e^{t\eta}] + \log \mathbb{E}[e^{tD_X}] = \sum_{l=1}^{+\infty} (\kappa_l + d_l) \frac{t^l}{l!},$$

where $d_l$ is the $l$-th order cumulant of the variable $D_X$. From Proposition A.2 we can deduce that $d_r \leq (2r\epsilon)^r$. combining with Lemma F.1, the conclusion follows.

**Proof of Theorem 5.1** For any $P \in \mathcal{P}$, let $\bar{\mu}'_k = \mathbb{E}_P[(T - \hat{g}(X))^k]$, then it is easy to see that

$$|\bar{\mu}'_k| \leq 2^{k-1}\big(\mathbb{E}\big[|T|^k\big] + \mathbb{E}\big[|\hat{g}(X)|^k\big]\big) \leq 2^k C_\mathsf{T}^k. \tag{81}$$

By Chebyshev's inequality, we have

$$\mathbb{P}[|\mu'_k - \bar{\mu}'_k| > \delta_k] \leq \frac{1}{\delta^2 n} \mathrm{Var}\left((T - \hat{g}(X))^k\right) \leq \frac{1}{\delta^2 n} \mathbb{E}\left[(T - \hat{g}(X))^{2k}\right] \leq \frac{2^{2k} C_\mathsf{T}^{2k}}{\delta_k^2 n}.$$

We choose $\delta_k = r^{1/2}(\gamma n)^{-1/2}(2C_\mathsf{T})^k$, then it is easy to see that with probability $\geq 1 - \gamma$, $|\mu'_k - \bar{\mu}'_k| \leq \delta_k$ for all $k \in [l]$. Let $\mathcal{E}$ be the event that all these inequalities hold. The following lemma bounds the difference between $\theta$ and its population version (with $\mu'_l$ replaced by $\bar{\mu}'_l$), defined as

$$\bar{\theta}_r = (-1)^{r+1} \begin{vmatrix} \bar{\mu}'_1 & 1 & 0 & 0 & 0 & 0 & \cdots & 0 \\ \bar{\mu}'_2 & \bar{\mu}'_1 & 1 & 0 & 0 & 0 & \cdots & 0 \\ \bar{\mu}'_3 & \bar{\mu}'_2 & \binom{2}{1}\bar{\mu}'_1 & 1 & 0 & 0 & \cdots & 0 \\ \vdots & \vdots & \vdots & \vdots & \vdots & \ddots & \ddots & \vdots \\ \bar{\mu}'_r & \bar{\mu}'_{r-1} & \cdots & \cdots & \cdots & \cdots & \cdots & \binom{r-1}{r-2}\bar{\mu}'_1 \end{vmatrix},$$

**Lemma F.2** (Moment–to–cumulant type bounds). *Let the sequences* $\{\theta_k\}_{k\geq 1}$, $\{\bar{\theta}_k\}_{k\geq 1}$, $\{\mu'_k\}_{k\geq 1}$ *and* $\{\bar{\mu}'_k\}_{k\geq 1}$ *satisfy the recursions*

$$\theta_1 = \mu'_k, \quad \bar{\theta}_1 = \bar{\mu}'_1 \tag{82}$$

$$\theta_k = \mu'_k - \sum_{j=1}^{k-1} \binom{k-1}{j-1} \mu'_{k-j} \theta_j, \tag{83}$$

$$\bar{\theta}_k = \bar{\mu}'_k - \sum_{j=1}^{k-1} \binom{k-1}{j-1} \bar{\mu}'_{k-j} \bar{\theta}_j. \tag{84}$$

*Assume there exist constants* $C_\mathsf{g}, \psi_\eta > 0$ *and* $l, \gamma, n \in (0, \infty)$ *such that for every* $k \geq 1$

$$|\bar{\mu}'_k| \leq (2C_\mathsf{T})^k, \tag{85}$$

$$|\mu'_k - \bar{\mu}'_k| \leq r^{1/2}(\gamma n)^{-1/2}(2C_\mathsf{T})^k. \tag{86}$$

*Then, for all* $k \geq 1$,

$$|\bar{\theta}_k| \leq 2(2C_\mathsf{T})^k(k-1)!, \tag{87}$$

$$|\bar{\theta}_k - \theta_k| \leq 10k^{1/2}(\gamma n)^{-1/2}(2C_\mathsf{T})^k(k-1)!. \tag{88}$$

*Proof* : By Triangle inequality, (68) implies that

$$|\theta_k| \leq (2C_\mathsf{T})^k + \sum_{j=1}^{k-1} \binom{k-1}{j-1}(2C_\mathsf{T})^{k-j}|\theta_j|.$$

Let $\bar{\theta}_k = (2C_\mathsf{T})^k \rho_k$, then this becomes

$$|\rho_k| \leq 1 + \sum_{j=1}^{k-1} \binom{k-1}{j-1}|\rho_j|. \tag{89}$$

Since
$$|\rho_1| = \frac{|\bar{\mu}_1'|}{2C_\mathsf{T}} \le \frac{\epsilon_1}{2C_\mathsf{T}} \le \frac{1}{2},$$
it is easy to prove by induction that
$$|\rho_k| \le 2(k-1)!,$$
so that
$$|\bar{\theta}_k| \le 2(2C_\mathsf{T})^k(k-1)!.$$

We now turn to bound $\bar{\theta}_k - \theta_k$. Set $\Delta_k := \bar{\theta}_k - \theta_k$. Subtracting (68) from (69) gives
$$\Delta_k = (\bar{\mu}_k' - \mu_k') - \sum_{j=1}^{k-1} \binom{k-1}{j-1}\Big[(\bar{\mu}_{k-j}' - \mu_{k-j}')\bar{\theta}_j + \mu_{k-j}'\Delta_j\Big].$$

Let $D_k = r^{1/2}(\gamma n)^{-1/2}(2C_\mathsf{T})^k$. Taking absolute values and invoking the bounds already proved yields

$$
\begin{aligned}
|\Delta_k| &\le D_k + \sum_{j=1}^{k-1} \binom{k-1}{j-1}\Big(D_{k-j}|\bar{\theta}_j| + (|\bar{\mu}_{k-j}'| + D_{k-j})\,|\Delta_j|\Big) \\
&\le D_k + \sum_{j=1}^{k-1} \binom{k-1}{j-1}\Big[D_{k-j}\cdot 2(2C_\mathsf{T})^j(j-1)! + (|\bar{\mu}_{k-j}'| + D_{k-j})\,|\Delta_j|\Big] \\
&\le D_k + 2D_k\sum_{j=1}^{k-1}\frac{1}{(k-j)!} + \sum_{j=1}^{k-1}\binom{k-1}{j-1}(A_{k-j} + D_{k-j})|\Delta_j| \\
&\le 5D_k + 2\sum_{j=1}^{k-1}\binom{k-1}{j-1}A_{k-j}|\Delta_j|.
\end{aligned}
\tag{90}
$$

Let $\Delta_k = D_k\delta_k$, then (75) and (78) implies that
$$|\delta_k| \le 5 + 2\sum_{j=1}^{k-1}\binom{k-1}{j-1}|\delta_j|. \tag{91}$$

We also have that
$$|\delta_1| = \frac{|\mu_k - \mu_k'|}{4(C_\mathsf{g} + \psi_\eta)} \le \frac{1}{4}r^{1/2}(\gamma n)^{-1/2},$$
so by induction we can deduce that $|\delta_k| \le 10(k-1)!$, and
$$|\Delta_k| \le 10r^{1/2}(\gamma n)^{-1/2}(2C_\mathsf{T})^k(k-1)!,$$
concluding the proof. $\qquad\square$

In view of the previous lemma, we only need to bound the difference between $\bar{\theta}_l$ and $\kappa_l$. Note the following well-known property of cumulants $\{\kappa_l\}_{l=1}^{+\infty}$:
$$\log \mathbb{E}_{P_0}[e^{t\eta}] = \sum_{l=1}^{+\infty}\kappa_l\frac{t^l}{l!}.$$

Let $\hat{\eta} = T - \hat{g}(X)$ and $D_X = g_0(X) - \hat{g}(X)$, then $\hat{\eta} = \eta + D_X$ and $\eta, D_X$ are independent by our assumption. By definition, $\bar{\theta}_l$ is the $l$-th order cumulant of $T - \hat{g}(X)$. Hence,
$$\sum_{l=1}^{+\infty}\bar{\theta}_l\frac{t^l}{l!} = \log \mathbb{E}[e^{t\hat{\eta}}] = \log \mathbb{E}[e^{t\eta}] + \log \mathbb{E}[e^{tD_X}] = \sum_{l=1}^{+\infty}(\kappa_l + d_l)\frac{t^l}{l!},$$
where $d_l$ is the $l$-th order cumulant of the variable $D_X$, and we have that $d_l \le l!\mathbb{E}\Big[|g_0(X) - \hat{g}(X)|^l\Big] \le l!\epsilon^l$, and the conclusion follows.

### F.3 Relaxing the independent noise assumption

In this subsection, we consider a case where the noise $\eta$ is almost independent of $X$ and show that our estimator is still better than the plug-in estimator.

**Proposition F.1** (Finite–sample accuracy of two cubic–moment estimators). *Let $T = g_0(X) + \eta$ with $\eta = \epsilon_0\eta_0 + \eta_1$ and assume*

(i) $\mathbb{E}[\eta_i] = 0$, $\mathrm{Var}(\eta_i) = \sigma_i^2$ *for* $i \in \{0,1\}$;

(ii) $\eta_1 \perp\!\!\!\perp X$ *(while making no restriction on the joint law of $\eta_0$ and $X$);*

(iii) $\eta_0, \eta_1$ *have finite third moments;*

(iv) *an estimator $\hat{g}$ satisfies $\delta(X) := \hat{g}_0(X) - g_0(X)$ with $\|\delta\|_{L^3(P_X)} \le \epsilon$;*

(v) $\mathbb{E}[\,|T - \hat{g}_0(X)|^6\,] < \infty$ *and set $M := \mathbb{E}[\,|T - \hat{g}_0(X)|^6\,]^{1/6}$.*

*Given i.i.d. samples $\{(X_i, T_i)\}_{i=1}^n$, put $Z_i := T_i - \hat{g}_0(X_i)$ and define*

$$\hat{\mu}_{k,n} := \frac{1}{n}\sum_{i=1}^n Z_i^k \quad (k = 1, 2, 3), \qquad \hat{\psi}_n := \hat{\mu}_{3,n} - 3\hat{\mu}_{2,n}\hat{\mu}_{1,n}, \qquad \hat{\nu}_n := \hat{\mu}_{3,n}.$$

*For any $0 < \delta < 1$, with probability at least $1 - \delta$,*

$$|\hat{\psi}_n - \mathbb{E}[\eta^3]| \le \underbrace{6\sigma_0^2\epsilon_0^2\epsilon + 3\sigma_0\epsilon_0\epsilon^2 + 4\epsilon^3}_{bias} + C\,M^3\sqrt{\frac{\log(6/\delta)}{n}}, \tag{92}$$

$$|\hat{\nu}_n - \mathbb{E}[\eta^3]| \le \underbrace{3\sigma_1^2\epsilon + 3\sigma_0^2\epsilon_0^2\epsilon + 3\sigma_0\epsilon_0\epsilon^2 + \epsilon^3}_{bias} + C\,M^3\sqrt{\frac{\log(6/\delta)}{n}}, \tag{93}$$

*where $C > 0$ is an absolute constant (e.g. $C = 10$). Consequently, if $\epsilon_0 \to 0$ and $\epsilon \to 0$ while $n \to \infty$, the plug–in estimator $\hat{\psi}_n$ is $o(\epsilon)$–biased and $\mathcal{O}_p(n^{-1/2})$–consistent, whereas the naive estimator $\hat{\nu}_n$ keeps a leading $\Theta(\epsilon)$ bias whenever $\sigma_1^2 > 0$.*

**Proof :** With $Z = T - \hat{g}_0(X) = \eta - \delta$ and $\mu_k := \mathbb{E}[Z^k]$, we have

$$\hat{\psi}_n - \mathbb{E}[\eta^3] = (\mu_3 - 3\mu_2\mu_1 - \mathbb{E}[\eta^3]) + \big[(\hat{\mu}_{3,n} - \mu_3) - 3\mu_2(\hat{\mu}_{1,n} - \mu_1) - 3(\hat{\mu}_{2,n} - \mu_2)\hat{\mu}_{1,n}\big], \tag{94}$$

$$\hat{\nu}_n - \mathbb{E}[\eta^3] = (\mu_3 - \mathbb{E}[\eta^3]) + (\hat{\mu}_{3,n} - \mu_3). \tag{95}$$

The first bracket in each line is the *bias*, the second the *sampling error*.

Now write $\eta = \epsilon_0\eta_0 + \eta_1$, $\delta = \delta(X)$. Expanding moments and using $\eta_1 \perp\!\!\!\perp X$,

$$\mu_1 = -\mathbb{E}[\delta], \quad \mu_2 = \sigma_1^2 + \epsilon_0^2\sigma_0^2 + \mathbb{E}[\delta^2], \quad \mu_3 = \mathbb{E}[\eta^3] - 3\sigma_1^2\mathbb{E}[\delta] - 3\epsilon_0^2\sigma_0^2\mathbb{E}[\delta] - \mathbb{E}[\delta^3] + 3\epsilon_0\mathbb{E}[\eta_0\delta^2].$$

Hölder yields $|\mathbb{E}[\delta]| \le \epsilon$, $\mathbb{E}[\delta^2] \le \epsilon^2$, $|\mathbb{E}[\delta^3]| \le \epsilon^3$, while Cauchy–Schwarz gives $|\mathbb{E}[\eta_0\delta^2]| \le \sigma_0\epsilon^2$. Substituting into (94)–(95) gives the bias terms displayed in (92)–(93).

Define $\Delta_k := \hat{\mu}_{k,n} - \mu_k$. By Bernstein's inequality for centred variables with sixth moment $M^6$,

$$\Pr\Big(|\Delta_k| > t\Big) \le 2\exp\Big(-\tfrac{nt^2}{2\mathbb{E}[Z^{2k}] + 2Mt/3}\Big) \quad (k = 1, 2, 3).$$

Taking $t_k := M^k\sqrt{(2\log(6/\delta))/n}$ and a union bound over $k$ ensures $|\Delta_k| \le t_k$ with probability $1 - \delta$. Using $|\mu_1| \le M$, $|\mu_2| \le M^2$ and (94),

$$|\hat{\psi}_n - \mu_3 + 3\mu_2\mu_1| \le |\Delta_3| + 3M^2|\Delta_1| + 3M|\Delta_2| + 3|\Delta_2||\Delta_1| \le 10M^3\sqrt{\frac{\log(6/\delta)}{n}}.$$

An analogous bound holds for $|\hat{\nu}_n - \mu_3|$. Combining with the bias bounds completes (92)–(93). $\square$

# G  Technical details in Section 5.2

In this section, we provide detailed proofs of results in Section 5.2. We let $\mathrm{P}_m$ be the set of all possible partitions of the integer $m$, i.e., the set of all multisets of positive integers that sum to $m$ (*e.g.*, for $m = 4$ the possible partitions are $(4), (1, 3), (2, 2), (1, 1, 2), (1, 1, 1, 1)$), and $\mathrm{P}_{m,j}$ be the set of partitions with $j$ terms. We define $p(m) = |\mathrm{P}_m|$ and $p(m, j) = |\mathrm{P}_{m,j}|$ respectively. Note that $\mathrm{P}_m$ is different from $\Pi_m$ defined in Section 5, which is the number of partitions of $[m]$ into distinct subsets.

**Proposition G.1** (Partition number bound). *We have $p(m) \leq 2^m$ for all $m \geq 1$.*

***Proof*** : Consider placing numbers $1, 2, \cdots, m$ in a row and delimiters are placed between some consecutive numbers. Clearly, the total number of ways to place the delimiters is $2^{m-1}$. For each partition $m = i_1 + \cdots + i_j$, there exists at least one way of placing the delimiters, such that their induced partition of $\{1, 2, \cdots, m\}$ contains subsets of sizes $i_1, \cdots, i_m$. This creates an injective mapping from $\mathrm{P}_m$ to the set of possible delimiters, implying that $p(m) \leq 2^{m-1}$. $\qquad\square$

## G.1  Proof of Lemma 5.1: Key lemma; higher-order insensitivity condition

First by definition, $\mathbb{E}\left[\hat{J}_r^{(k)}(T - g_0(X))\right] = \sum_{i=k}^{r} \frac{i!}{(i-k)!} \hat{a}_{i+1,r} \mu_{i-k}$ where $\mu_{i-k} = \sum_{\pi' \in \Pi_{i-k}} \prod_{B' \in \pi'} \kappa_{|B'|}$ is the $(i-k)$-th moment of $\eta$. It suffices to show that

$$\sum_{\pi \in \Pi_{r-k}} \prod_{B \in \pi} \left(\kappa_{|B|} - \hat{\kappa}_{|B|}\right) = \sum_{i=k}^{r} \binom{r-k}{r-i} \sum_{\pi \in \Pi_{r-i}} (-1)^{|\pi|} \prod_{B \in \pi} \hat{\kappa}_{|B|} \cdot \sum_{\pi' \in \Pi_{i-k}} \prod_{B' \in \pi'} \kappa_{|B'|}. \quad (96)$$

To establish this, we note that the corresponding summands on each side are of the form $(-1)^q \kappa_{i_1} \cdots \kappa_{i_p} \hat{\kappa}_{j_1} \cdots \hat{\kappa}_{j_q}$. Fix $i_1, \cdots, i_p, j_1, \cdots, j_q$ such that $\sum_{s=1}^{p} i_s + \sum_{t=1}^{q} j_t = r - k$, and let $i = k + \sum_{s=1}^{p} i_s \leq j$. Let $N_{m, \{\alpha_s\}_{s=1}^{s_0}}$ be the number of ways to partition $[m]$ into subsets of sizes $\{\alpha_s\}_{s=1}^{s_0}$ where $m = \sum_{s=1}^{s_0} \alpha_s$. Then the coefficient of the term $(-1)^q \kappa_{i_1} \cdots \kappa_{i_p} \hat{\kappa}_{j_1} \cdots \hat{\kappa}_{j_q}$ on the right-hand side (RHS) is $\binom{r-k}{r-i} N_{i-k, \{i_s\}_{s=1}^{p}} N_{r-i, \{j_t\}_{t=1}^{q}}$. However, the left-hand side has precisely the same coefficient, because there exist $\binom{r-k}{r-i}$ ways to partition $[r-k]$ into two subsets with sizes $r-i$ and $i-k$ respectively and inside each subset the number of partitions with desired subset sizes are $N_{i-k, \{i_s\}_{s=1}^{p}}$ and $N_{r-i, \{j_t\}_{t=1}^{q}}$ respectively.

## G.2  Proof of Theorem 5.4: ACE estimation error: sub-Gaussian noise

**Lemma G.1** (Log inequalities). *Let $a, b > 0$.*

    *(1) For every $0 < x \leq \dfrac{b}{a} - \dfrac{1}{a} \log \dfrac{b}{a}$ we have $ax + \log x \leq b$.*

    *(2) Assume, in addition, that $ab \geq e$ (so $\log(ab) \geq 1$). Then for every $0 < x \leq \dfrac{b}{a \log(ab)}$ we have $x \log(ax) \leq b$.*

***Proof*** : **Proof of Item (1).** Define $g(x) \triangleq ax + \log x - b$, $x > 0$. Because $g'(x) = a + 1/x > 0$, the map $g$ is strictly increasing. Put

$$x_0 \triangleq \frac{b}{a} - \frac{1}{a} \log \frac{b}{a} \quad (> 0).$$

We show $g(x_0) \leq 0$. Set $y \triangleq \dfrac{b}{a}$; then

$$g(x_0) = a\left(y - \tfrac{1}{a} \log y\right) + \log\left(y - \tfrac{1}{a} \log y\right) - b = -\log y + \log\left(y - \tfrac{1}{a} \log y\right).$$

Since $y - \tfrac{1}{a} \log y < y$, the argument of the second logarithm is smaller than $y$, hence $\log\left(y - \tfrac{1}{a} \log y\right) < \log y$ and $g(x_0) < 0$. Because $g$ is increasing, $x \leq x_0$ implies $g(x) \leq g(x_0) \leq 0$, i.e. $ax + \log x \leq b$.

**Proof of Item (2).** Define $h(x) \triangleq x \log(ax) - b, \ x > 0$. We have $h'(x) = \log(ax) + 1$, so $h$ is strictly increasing for $x \geq e^{-1}/a$. The equation $h(x) = 0$ has the unique positive root

$$x^\star = \frac{b}{a\,W(ab)},$$

where $W$ is the Lambert-$W$ function (the solution of $z = W(z)e^{W(z)}$). When $ab \geq e$ we have $W(ab) \geq \log(ab)$ (standard lower bound for $W$ on $[e, \infty)$), hence

$$x^\star = \frac{b}{a\,W(ab)} \ \geq \ \frac{b}{a \log(ab)}.$$

Thus every $x \leq b/(a \log(ab))$ satisfies $x \leq x^\star$ and, by monotonicity of $h$, $h(x) \leq h(x^\star) = 0$; that is $x \log(ax) \leq b$. $\qquad\square$

**Lemma G.2** (Condition for bias domination). *Let*

$$\Delta_i \triangleq \left|\kappa_i - \hat{\kappa}_i\right| \ \leq \ \underbrace{3\big[12i(C_{\mathsf{g}} + \psi_{\mathfrak{n}})\big]^i r^{1/2}(\gamma n)^{-1/2}}_{variance} + \underbrace{(2i\epsilon_1)^i}_{bias}, \qquad 1 \leq i \leq r.$$

*Let $a \triangleq 2\log\left(6(C_{\mathsf{g}} + \psi_{\mathfrak{n}})\epsilon_1^{-1}\right), b \triangleq \log(\gamma n/9)$ with $a, b > 0$. If*

$$l \ \leq \ \frac{b}{a} - \frac{1}{a}\log\frac{b}{a}, \tag{97}$$

*then for every $1 \leq i \leq r$*

$$3\big[12i(C_{\mathsf{g}} + \psi_{\mathfrak{n}})\big]^i r^{1/2}(\gamma n)^{-1/2} \ \leq \ (2i\epsilon_1)^i,$$

*i.e. the bias term $(2i\epsilon_1)^i$ dominates the variance term in Lemma G.2.*

**Proof :** Note that

$$\frac{variance}{bias} = 3\big[6(C_{\mathsf{g}} + \psi_{\mathfrak{n}})\epsilon_1^{-1}\big]^i r^{1/2}(\gamma n)^{-1/2} \leq 1, \quad \forall i \in [r]$$

is equivalent to

$$2\log\left(6(C_{\mathsf{g}} + \psi_{\mathfrak{n}})\epsilon_1^{-1}\right)l + \log l \leq \log(\gamma n/9).$$

We can apply Lemma G.1 with $a = 2\log\left(6(C_{\mathsf{g}} + \psi_{\mathfrak{n}})\epsilon_1^{-1}\right)$ and $b = \log(\gamma n/9)$ to obtain the desired conclusion. $\qquad\square$

Let

$$\begin{aligned}
J_r(w) &= \sum_{i=1}^{r+1} a_{ir} w^{i-1}, \\
a_{1k} &= 1, a_{ir} = \frac{1}{(i-1)!(r+1-i)!} \sum_{\pi \in \Pi_{r+1-i}} (-1)^{|\pi|} \prod_{B \in \pi} \kappa_{|B|}, 2 \leq i \leq r+1.
\end{aligned} \tag{98}$$

be the exact version of $\hat{J}_r$. We first derive some bounds for the coefficients of $J_r$ and $\hat{J}_r$.

**Lemma G.3** (Bounding polynomial coefficients). *For any $i \in [r+1]$ we have*

$$|a_{ir}| \leq \frac{1}{(i-1)!}(8\psi_{\mathfrak{n}})^{r+1-i}. \tag{99}$$

**Proof** : By definition, we have

$$|a_{ir}| \leq \frac{1}{(i-1)!(r+1-i)!} \sum_{\pi \in \Pi_{r+1-i}} \prod_{B \in \pi} \kappa_{|B|} \tag{100}$$

$$\leq \frac{1}{(i-1)!(r+1-i)!} \sum_{\pi \in \Pi_{r+1-i}} \prod_{B \in \pi} 2^{2|B|} |B|^{|B|/2} \psi_\eta^{|B|} \tag{101}$$

$$= \frac{1}{(i-1)!(r+1-i)!} (4\psi_\eta)^{r+1-i} \sum_{\pi \in \Pi_{r+1-i}} |B|^{|B|/2} \tag{102}$$

$$= \frac{1}{(i-1)!(r+1-i)!} (4\psi_\eta)^{r+1-i} \sum_{j=1}^{r+1-i} \sum_{(i_1,\cdots,i_j) \in \mathrm{P}_{r+1-i,j}} \binom{r+1-i}{i_1,\cdots,i_j} \prod_{s=1}^{j} i_s^{i_s/2} \tag{103}$$

$$\leq \frac{1}{(i-1)!} (4\psi_\eta)^{r+1-i} \sum_{j=1}^{r+1-i} p(r+1-i,j) \tag{104}$$

$$\leq \frac{1}{(i-1)!} (8\psi_\eta)^{r+1-i}, \tag{105}$$

where (100) follows from triangle inequality, (101) follows from the cumulant bound in Proposition A.2, (103) rearranges the summation term according to the number of subsets in the partition $\pi \in \Pi_{r+1-i}$, (104) follows from

$$\binom{r+1-i}{i_1,\cdots,i_j} \prod_{s=1}^{j} i_s^{i_s} = (r+1-i)! \prod_{s=1}^{j} \frac{i_s^{i_s/2}}{i_s!} \leq (r+1-i)!,$$

and (105) follows from Proposition G.1. □

**Lemma G.4** (Bounding the estimation error of polynomial coefficients)**.** *For any* $i \in [r+1]$ *we have*

$$|a_{ir} - \hat{a}_{ir}| \leq \frac{1}{(i-1)!} (16\psi_\eta)^{r+1-i} \epsilon_1. \tag{106}$$

**Proof** : By definition, we have

$$|a_{ir} - \hat{a}_{ir}| = \frac{1}{(i-1)!(r+1-i)!} \left| \sum_{\pi \in \Pi_{r+1-i}} (-1)^{|\pi|-1} \left( \prod_{B \in \pi} \kappa_{|B|} - \prod_{B \in \pi} \hat{\kappa}_{|B|} \right) \right|$$

$$\leq \frac{1}{(i-1)!(r+1-i)!} \sum_{\pi \in \Pi_{r+1-i}} \left| \left( \prod_{B \in \pi} \kappa_{|B|} - \prod_{B \in \pi} \hat{\kappa}_{|B|} \right) \right|$$

$$\leq \frac{1}{(i-1)!(r+1-i)!} \sum_{\pi \in \Pi_{r+1-i}} \sum_{B \in \pi} |\kappa_{|B|} - \hat{\kappa}_{|B|}| \cdot \prod_{B' \in \pi \setminus \{B\}} \max\left\{ |\kappa_{|B'|}|, |\hat{\kappa}_{|B'|}| \right\}$$

$$\leq \frac{1}{(i-1)!(r+1-i)!} \sum_{\pi \in \Pi_{r+1-i}} \sum_{B \in \pi} \left( 2|B|\epsilon_1 \right)^{|B|} \prod_{B' \in \pi \setminus \{B\}} 2^{2|B'|+1} |B'|^{|B'|/2} \psi_\eta^{|B'|}$$

$$\leq \frac{1}{(i-1)!(r+1-i)!} (8\psi_\eta)^{r+1-i} \epsilon_1 \sum_{j=1}^{r+1-i} \sum_{(i_1,\cdots,i_j) \in \mathrm{P}_{r+1-i,j}} \binom{r+1-i}{i_1,\cdots,i_j} \prod_{s=1}^{j} i_s^{i_s/2}$$

$$\leq \frac{1}{(i-1)!} (8\psi_\eta)^{r+1-i} \epsilon_1 \sum_{j=1}^{r+1-i} p(r+1-i,j)$$

$$\leq \frac{1}{(i-1)!} (16\psi_\eta)^{r+1-i} \epsilon_1,$$

as desired. Here in the last but one step, we use the fact that

$$\binom{r+1-i}{i_1,\cdots,i_j} \prod_{s=1}^{j} i_s^{i_s} = (r+1-i)! \prod_{s=1}^{j} \frac{i_s^{i_s/2}}{i_s!} \leq (r+1-i)!,$$

and the last step follows from Proposition G.1. □

**Lemma G.5** (Key lemma; approximate orthogonality). *Let*

$$r \leq \frac{b}{a} - \frac{1}{a} \log \frac{b}{a}, \tag{107}$$

*where $a \triangleq 2 \log \left( 6(C_{\mathbf{g}} + \psi_{\mathfrak{n}})\epsilon_1^{-1} \right), b \triangleq \log(\gamma n/9)$ as in Lemma G.2, then under $\mathcal{E}$ we have*

$$\mathbb{E}\left[ \hat{J}_r^{(k)}(T - g_0(X)) \right] \leq (4e\epsilon_1)^{r-k}.$$

***Proof*** **:** For any $k \in [r]$ we have that

$$
\begin{aligned}
\mathbb{E}\left[ \hat{J}_r^{(k)}(T - g_0(X)) \right] &= \sum_{i=k}^{r} \frac{i!}{(i-k)!} \hat{a}_{i+1,r} \mu_{i-k} \\
&= \sum_{i=k}^{r} \frac{1}{(r-i)!(i-k)!} \sum_{\pi \in \Pi_{r-i}} (-1)^{|\pi|-1} \prod_{B \in \pi} \hat{\kappa}_{|B|} \cdot \sum_{\pi' \in \Pi_{i-k}} \prod_{B' \in \pi'} \kappa_{|B'|} \\
&= \frac{1}{(r-k)!} \sum_{\pi \in \Pi_{r-k}} \prod_{B \in \pi} \left( \kappa_{|B|} - \hat{\kappa}_{|B|} \right).
\end{aligned}
\tag{108}
$$

By Lemma G.2, we have $|\kappa_i - \hat{\kappa}_i| \leq 2(2i\epsilon_1)^i$ for all $i \in [r]$. Then for any $1 \leq j \leq r - k$, we have

$$
\begin{aligned}
\sum_{\pi \in \Pi_{r-k}, |\pi|=j} \left| \prod_{B \in \pi} \left( \kappa_{|B|} - \hat{\kappa}_{|B|} \right) \right| &\leq \sum_{(i_1, \cdots, i_j) \in \mathrm{P}_{r+1-k,j}} \binom{r-k}{i_1, i_2, \cdots, i_j} \prod_{s=1}^{j} 2(i_s \epsilon_1)^{i_s} \\
&\leq 2^{r-k}(r-k)!\epsilon_1^{r-k} \sum_{(i_1, \cdots, i_j) \in \mathrm{P}_{r-k,j}} \prod_{s=1}^{j} \frac{i_s^{i_s}}{i_s!} \\
&\leq (r-k)!\, p(r-k, j)(2e\epsilon_1)^{r-k}.
\end{aligned}
$$

Plugging into (108), we have

$$\mathbb{E}\left[ \hat{J}_r^{(k)}(T - g_0(X)) \right] \leq p(l-k)(2e\epsilon_1)^{r-k} \leq (4e\epsilon_1)^{r-k}.$$

$\square$

**Lemma G.6** (Identifiability coefficient). $\mathbb{E}[(T - g_0(X))J_r(T - g_0(X))] = \frac{1}{r!}\kappa_{r+1}$.

***Proof*** **:** By definition, we have

$$
\begin{aligned}
&\mathbb{E}[(T - g_0(X))J_r(T - g_0(X))] \\
&= \sum_{i=1}^{r+1} \frac{1}{(i-1)!(r+1-i)!} \mu_i \sum_{\pi \in \Pi_{r+1-i}} (-1)^{|\pi|} \prod_{B \in \pi} \kappa_{|B|} \\
&= \sum_{i=1}^{r+1} \frac{1}{(i-1)!(r+1-i)!} \sum_{\pi' \in \Pi_i} \prod_{B' \in \pi'} \kappa_{|B'|} \sum_{\pi \in \Pi_{r+1-i}} (-1)^{|\pi|} \prod_{B \in \pi} \kappa_{|B|}.
\end{aligned}
\tag{109}
$$

Consider any partition $\pi \in \Pi_i$. Without loss of generality, assume that $1 \in B_1$ and $|B_1| = k, k \leq i$. There are a total of $\binom{i-1}{k-1}$ possible choices of $B_1$, and the remaining sets form a partition of $[i-k]$. Hence we can write

$$\sum_{\pi' \in \Pi_i} \prod_{B' \in \pi'} \kappa_{|B'|} = \sum_{k=1}^{i} \binom{i-1}{k-1} \sum_{\pi'' \in \Pi_{i-r}} \sum_{B'' \in \pi''} \kappa_{|B''|}.$$

Plugging into (109), the right-hand side

$$
\begin{aligned}
&= \frac{1}{r!} \sum_{k=1}^{r+1} \binom{r}{k-1} \kappa_k \sum_{i=r}^{r+1} \binom{r-k+1}{i-r} \sum_{\pi'' \in \Pi_{i-r}} \sum_{B'' \in \pi''} \kappa_{|B''|} \sum_{\pi \in \Pi_{r+1-i}} (-1)^{|\pi|} \prod_{B \in \pi} \kappa_{|B|} \\
&= \frac{1}{r!}\left[ \kappa_{r+1} + \sum_{k=1}^{r+1} \binom{r}{k-1} \kappa_k \sum_{\pi \in \Pi_{r+1-k}} \prod_{B \in \pi} \left( \kappa_{|B|} - \kappa_{|B|} \right) \right] = \frac{1}{r!}\kappa_{r+1},
\end{aligned}
$$

concluding the proof.

$\square$

**Lemma G.7** (Linear–in–$\varepsilon_1$ moment difference). *Let*

$$T = g_0(X) + \eta, \qquad \mathbb{E}[\eta] = 0, \qquad \eta \perp\!\!\!\perp X, \qquad \|\eta\|_{\psi_2} =: \psi_\eta < \infty.$$

*Assume an estimate $\hat{g}$ satisfies*

$$\|\hat{g} - g_0\|_{L^s(P)} \leq \epsilon_1 \leq \psi_\eta, \quad \text{for some } s \geq 2,$$

*and fix an integer $1 \leq i \leq s/2$. Then*

$$\mathbb{E}\big|(T - g_0(X))^i - (T - \hat{g}(X))^i\big| \;\leq\; i(3\psi_\eta\sqrt{i})^i\epsilon_1. \tag{110}$$

***Proof :*** Write $\delta(X) \triangleq g_0(X) - \hat{g}(X)$ so that $T - g_0(X) = \eta$ and $T - \hat{g}(X) = \eta + \delta(X)$. Binomial expansion gives

$$(\eta)^i - (\eta + \delta)^i \;=\; -\sum_{j=0}^{i-1} \binom{i}{j} \eta^j \, \delta^{i-j}.$$

Taking absolute values, expectations, and using independence ($\eta \perp\!\!\!\perp X$) together with Hölder,

$$\mathbb{E}\big|(\eta)^i - (\eta + \delta)^i\big| \;\leq\; \sum_{j=0}^{i-1} \binom{i}{j} \mathbb{E}|\eta|^j \, \mathbb{E}|\delta|^{\,i-j} \;\leq\; \sum_{j=0}^{i-1} \binom{i}{j} \mathbb{E}|\eta|^j \, \varepsilon_1^{\,i-j}. \tag{A}$$

Since $\eta$ is sub-Gaussian with $\|\eta\|_{\psi_2} = \psi_\eta$, we have $\mathbb{E}|\eta|^j \leq (2\psi_\eta\sqrt{j})^j \leq (2\psi_\eta\sqrt{i-1})^j$, so that

$$\mathbb{E}\big|(\eta)^i - (\eta + \delta)^i\big| \leq \epsilon_1 \sum_{j=0}^{i-1} \binom{i}{j}(2\psi_\eta\sqrt{i-1})^j\epsilon_1^{i-1-j} \leq i\epsilon_1 \sum_{j=0}^{i-1} \binom{i-1}{j}(2\psi_\eta\sqrt{i-1})^j\epsilon_1^{i-1-j}$$

$$\leq i\epsilon_1(2\psi_\eta\sqrt{i-1} + \epsilon_1)^{i-1} \leq i(3\psi_\eta\sqrt{i})^i\epsilon_1.$$

$\square$

**Lemma G.8** (Identifiability guarantee). *We have*

$$\left| \mathbb{E}_P\left[(T - \hat{g}(X))\hat{J}_r(T - \hat{g}(X))\right] \right| \geq \frac{1}{2r!}\delta_{\mathsf{id}}.$$

***Proof :*** By assumption (2) in Theorem 5.4 and Lemma G.6, we know that

$$\left| \mathbb{E}_P\left[(T - g_0(X))J_r(T - g_0(X))\right] \right| \geq \frac{1}{r!}\delta_{\mathsf{id}},$$

By Lemma G.3, we have $|a_{ir}| \leq \frac{1}{(i-1)!}(8\psi_\eta)^{r+1-i}$. Moreover, by Lemma G.7, for $i \in [r+1]$ it holds that

$$\mathbb{E}\left|(T - g_0(X))^i - (T - \hat{g}(X))^i\right| \leq i(3\psi_\eta\sqrt{i})^i\epsilon_1.$$

Hence, we have

$$\left| \mathbb{E}_P\left[(T - g_0(X))J_r(T - g_0(X))\right] - \mathbb{E}_P\left[(T - \hat{g}(X))\hat{J}_r(T - \hat{g}(X))\right] \right|$$

$$= \left| \sum_{i=1}^{r+1} \big(a_{ir}\mathbb{E}\big[(T - g_0(X))^i\big] - \hat{a}_{ir}\mathbb{E}\big[(T - \hat{g}(X))^i\big]\big) \right|$$

$$\leq \sum_{i=1}^{r+1} |a_{ir}| \cdot \mathbb{E}\big|(T - g_0(X))^i - (T - \hat{g}(X))^i\big| + |a_{ir} - \hat{a}_{ir}| \cdot \big|\mathbb{E}\big[(T - \hat{g}(X))^i\big]\big| \tag{111}$$

$$\leq \sum_{i=1}^{r+1} \frac{1}{(i-1)!}\big[(8\psi_\eta)^{r+1-i} \cdot i(3\psi_\eta\sqrt{i})^i\epsilon_1 + (8\psi_\eta)^{r+1-i}\epsilon_1 \cdot 2^{2i}(C_\mathsf{g} + \psi_\eta)^i i^{i/2}\big] \tag{112}$$

$$\leq \big[8(C_\mathsf{g} + \psi_\eta)\big]^r\epsilon_1 \sum_{i=1}^{r+1} \frac{1}{(i-1)!}i^{1+i/2}$$

$$\leq 100\big[8(C_\mathsf{g} + \psi_\eta)\big]^r\epsilon_1,$$

where (111) follows from triangle inequality, (112) follows from Lemma G.3, Lemma G.4, Lemma G.7 and (66). From our assumption (21) and Lemma G.1, we can deduce that this quantity is smaller than $\frac{1}{2r!}\delta_{\mathsf{id}}$, concluding the proof. $\square$

**Remark G.1.** *With a similar reasoning, one can also deduce that*

$$\left| \mathbb{E}_P \left[ (T - g_0(X)) J_r(T - g_0(X)) \right] - \mathbb{E}_P \left[ (T - g_0(X)) \hat{J}_r(T - \hat{g}(X)) \right] \right| \leq 3 \left[ 8(C_{\mathsf{g}} + \psi_{\mathfrak{n}}) \right]^r \epsilon_1. \tag{113}$$

*This inequality will be used later in the proof.*

**Lemma G.9** (Second-order moment bounds). *The following inequalities hold:*

$$\mathbb{E}_P \left[ (T - \hat{g}(X))^2 \hat{J}_r(T - \hat{g}(X))^2 \right] \leq 2r^2 \left[ 8(C_{\mathsf{g}} + \psi_{\mathfrak{n}}) \right]^{2(r+1)} \tag{114}$$

$$\mathbb{E}_P \left[ (Y - \hat{q}(X))^2 \hat{J}_r(T - \hat{g}(X))^2 \right] \leq 112(\psi_{\xi} + C_{\theta}\psi_{\mathfrak{n}})^2 \left[ 16(C_{\mathsf{g}} + \psi_{\mathfrak{n}}) \right]^{2r} \tag{115}$$

***Proof*** : Recall that

$$\hat{J}_r(w) = \sum_{i=1}^{r+1} a_{ir} w^{i-1}, \quad |a_{ir}| \leq \frac{1}{(i-1)!} (8\psi_{\mathfrak{n}})^{r+1-i} \tag{116}$$

by [Lemma G.3](), and

$$\mathbb{E} \left[ (T - \hat{g}(X))^r \right] \leq 2^{2r} (C_{\mathsf{g}} + \psi_{\mathfrak{n}})^r k^{k/2} \tag{117}$$

by [(66)](), we can deduce that

$$\mathbb{E}_P \left[ (T - \hat{g}(X))^2 \hat{J}_r(T - \hat{g}(X))^2 \right]$$

$$\leq \mathbb{E}_P \left[ \left( \sum_{i=1}^{r+1} a_{ir}(T - \hat{g}(X))^i \right)^2 \right]$$

$$\leq (8\psi_{\mathfrak{n}})^{2(r+1)} \mathbb{E}_P \left[ \left( \sum_{i=1}^{r+1} \frac{1}{(i-1)!} \left( \frac{T - \hat{g}(X)}{8\psi_{\mathfrak{n}}} \right)^i \right)^2 \right] \tag{118}$$

$$= (8\psi_{\mathfrak{n}})^{2(r+1)} \sum_{1 \leq i,j \leq l+1} \frac{1}{(i-1)!(j-1)!} \mathbb{E} \left[ ((T - \hat{g}(X))/(8\psi_{\mathfrak{n}}))^{i+j} \right]$$

$$\leq (8\psi_{\mathfrak{n}})^{2(r+1)} \sum_{1 \leq i,j \leq l+1} \frac{1}{(i-1)!(j-1)!} 2^{2(i+j)} \left( 1 + \frac{C_{\mathsf{g}}}{\psi_{\mathfrak{n}}} \right)^{i+j} (i+j)^{(i+j)/2} \tag{119}$$

$$\leq r^2 (8\psi_{\mathfrak{n}})^{2(r+1)} \sum_{1 \leq i,j \leq l+1} \frac{1}{i!j!} 2^{3(i+j)} \left( 1 + \frac{C_{\mathsf{g}}}{\psi_{\mathfrak{n}}} \right)^{i+j} i^{i/2} j^{j/2} \tag{120}$$

$$\leq r^2 (8\psi_{\mathfrak{n}})^{2(r+1)} \sum_{1 \leq i,j \leq l+1} 2^{3(i+j)} \left( 1 + \frac{C_{\mathsf{g}}}{\psi_{\mathfrak{n}}} \right)^{i+j} \tag{121}$$

$$= r^2 (8\psi_{\mathfrak{n}})^{2(r+1)} \left[ \sum_{i=1}^{r+1} 8^i \left( 1 + \frac{C_{\mathsf{g}}}{\psi_{\mathfrak{n}}} \right)^i \right]^2 \tag{122}$$

$$\leq 2r^2 \left[ 8(C_{\mathsf{g}} + \psi_{\mathfrak{n}}) \right]^{2(r+1)}, \tag{123}$$

where [(118)]() follows from [(116)](), [(119)]() follows from [(117)](), [(120)]() follows from $\left( \frac{i+j}{2} \right)^{i+j} \leq i^i j^j$ which is a direct consequence of Jensen's inequality, [(103)]() follows from $i! \geq i^{i/2}$. This concludes the proof of [(114)]().

With a similar reasoning, we can deduce that

$$\mathbb{E}_P\left[\hat{J}_r(T-\hat{g}(X))^4\right]$$

$$\leq (8\psi_\eta)^{4r}\mathbb{E}_P\left[\left(\sum_{i=1}^{r+1}\frac{1}{(i-1)!}\left(\frac{T-\hat{g}(X)}{8\psi_\eta}\right)^{i-1}\right)^4\right]$$

$$\leq (8\psi_\eta)^{4r}\sum_{0\leq i,j,u,v\leq l}\frac{1}{i!j!u!v!}\mathbb{E}\left[\left((T-\hat{g}(X))/(8\psi_\eta)\right)^{i+j+u+v}\right]$$

$$\leq (8\psi_\eta)^{4r}\sum_{0\leq i,j,u,v\leq l}\frac{1}{i!j!u!v!}2^{2(i+j+u+v)}\left(1+\frac{C_{\mathsf{g}}}{\psi_\eta}\right)^{i+j+u+v}(i+j+u+v)^{(i+j+u+v)/2}$$

$$\leq (8\psi_\eta)^{4r}\sum_{0\leq i,j,u,v\leq l}\frac{1}{i!j!u!v!}2^{4(i+j+u+v)}\left(1+\frac{C_{\mathsf{g}}}{\psi_\eta}\right)^{i+j+u+v}i^{i/2}j^{j/2}u^{u/2}v^{v/2}$$

$$\leq (8\psi_\eta)^{4r}\left[\sum_{i=0}^{r}16^i\left(1+\frac{C_{\mathsf{g}}}{\psi_\eta}\right)^i\right]^4$$

$$\leq 2\left[16(C_{\mathsf{g}}+\psi_\eta)\right]^{4r}.$$

Since $\xi$ is $\psi_\xi$-sub-Gaussian, we have

$$\begin{aligned}
\mathbb{E}_P\left[(Y-\hat{q}(X))^4\right] &= \mathbb{E}_P\left[(\xi+\theta\eta+q(X)-\hat{q}(X))^4\right]\\
&\leq 22\left(\mathbb{E}_P[\xi^4]+C_\theta^4\mathbb{E}_P[\eta^4]+\mathbb{E}_P\left[(q(X)-\hat{q}(X))^4\right]\right)\\
&\leq 22\left[4^4(\psi_\xi^4+C_\theta^4\psi_\eta^4)+\epsilon_2^4\right],
\end{aligned}$$

so that

$$\begin{aligned}
\mathbb{E}_P\left[(Y-\hat{q}(X))^2\hat{J}_r(T-\hat{g}(X))^2\right] &\leq \mathbb{E}_P\left[(Y-\hat{q}(X))^4\right]^{1/2}\cdot\mathbb{E}_P\left[\hat{J}_r(T-\hat{g}(X))^4\right]^{1/2}\\
&\leq 112(\psi_\xi+C_\theta\psi_\eta)\left[16(C_{\mathsf{g}}+\psi_\eta)\right]^{2r}.
\end{aligned}$$

$\square$

By definition, we have for any $P\in\mathcal{P}$ that

$$\begin{aligned}
D_q D_g^r\mathbb{E}_P[m(Z,\theta,h_0(X))] &= (-1)^{r+1}\mathbb{E}_P\left[\hat{J}_r^{(k)}(T-g_0(X))\right]\\
D_g^{r+1}\mathbb{E}_P[m(Z,\theta,h_0(X))] &= (-1)^r\theta\mathbb{E}_P\left[\hat{J}_r^{(k)}(T-g_0(X))\right].
\end{aligned}$$

Since $m(Z, \theta, h(X))$ with $h = (g, q)$ can be viewed as an $(l + 2)$-th order polynomial of $g$ and $q$, we have

$$
\begin{aligned}
&\left| \mathbb{E}_P \big[ m(Z, \theta_0, \hat{h}(X)) \big] \right| \\
&= \left| \mathbb{E}_P \big[ m(Z, \theta_0, \hat{h}(X)) \big] - \mathbb{E}_P \big[ m(Z, \theta_0, h_0(X)) \big] \right| \\
&= \left| \mathbb{E}_P \left[ \sum_{k=1}^{r+1} \frac{1}{k!} \Big( k D_q D_g^{k-1} m(Z, \theta_0, h_0(X)) \cdot (\hat{q}(X) - q_0(X))(\hat{g}(X) - g_0(X))^{k-1} \right.\right. \\
&\qquad\qquad \left.\left. + \ D_g^r m(Z, \theta_0, h_0(X)) \cdot (\hat{g}(X) - g_0(X))^r \Big) \right] \right| \\
&\le \sum_{k=1}^{r+1} \frac{1}{(k-1)!} \mathbb{E} \left| \mathbb{E} \left[ \hat{J}_r^{k-1}(\eta) \right] \cdot (\hat{q}(X) - q_0(X))(\hat{g}(X) - g_0(X))^{k-1} \right| \\
&\qquad + \sum_{k=1}^{r+1} \frac{1}{k!} \mathbb{E} \left| \theta_0 \mathbb{E} \left[ \hat{J}_r^{k-1}(\eta) \right] (\hat{g}(X) - g_0(X))^r \right| \\
&\le \sum_{k=1}^{r+1} \frac{1}{k!} \Big( k(4e\epsilon_1)^{r+1-k} \epsilon_1^{k-1} \epsilon_2 + C_\theta (4e\epsilon_1)^{r+1-k} \epsilon_1^r \Big) \\
&\le (4e)^r \left( \epsilon_1^r \epsilon_2 + C_\theta \epsilon_1^{r+1} \right),
\end{aligned}
\tag{124}
$$

where in the last but one step we use the fact that $|\theta_0| \le C_\theta$. By (114) and Chebyshev inequality, we have

$$
\begin{aligned}
&\left| \frac{2}{n} \sum_{i=n/2+1}^{n} (T_i - \hat{g}(X_i)) \hat{J}_r (T_i - \hat{g}(X_i)) - \mathbb{E}_P \left[ (T - \hat{g}(X)) \hat{J}_r (T - \hat{g}(X)) \right] \right| \\
&\le 4(\gamma n)^{-1/2} \mathbb{E}_P \left[ (T - \hat{g}(X))^2 \hat{J}_r (T - \hat{g}(X))^2 \right]^{1/2} \\
&\le 8(\gamma n)^{-1/2} r^2 \big[ 8(C_g + \psi_\eta) \big]^{r+1}
\end{aligned}
\tag{125}
$$

and

$$
\begin{aligned}
&\left| \frac{2}{n} \sum_{i=n/2+1}^{n} (Y_i - \hat{q}(X_i)) \hat{J}_r (T_i - \hat{g}(X_i)) - \mathbb{E}_P \left[ (Y - \hat{q}(X)) \hat{J}_r (T - \hat{g}(X)) \right] \right| \\
&\le 4(\gamma n)^{-1/2} \mathbb{E}_P \left[ (Y - \hat{q}(X))^2 \hat{J}_r (T - \hat{g}(X))^2 \right]^{1/2} \\
&\le 50(\gamma n)^{-1/2} (\psi_\xi + C_\theta \psi_\eta) \big[ 16(C_g + \psi_\eta) \big]^r
\end{aligned}
\tag{126}
$$

with probability $\ge 1 - \gamma$. Our assumption on $l$, Lemma G.8, and (125) together imply that

$$
\begin{aligned}
&\left| \frac{2}{n} \sum_{i=n/2+1}^{n} (T_i - \hat{g}(X_i)) \hat{J}_r (T_i - \hat{g}(X_i)) - \mathbb{E}_P \left[ (T - g_0(X)) J_r (T - g_0(X)) \right] \right| \le \frac{3}{4r!} \delta_{\mathsf{id}}, \\
&\left| \frac{2}{n} \sum_{i=n/2+1}^{n} (T_i - \hat{g}(X_i)) \hat{J}_r (T_i - \hat{g}(X_i)) \right| \ge \left| \mathbb{E}_P \left[ (T - g_0(X)) J_r (T - g_0(X)) \right] \right| - \frac{3}{4r!} \delta_{\mathsf{id}} \ge \frac{1}{4r!} \delta_{\mathsf{id}}
\end{aligned}
\tag{127}
$$

It also directly follows from (126) that

$$
\left| \frac{2}{n} \sum_{i=n/2+1}^{n} (Y_i - \hat{q}(X_i)) \hat{J}_r (T_i - \hat{g}(X_i)) \right| \le 51(\psi_\xi + C_\theta \psi_\eta) \big[ 16(C_g + \psi_\eta) \big]^r.
$$

By triangle inequality,

$$\left| \mathbb{E}_P \left[ (Y - \hat{q}(X)) \hat{J}_r (T - \hat{g}(X)) \right] \right|$$

$$\leq \left| \mathbb{E}_P \left[ (Y - q_0(X)) \hat{J}_r (T - \hat{g}(X)) \right] \right| + \mathbb{E}_P |\hat{q}(X) - q_0(X)| |\hat{J}_r (T - \hat{g}(X))|$$

$$\leq |\theta_0| \cdot \left| \mathbb{E}_P \left[ (T - g_0(X)) \hat{J}_r (T - \hat{g}(X)) \right] \right| + \mathbb{E}_P \left[ (\hat{q}(X) - q_0(X))^2 \right]^{1/2} \mathbb{E}_P \left[ \hat{J}_r (T - \hat{g}(X))^2 \right]^{1/2}$$

$$\leq C_\theta \left| \mathbb{E}_P \left[ (T - g_0(X)) J_r (T - g_0(X)) \right] \right| + \left[ 8(C_{\mathsf{g}} + \psi_{\mathsf{\eta}}) \right]^r \epsilon_2 + 3 \left[ 8(C_{\mathsf{g}} + \psi_{\mathsf{\eta}}) \right]^r \epsilon_1$$

$$\leq C_\theta \left| \mathbb{E}_P \left[ (T - g_0(X)) J_r (T - g_0(X)) \right] \right| + \left[ 8(C_{\mathsf{g}} + \psi_{\mathsf{\eta}}) \right]^r (3\epsilon_1 + \epsilon_2),$$

where the penultimate inequality follows from $\mathbb{E}_P \left[ \hat{J}_r (T - \hat{g}(X))^2 \right] \leq 3 \left[ 8(C_{\mathsf{g}} + \psi_{\mathsf{\eta}}) \right]^r \epsilon_1$ which can be shown in a similar fashion as Lemma G.9, and the final inequality follows from (113). Hence we can deduce that

$$\left| \frac{2}{n} \sum_{i=n/2+1}^{n} (Y_i - \hat{q}(X_i)) \hat{J}_r (T_i - \hat{g}(X_i)) \right|$$

$$\leq \left| \mathbb{E}_P \left[ (Y - \hat{q}(X)) \hat{J}_r (T - \hat{g}(X)) \right] \right| + 50(\gamma n)^{-1/2} (\psi_{\mathsf{\xi}} + C_\theta \psi_{\mathsf{\eta}}) \left[ 16(C_{\mathsf{g}} + \psi_{\mathsf{\eta}}) \right]^r$$

$$\leq C_\theta \kappa_{r+1} + \left[ 8(C_{\mathsf{g}} + \psi_{\mathsf{\eta}}) \right]^r (3\epsilon_1 + \epsilon_2) + 50(\gamma n)^{-1/2} (\psi_{\mathsf{\xi}} + C_\theta \psi_{\mathsf{\eta}}) \left[ 16(C_{\mathsf{g}} + \psi_{\mathsf{\eta}}) \right]^r \qquad (128)$$

$$\leq 6 C_\theta \kappa_{r+1}, \qquad (129)$$

where (128) follows from Lemma G.6 and (129) uses the constraint on $r$ given in (21). Recall that $\hat{\theta}$ satisfies $\frac{2}{n} \sum_{i=n/2+1}^{n} m(Z_i, \theta, \hat{h}(X_i)) = 0$, so (127) and (129) together imply that

$$|\hat{\theta}| \leq 6 C_\theta.$$

As a result, (125) and (126) yield

$$\left| \mathbb{E}_P \left[ m(Z, \hat{\theta}, \hat{h}(X)) \right] \right| \leq 64(\gamma n)^{-1/2} \left[ 16(C_{\mathsf{g}} + \psi_{\mathsf{\eta}}) \right]^r \left[ r^2 (C_{\mathsf{g}} + \psi_{\mathsf{\eta}}) + \psi_{\mathsf{\xi}} + C_\theta \psi_{\mathsf{\eta}} \right]$$

Subtracting this inequality from (124), we obtain

$$\left| (\theta_0 - \hat{\theta}) \mathbb{E}_P \left[ (Y - \hat{q}(X)) \hat{J}_r (T - \hat{g}(X)) \right] \right|$$

$$\leq 16^r \left[ \epsilon_1^r \epsilon_2 + C_\theta \epsilon_1^{r+1} + 64(C_{\mathsf{g}} + \psi_{\mathsf{\eta}})^r \left( r^2 (C_{\mathsf{g}} + \psi_{\mathsf{\eta}}) + \psi_{\mathsf{\xi}} + C_\theta \psi_{\mathsf{\eta}} \right) (\gamma n)^{-1/2} \right].$$

Therefore, we conclude that

$$\left| \theta_0 - \hat{\theta} \right| \leq r! 16^r \delta_{\mathsf{id}}^{-1} \left[ \epsilon_1^r \epsilon_2 + C_\theta \epsilon_1^{r+1} + 64(C_{\mathsf{g}} + \psi_{\mathsf{\eta}})^r \left( r^2 (C_{\mathsf{g}} + \psi_{\mathsf{\eta}}) + \psi_{\mathsf{\xi}} + C_\theta \psi_{\mathsf{\eta}} \right) (\gamma n)^{-1/2} \right].$$

### G.3   Proof of Theorem 5.3: ACE estimation error

In this section, we outline how the proof of Theorem 5.4 in the previous section can be slightly modified to obtain Theorem 5.3.

**Lemma G.10** (Condition for bias domination). *Suppose*

$$\Delta_i \triangleq \left| \kappa_i - \hat{\kappa}_i \right| \leq \underbrace{10(2C_{\mathsf{T}})^i (i-1)! r^{1/2} (\gamma n)^{-1/2}}_{variance} + \underbrace{(2i\epsilon)^i}_{bias}, \quad for \quad 1 \leq i \leq r,$$

$a \triangleq 2 \log \left( C_{\mathsf{T}} \epsilon_1^{-1} / 2 \right) > 0$, *and* $b \triangleq \log(\gamma n / 100) > 0$. *If*

$$r \leq \frac{b}{a}, \qquad (130)$$

*then for every* $1 \leq i \leq r$

$$10(2C_{\mathsf{T}})^i (i-1)! r^{1/2} (\gamma n)^{-1/2} \leq (2i\epsilon_1)^i,$$

*i.e. the bias term* $(2i\epsilon_1)^i$ *dominates the variance term in Lemma G.10.*

**Lemma G.11** (Bounding polynomial coefficients)**.** *For any $i \in [r+1]$ we have*

$$|a_{ir}| \leq \frac{1}{(i-1)!} C_{\mathsf{T}}^{r+1-i}. \tag{131}$$

**Proof** : The proof is the same as that of Lemma G.3. □

**Lemma G.12** (Bounding the estimation error of polynomial coefficients)**.** *For any $i \in [r+1]$ we have*

$$|a_{ir} - \hat{a}_{ir}| \leq \frac{1}{(i-1)!} (2C_{\mathsf{T}})^{r+1-i} \epsilon_1. \tag{132}$$

**Proof** : The proof is the same as that of Lemma G.4. □

**Lemma G.13** (Key lemma; approximate orthogonality)**.** *If $r$ satisfies Lemma G.10, then under $\mathcal{E}$ (defined in the proof of Theorem 5.1 in Appendix F.2) we have*

$$\mathbb{E}\left[\hat{J}_r^{(k)}(T - g_0(X))\right] \leq (4e\epsilon_1)^{r-k}.$$

**Proof** : The proof is the same as that of Lemma G.5. □

**Lemma G.14** (Linear–in–$\epsilon_1$ moment difference)**.** *Let*

$$T = g_0(X) + \eta, \qquad \mathbb{E}[\eta] = 0, \qquad \eta \perp X, \qquad \|\eta\|_{\psi_2} =: \psi_\eta < \infty.$$

*Assume an estimate $\hat{g}$ satisfies*

$$\|\hat{g} - g_0\|_{L^s(P)} \leq \epsilon_1 \leq \psi_\eta, \quad \text{for some } s \geq 2,$$

*and fix an integer $1 \leq i \leq s/2$. Then*

$$\mathbb{E}\left|(T - g_0(X))^i - (T - \hat{g}(X))^i\right| \leq i(2C_{\mathsf{T}})^i \epsilon_1. \tag{133}$$

**Proof** : The proof is similar to Lemma G.7; the only difference is that the moment bound of $\eta$ becomes $\mathbb{E}|\eta|^j \leq C_{\mathsf{T}}^j$. □

**Lemma G.15** (Identifiability guarantee)**.** *We have*

$$\left|\mathbb{E}_P\left[(T - \hat{g}(X))\hat{J}_r(T - \hat{g}(X))\right]\right| \geq \frac{1}{2r!}\delta_{\mathsf{id}}.$$

**Proof** : Similar to the proof of Lemma G.8, the left-hand-side can be shown to be $\leq 3(2C_{\mathsf{T}})^r \epsilon_1$. Combining with the constraint (19) yields the desired conclusion. □

**Lemma G.16** (Second-order moment bounds)**.** *The following inequalities hold:*

$$\mathbb{E}_P\left[(T - \hat{g}(X))^2 \hat{J}_r(T - \hat{g}(X))^2\right] \leq 2r^2(2C_{\mathsf{T}})^{2(r+1)} \tag{134}$$

$$\mathbb{E}_P\left[(Y - \hat{q}(X))^2 \hat{J}_r(T - \hat{g}(X))^2\right] \leq 112 C_{\mathsf{Y}}^2 (2C_{\mathsf{T}})^{2r} \tag{135}$$

Equipped with the above lemmas, we can then follow the arguments in Appendix G.2 to deduce that

$$\left|\theta_0 - \hat{\theta}\right| \leq r! 4^r \delta_{\mathsf{id}}^{-1} \left[\epsilon_1^r \epsilon_2 + C_\theta \epsilon_1^{r+1} + 64 C_{\mathsf{T}}^r (r^2 C_{\mathsf{T}} + C_{\mathsf{Y}})(\gamma n)^{-1/2}\right].$$

## G.4   Special cases of the ACE estimator

When $r = 3$, Theorem 5.4 immediately implies the following result:

**Corollary G.1** (Third-order ACE estimator)**.** *Let $\delta_{\mathsf{id}} > 0$ and $C_\theta, C_{\mathsf{g}}, C_{\mathsf{q}}, \psi_\eta, \psi_\xi \geq 1$ be constants and $\mathcal{P}_0$ be the set of all distributions in $\mathcal{P}(C_\theta, C_{\mathsf{g}}, C_{\mathsf{q}}; \psi_\xi, \psi_\eta)$ such that $\eta$ is independent of $X$ and $|\kappa_4| \geq \delta_{\mathsf{id}}$. Suppose that $\theta$ is the solution to (59) with $m(Z, \theta, h(X)) = [Y - q(X) - \theta(T - g(X))]\left[(T - g(X))^3 - 3\mu_2'(T - g(X)) - (\mu_3' - 3\mu_1'\mu_2')\right]$. Then for any $\gamma \in (0, 1)$, there exists a constant $C_\gamma$ such that*

$$\mathfrak{R}_{n,1-\gamma}(\hat{\theta}; \mathcal{P}_{s,\epsilon}(\hat{h})) \leq C_\gamma \delta_{\mathsf{id}}^{-1}\left[\epsilon_1^3 \epsilon_2 + C_\theta \epsilon_1^4 + (C_{\mathsf{g}} + \psi_\eta + \psi_\xi + C_\theta C_\eta)(C_{\mathsf{g}} + \psi_\eta)^3 (\gamma n)^{-1/2}\right].$$

The choice of the moment function in Corollary G.1 has also been proposed in Mackey et al. [2018], though their results are restricted to the high-dimensional linear regression setting. However, the rate that we derive from Corollary G.1 is faster than theirs, and as a consequence, in Corollary G.3 we need a weaker sparsity assumption to achieve $\mathcal{O}(n^{-1/2})$ rate. The main insight for deriving this improved rate is that the moment function is, in fact, third-order orthogonal. By contrast Mackey et al. [2018] only shows that it is second-order orthogonal. We will revisit this setting in Section 6, where we empirically verify the effectiveness of ACE for different choices of $r$.

For $r \geq 4$, to the best of our knowledge, the estimators derived from Theorem 5.4 are new. For illustration purpose, we derive the guarantee for $r = 4$ in the following:

**Corollary G.2** (Fourth-order ACE estimator). *Let $\delta_{\mathsf{id}} > 0$ and $C_\theta, C_{\mathsf{g}}, C_{\mathsf{q}}, \psi_\eta, \psi_\xi \geq 1$ be constants and $\mathcal{P}_0$ be the set of all distributions in $\mathcal{P}(C_\theta, C_{\mathsf{g}}, C_{\mathsf{q}}; \psi_\xi, \psi_\eta)$ such that $\eta$ is independent of $X$ and $|\kappa_5| \geq \delta_{\mathsf{id}}$. Suppose that $\theta$ is the solution to (59) with*

$$
\begin{aligned}
m(Z, \theta, h(X)) = [Y - q(X) - \theta\,(T - g(X))]\big[(T - g(X))^4 - 6\mu_2'(T - g(X))^2 \\
- 4(\mu_3' - 3\mu_1'\mu_2')(T - g(X)) - (\mu_4' - 6\mu_2'^2 - 4\mu_1'\mu_3' + 12\mu_1'^2\mu_2' - 6\mu_1'^4)\big].
\end{aligned}
$$

*Then for any $\gamma \in (0, 1)$, there exists a constant $C_\gamma$ such that*

$$
\mathfrak{R}_{n,1-\gamma}(\hat{\theta}; \mathcal{P}_{s,\epsilon}(\hat{h})) \leq C_\gamma \delta_{\mathsf{id}}^{-1}\big[\epsilon_1^4 \epsilon_2 + C_\theta \epsilon_1^5 + (C_{\mathsf{g}} + \psi_\eta + \psi_\xi + C_\theta C_\eta)(C_{\mathsf{g}} + \psi_\eta)^4 (\gamma n)^{-1/2}\big].
$$

**Remark G.2** (Cumulants versus moments). *Generalizing the $r = 3$ case to $r \geq 4$ is highly nontrivial. Indeed, given the construction in Corollary G.1, one might be tempted to consider*

$$
m(Z, \theta, h(X)) = [Y - q(X) - \theta\,(T - g(X))]\big[(T - g(X))^r - r\mu_{l-1}'(T - g(X)) - (\mu_r' - r\mu_1'\mu_{r-1}')\big]
$$

*with $h = (g, q)$. In this case, let $\Delta(x) = g(x) - \hat{g}(x)$, then we have*

$$
\begin{aligned}
\mathbb{E}\big[D^{(0,1)}m(Z, \theta, h(X)) \mid X\big] &= -\mu_r + \mu_r' - r\mu_1'\mu_{r-1}' \\
&= -\mathbb{E}[\eta^r] + \mathbb{E}[(\eta + \Delta(X))^r \mid X] - r\mathbb{E}[\eta + \Delta(X) \mid X]\mathbb{E}[(\eta + \Delta(X))^{r-1} \mid X] + \mathcal{O}_P(n^{-1/2}) \\
&\approx -\frac{r(r-1)}{2}\mathbb{E}[\eta^{r-2}]\Delta(X)^2 + \mathcal{O}_P(n^{-1/2}) = \mathcal{O}_P(\epsilon_1^2 + n^{-1/2}),
\end{aligned}
$$

*which is the same as the $r = 3$ case up to constants. As a result, this approach does not yield rates faster than Corollary G.1.*

In Corollaries G.1 and G.2, we omit the constants in the upper bounds. As shown in Theorem 5.4, the constants for $r$-th order orthogonal estimators can be at most $(Cr)^r$ for some constant $C$, that grows super-exponentially, and, as demonstrated in Remark 5.4, the growth of this constant is in some cases offset by the growth of the absolute cumulant $|\kappa_{r+1}|$.

### G.5 ACE estimation error for high-dimensional sparse linear regression

**Corollary G.3** (ACE estimation error for high-dimensional sparse linear regression). *In the setting of Theorem 5.4 with high-dimensional linear nuisance (24), suppose that the nuisance estimators $(\hat{g}, \hat{q})$ are respectively constructed via Lasso regression of $T$ and $Y$ onto $X$ with an appropriately chosen regularization parameter. If $\max(s_1, s_1^{r/(r+1)} s_2^{1/(r+1)}) = o(n^{r/(r+1)} / \log p)$, then, with probability $\geq 1 - \gamma$, we have $|\hat{\theta} - \theta_0| \leq C_\gamma C_r \delta_{\mathsf{id},r}^{-1} n^{-1/2}$ for constants $C_\gamma$ and $C_r$ depending only on $\gamma$ and $r$ respectively.*

**Proof** : As derived in Mackey et al. [2018, Sec. I], there exists some constant $C_\gamma' > 0$ such that on an event $\mathcal{E}$ with probability $\geq 1 - \gamma/2$, the Lasso nuisance estimates simultaneously provide the bounds

$$
\|\hat{\alpha} - \alpha_0\|_2 \leq C_\gamma \sqrt{(s_1/n)\log p}, \quad \|\hat{\beta} - \beta_0\|_2 \leq C_\gamma \sqrt{(s_2/n)\log p}, \tag{136}
$$

where we recall that $\beta_0, \alpha_0 \in \mathbb{R}^p$. Since $X \sim \mathcal{N}(0, \boldsymbol{I})$, Cauchy-Schwarz and Khintchine's inequality [Vershynin, 2018, Corollary 2.6.4] imply that, on this event,

$$
\begin{aligned}
\epsilon_1 = \|\hat{g} - g_0\|_{L^{2r+2}(P_X)} &= \|\langle X, \hat{\alpha} - \alpha_0\rangle\|_{L^{2r+2}(P_X)} \\
&\leq K\sqrt{2r+2}\|\hat{\alpha} - \alpha_0\|_2 \leq K\sqrt{(2r+2)(s_1/n)\log p} \quad \text{and, similarly,} \tag{137} \\
\epsilon_2 = \|\hat{q} - q_0\|_{L^{2r+2}(P_X)} &\leq K\sqrt{(2r+2)(s_2/n)\log p},
\end{aligned}
$$

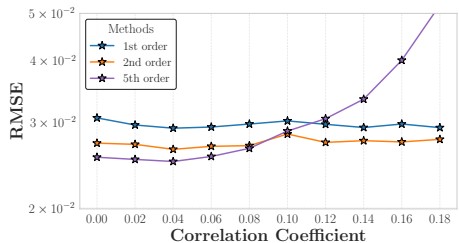

(a) The average MSE of ACE estimation with order $l = 1, 2, 5$ as functions of $\xi$.

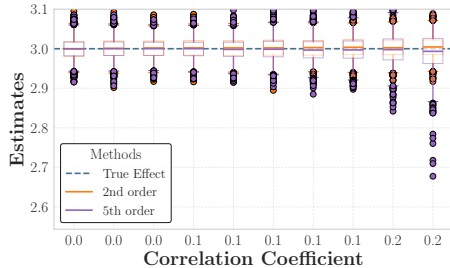

(b) The distribution of estimates for second- and fifth-order ACE estimation, with varying $\xi$.

Figure 2: The sensitivity of ACE estimators to correlation of the covariate $X$ and the noise variable $\eta$.

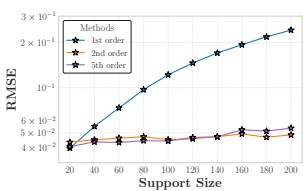

(a) The average MSE of ACE estimator with $l = 1, 2, \cdots, 6$ as functions of sample size.

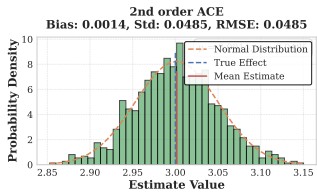

(b) The distribution of estimates for second-order ACE estimation when $n = 10000, s = 200$.

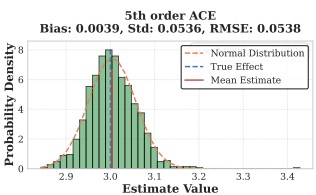

(c) The distribution of estimates for fifth-order ACE estimation when $n = 10000, s = 200$.

Figure 3: Experiment results for ACE estimators with fixed sample size $n = 10000$ and varying sparsity.

for some universal constant $K > 0$. Our assumption that

$$\max(s_1, s_1^{r/(r+1)} s_2^{1/(r+1)}) = o(n^{r/(r+1)}/\log p) \tag{138}$$

further implies that, on the event $\mathcal{E}$, $\max\{\epsilon_1^{r+1}, \epsilon_1^r \epsilon_2\} = o(n^{-1/2})$. Applying Theorem 5.4 with $\gamma$ replaced by $\gamma/2$, we can therefore conclude that the advertised bound holds with probability $(1 - \gamma/2)^2 \geq 1 - \gamma$.

$\square$

## H   Additional experiment results and discussion

In view of Theorem 5.4, the error rate of $r$-th order ACE estimator depends on a *bias* term that scales as $\mathcal{O}(\epsilon_1^r \epsilon_2 + \epsilon_1^{r+1})$ and a *variance* term that scales as $\mathcal{O}(n^{-1/2})$, multiplied by a constant depending on $r$. In practice, this constant is often non-negligible. Hence, to choose an appropriate order $r$, one should take into consideration its effect on the final estimation error.

**Varying sample size.** We first investigate the performance of ACE estimators with $r \leq 6$ with varying sample size. The results are reported in Figure 2. We set $r = 1$ as a baseline and only plot the results of estimators that are better than $r = 1$. From Figure 1 one can see that the fifth order estimator performs the best, followed by the second order one. The fifth order estimator incurs large errors for small sample sizes ($n = 2000$) but the error decreases rapidly when $n$ grows larger.

When $n \geq 8000$, the decreasing rates of different estimators are roughly the same. This is because in this regime, variance becomes the dominating term in the total mean-squared error. From a theoretical perspective, this is because for fixed $s$, the LASSO estimates induce errors that scale as $n^{-1/2}$, so for any $r \geq 1$, the bias term becomes $n^{-(r+1)/2} \ll n^{1/2}$. In Figure 1d and Figure 1e we plot the distributions of the estimates produced by first and fifth order ACE estimators. One can see that the variances of these two estimators are roughly at the same level, while the first-order estimator induces significantly larger biases, especially when $n$ is small.

In Figure 1f, we plot the $95\%$ confidence intervals for each individual estimates. We observe that the actual percentage of confidence intervals covering the ground-truth parameter is $94.9\%$, quite close to what Corollary E.1 predicts.

**Correlation between covariate and treatment noise.** The theoretical benefits of ACE estimators with $r \geq 3$ crucially relies on the assumption that $X$ and $\eta$ are independent, which might be restrictive. However, it might be the case that they are weakly correlated, *i.e.*, only a small part of $\eta$ is correlated with $X$. We would like to understand the sensitivity of our estimators' performance with respect to such correlation.

Specifically, we assume that the treatment variable is drawn from

$$T = g_0(X) + (1 + \xi X_1)\eta, \tag{139}$$

where $X_1$ is the first component of $X$ and $\eta$ is a mean-zero random variable independent of $X$. We set $p = 100, n = 20000, s = 40$ investigate the estimation error of ACE with different $r$'s as a function of the correlation coefficient $\xi$.

The results are reported in Figure 2, where we compare the top-3 estimators among $r = 1, 2, \cdots, 6$. The first- and second-order estimators have stable performance across different correlations, while the performance of the fifth-order one deteriorates rapidly when $\xi \geq 0.1$. This suggests that it would be better to use the second-order estimator unless one has strong prior knowledge that $X$ and $\eta$ are weakly correlated. In the context of pricing experiments, the data is drawn from a company's historical experimentation records, so that scientists are likely to have knowledge about the high-level design principles of such experiments.

**Varying sparsity.** Lastly, we investigate the relative performance of ACE estimators with different level of sparsity, for fixed $p = 1000, n = 10000$. Recall that sparsity affects the first-stage nuisance errors, which in turn affects the bias of our estimates.

The results are reported in Figure 3. From Figure 3a we can see that the performance of first-order estimator deteriorates rapidly when the support size grows. By contrast, the performances of second- and fifth-order estimators are quite stable, with the fifth-order one slightly better for smaller $s$. This is not surprising, since one can see from Theorem 5.4 that the bias term for the fifth-order estimator has a larger bias, so it would only be smaller than the second-order counterpart when the nuisance errors are small enough. As shown in Figures 3b and 3c, when $s = 200$, the fifth-order estimator indeed incurs a larger bias.

# I More results for orthogonal machine learning

## I.1 Construction of orthogonal moment functions

**Theorem I.1** (Construction of higher-order orthogonal moments). *Let $a_{ik}(x)$ and $\rho_{ik}(w)$ be as defined in Lemma 4.1, then the following statements hold:*

*(1). If $\mathbb{E}\left[(1 + |\eta|)|\rho_{ik}(\eta)| \mid X = x\right] < +\infty$ holds for all $1 \leq i \leq M_r$ and $x \in \mathcal{X}$, then the moment function*

$$m_r\left(Z, \theta, q(X), g(X), \{a_{ik}(X)\}_{i=1}^{M_r}\right) = \left[Y - q(X) - \theta(T - g(X))\right] J_r(T - g(X), X)$$

*with nuisance functions $h_1 = g, h_2 = q$ and $h_{i+2} = a_{ik}, i \in [M_r]$ is $(S_0, S_1)$-orthogonal where $S_0$ contains all $\alpha \in \mathbb{Z}_{\geq 0}^\ell$ that satisfies at least one of the following conditions:*

*(i).$\|\alpha\|_1 \leq 1$; (ii).$\alpha_1 + \alpha_2 \leq k$; (iii). $\alpha_4 = 1$, $(\alpha_1, \alpha_2) \in \{(0, 1), (1, 0)\}$, the remaining $\alpha_i$'s equal zero;*

*and $S_1 = \left\{\alpha \in \mathbb{Z}_{\geq 0}^\ell : \max\left\{\alpha_2, \sum_{i=3}^{M_r+2} \alpha_i\right\} \geq 2\right\}$. Moreover, $D^\alpha m$ exists and is continuous for all $\|\alpha\|_1 \leq k + 1$.*

*(2). Let $C_\theta, C_T, C_Y > 0$ be constants and $\mathcal{P} = \mathcal{P}_b^\star(C_\theta, C_T, C_Y)$. Let $\theta$ be the solution of (59) with $m = m_r$. Then under Assumption E.1 (2) and (3), for any $\gamma \in (0, 1)$ there exists a*

*constant $C_\gamma > 0$ that only depends on $\gamma, C_\theta, C_T, C_Y$, such that*

$$\mathfrak{R}_{n,1-\gamma}(\hat{\theta}; \mathcal{P}_{s,\epsilon}(\hat{h})) \leq C_\gamma \delta_{\mathsf{id}}^{-1} \left[ \sqrt{\frac{V_{\mathsf{m}}}{n}} + k\lambda_\star(\epsilon_1^{r+1} + \epsilon_1^r \epsilon_2) \right. \tag{140}$$
$$\left. + M_{r+1}\lambda_\star\big((\epsilon_1 + \epsilon_2)\tilde{\epsilon} + (\epsilon_1^2 + \epsilon_1\epsilon_2)\epsilon_4\big) \right].$$

*holds with probability $\geq 1 - \gamma$, where $\tilde{\epsilon} = \max_{i \neq 1,2,4} \epsilon_i$, $s \geq k+1$ and*

$$\lambda_\star = \sup_{P \in \mathcal{P}} \max_{\alpha \in (\mathcal{I}_{r,\ell} \setminus S_0) \cup (\mathcal{I}_{r+1,\ell,0} \setminus S_1)} \|D^\alpha m_r(Z, \theta_0, h_0(X))\|_{L^2(P)}.$$

Theorem I.1 is proven in Appendix I.2. It shows that by successive integration of $J_0$, we can construct moment functions that are orthogonal with respect to $g_0(X)$ and $q_0(X)$ with arbitrarily high order, while being only first-order orthogonal with respect to the nuisances $a_{ik}(X)$. In (3), the constants $\delta_{\mathsf{id}}$ and $V_{\mathsf{m}}$ depend on the order $k$; we will write them as $\delta_{\mathsf{id},k}$ and $V_{\mathsf{m},k}$ to avoid confusion.

From another perspective, Theorem I.1 can be viewed as a special case of Theorem 4.1, because the construction of the moment function there ensures that $\mathbb{E}_P\left[\sum_{i=1}^{M_j} a_{ij}(X)\rho_{ij}(T - g_0(X)) \mid X = x\right] = 0, j = 1, 2, \cdots, k$. In Theorem I.1, the $a_{ij}(\cdot)$'s are viewed as nuisance functions which allows for *exact* orthogonalily properties. By contrast, in Theorem 4.1 only $q(\cdot)$ and $g(\cdot)$ are treated as nuisance functions and we only ask for *approximate* orthogonality. While they look similar at first glance, an important observation is that this result does *not* rely on any explicit assumptions on the estimation errors of $\hat{a}_{ij}(\cdot)$'s. Indeed, it might be possible that the left-hand side of (15) is much smaller than the individual estimation errors of $\hat{a}_{ij}(\cdot)$'s, because these individual errors cancel out in the summation. This observation will prove helpful in Section 5.

Recall that in Theorem 3.2 we show that higher-order orthogonality is impossible for the Gaussian treatment, even with known variance. In this case, since the distribution of $\eta = T - g_0(X) \mid X$ is known, the functions $a_{ik}$'s are known as well. However, in the Gaussian case, any moment function constructed from (8) when $k \geq 2$ would violate the identifiability assumption (Assumption E.1 (3)). We prove this in Proposition I.2.

## I.2 Proof of Theorem I.1

In this subsection, we provide a straightforward instantiation of Theorem E.1 using the moment functions constructed in Lemma 4.1. For simplicity, we restrict ourselfs to the case where the treatment and outcome are both bounded.

*Proof of (1).* The statement follows directly from induction. By assumption, the conclusion holds for $k = 0$. Now assume that it holds for some $k - 1 \geq 0$, then

$$I_r(w, x) = \int_0^w J_{r-1}(w', x)\mathrm{d}w' = \sum_{i=1}^{M_r} a_{ir}(x) \int_0^w \rho_{ir}(w')\mathrm{d}w',$$

so we can choose $M_r = M_{r-1} + 1$, $a_{ir}(x) = a_{i-1,r-1}(x), \rho_{ir}(w) = \int_0^w \rho_{i-1,r-1}(w')\mathrm{d}w', 2 \leq i \leq M_r - 1$ and $a_{1,r}(x) = -\mathbb{E}[I_r(T - g_0(X), X) \mid X = x], \rho_{1,r}(w) \equiv w$, proving the result for $r$.

*Proof of (2).* First note that
$$Y - q_0(X) - \theta_0(T - g_0(X)) = \epsilon,$$
so that
$$\mathbb{E}[m(Z, \theta_0, h_0(X))] = \mathbb{E}[J_r(T - g_0(X), X)\mathbb{E}[\epsilon \mid X, T]] = 0.$$

For any $\alpha \in \mathbb{Z}_{\geq 0}^{M_r+2}$ with $\|\alpha\|_1 = 1$, we consider three cases,

- $\alpha_1 = 1$, then (8) implies that

$$\mathbb{E}[D^\alpha m(Z, \theta_0, h_0(X))] = -\mathbb{E}[J_r(T - g_0(X), X)] = 0.$$

- $\alpha_2 = 1$, then

$$\mathbb{E}\left[D^\alpha m(Z, \theta_0, h_0(X))\right]$$
$$= \theta_0 \mathbb{E}\left[J_r(T - g_0(X), X)\right] - \mathbb{E}\left[(Y - q_0(X) - \theta_0(T - g_0(X)))J_{r-1}(T - g_0(X), X)\right] = 0.$$

- $\alpha_{i+2} = 1$ for some $i \geq 1$, then

$$\mathbb{E}\left[D^\alpha m(Z, \theta_0, h_0(X))\right] = \mathbb{E}\left[(Y - q_0(X) - \theta_0(T - g_0(X)))\rho_{ir}(T - g_0(X))\right] = 0.$$

Next, we consider $\alpha$'s of form $(\alpha_1, \alpha_2, 0, \cdots, 0)$ where $\alpha_1 + \alpha_2 \leq r$. Since $m$ is affine in $q$, when $\alpha_1 \geq 2$ we have $D^\alpha m(Z, \theta_0, h_0(X)) = 0$, so we only need to consider the case when $\alpha_1 \in \{0, 1\}$. Similar to the arguments above,

- If $\alpha_1 = 1, \alpha_2 \leq r - 1$ then

$$\mathbb{E}\left[D^\alpha m(Z, \theta_0, h_0(X))\right] = (-1)^{\alpha_2+1}\mathbb{E}\left[J_{r-\alpha_2}(T - g_0(X), X)\right] = 0.$$

- If $\alpha_1 = 0, \alpha_2 \leq r$ then

$$\mathbb{E}\left[D^\alpha m(Z, \theta_0, h_0(X))\right] = (-1)^{\alpha_2}\alpha_2\theta_0\mathbb{E}\left[J_{r+1-\alpha_2}(T - g_0(X), X)\right]$$
$$- \mathbb{E}\left[(Y - q_0(X) - \theta_0(T - g_0(X)))J_{r-\alpha_2}(T - g_0(X), X)\right] = 0.$$

Furthermore,

- If $\alpha_1 = 1, \alpha_4 = 1, \rho_{4,r}(w) \equiv w$ and the remaining $\alpha_j$'s are all zero, then

$$\mathbb{E}\left[D^\alpha m(Z, \theta_0, h_0(X))\right] = -\mathbb{E}[T - g_0(X)] = 0.$$

- If $\alpha_1 = 1, \alpha_4 = 1, \rho_{4,r}(w) \equiv w$ and the remaining $\alpha_j$'s are all zero, then

$$\mathbb{E}\left[D^\alpha m(Z, \theta_0, h_0(X))\right] = 2\theta_0\mathbb{E}[T - g_0(X)] = 0.$$

This proves the orthogonality properties related to $S_0$. Since $m(Z, \theta, \gamma)$ is affine in $\gamma_i, i \geq 2$ (which corresponds to the nuisance functions $q$ and $a_{ir}, i \in [M_r]$), we have $D^\alpha m \equiv 0$ as long as $\sum_{i \geq 2} \alpha_i \geq 2 \Leftrightarrow \alpha \in S_1$. Hence $m$ is $(S_0, S_1)$-orthogonal as desired.

Finally, the continuity of $D^\alpha m$ is obvious since $m$ is a quadratic function in terms of the nuisance functions.

*Proof of (3).* By Theorem E.1,

$$\Re_{n,1-\gamma}(\hat{\theta}_{\mathsf{OML}}; \mathcal{P}_{s,\epsilon}(\hat{h})) \leq C_\gamma \delta_{\mathsf{id}}^{-1} \times \left(\sqrt{\frac{V_{\mathsf{m}}}{n}} + \lambda_\star \sum_{\alpha \in (\mathcal{I}_{r,\ell}\setminus S_0) \cup (\mathcal{I}_{r+1,\ell,0}\setminus S_1)} \frac{1}{\|\alpha\|_1!}\prod_{i=1}^\ell \epsilon_i^{\alpha_i}\right). \quad (141)$$

As shown in part (2), $\mathcal{I}_{r,\ell}\setminus S_0$ only contains $\alpha$ with $\alpha_2 \in \{0, 1\}, \alpha_1 + \alpha_2 \geq 1, \min\{\alpha_1, \alpha_2\} + \alpha_4 \geq 3$ and $\sum_{i \geq 3} \alpha_i = 1$. Thus

$$\sum_{\alpha \in (\mathcal{I}_{r,\ell}\setminus S_0)} \frac{1}{\|\alpha\|_1!}\prod_{i=1}^\ell \epsilon_i^{\alpha_i} \leq \sum_{j=2}^{r+1} \frac{1}{j!}\left(\epsilon_1^{j-2}\epsilon_2 + \epsilon_1^{j-1}\right)\tilde{\epsilon} + (\epsilon_1^2 + \epsilon_1\epsilon_2)\epsilon_3 \leq 3(\epsilon_1 + \epsilon_2)\tilde{\epsilon} + (\epsilon_1^2 + \epsilon_1\epsilon_2)\epsilon_3.$$

On the other hand, $\mathcal{I}_{r+1,\ell,0} \setminus S_1$ contains $\alpha$ with $\alpha_2 \leq 1$ and $\sum_{i \geq 3} \alpha_1 \leq 1$, so that

$$\sum_{\alpha \in \mathcal{I}_{r+1,\ell,0}\setminus S_1} \frac{1}{\|\alpha\|_1!}\prod_{i=1}^\ell \epsilon_i^{\alpha_i} \leq \frac{1}{(r+1)!}\left[(\epsilon_1^{r+1} + \epsilon^r\epsilon_2) + (\epsilon_1^r + \epsilon^{r-1}\epsilon_2)\tilde{\epsilon}\right]$$

Combining the above two inequalities, the conclusion follows.

## I.3 Example: heteroscedastic nonparametric regression

In general, $a_{ik}(\cdot)$ may be hard to estimate since it is a linear combination of conditional moment functions. Generally speaking, there is no guarantee that estimating these conditional moment functions is easier than estimating the nuisance functions.

In this section, we revisit the nonparametric regression problem with heteroscedastic noise, where fast rates for estimating $a_{ik}$'s are indeed achievable. Specifically, suppose that the treatment variable is sampled from the regression model

$$T = g_0(X) + \eta,$$

where the noise variable $\eta = V_0(X)^{1/2}\eta^\star$ and $\eta^\star$ satisfies $|\eta^\star| \leq C_{\eta^\star}$ a.s., $\mathbb{E}[\eta^\star \mid X] = 0$ and $\mathbb{E}[\eta^{\star 2} \mid X] = 1$. For this problem, the following result is known from Wang et al. [2008].

**Proposition I.1.** *[Wang et al., 2008, Theorems 1,2 and Remark 3] Assuming that $\mathcal{X} = [0,1]$ and $g_0(\cdot), V_0(\cdot)$ are $\alpha$ and $\beta$-th order smooth respectively, then given i.i.d. data $\{(X_i, T_i)\}_{i=1}^n$ from some distribution such that the marginal density of $X$ exists and is bounded away from $0$, there exists an estimator $\hat{V}(\cdot)$ that achieves the optimal mean-square error rate $\|\hat{V}(X) - V_0(X)\|_{P,2} = \mathcal{O}_P\left(n^{-\min\{2\alpha, \beta/(2\beta+1)\}}\right)$.*

In particular, when $\beta > \alpha$, one can in fact estimate $V_0(\cdot)$ with higher accuracy than estimating $g_0(\cdot)$. In this subsection, we additionally assume that the distribution of $\eta^\star$ is known and let $\mu_r^\star(x) = \mathbb{E}[\eta^{\star r} \mid X = x]$ be its $r$-th moment. Thus we can estimate $\mu_r(x) = \mathbb{E}[\eta^r \mid x = x] = V_0(x)^{r/2}\mu_r^\star(x)$ with $\hat{V}(x)^{r/2}\mu_r^\star(x)$. We conjecture that by using a similar approach as Wang et al. [2008] one can directly construct higher-order moment estimates for unknown $\eta^\star$, so the assumption that $\eta^\star$ is known can be removed.

To state our main result for this setting, we assume that $g_0$ and $V_0$ are $\alpha$ and $\beta$-th order smooth respectively:

**Assumption I.1.** *$g \in \Lambda^\alpha(M_g)$ and $V \in \Lambda^\beta(M_V)$ for some constants $M_g, M_V > 0$ and $\alpha, \beta > 0$.*

Let $\hat{g}$ be an optimal estimate of $g_0$ under the $L^\infty$-norm, that achieve the rate $\epsilon_1 = \mathcal{O}\left((\log n/n)^{\alpha/(2\alpha+1)}\right)$ [Stone, 1982]. Also let $\hat{V}$ be the estimate of $V_0$ in Wang et al. [2008] that achieves the $L^2$-rate

$$\epsilon_{\mathsf{v}} = \mathcal{O}\left(n^{-\min\{4\alpha, 2\beta/(2\beta+1)\}}\right).$$

Consider the moment function $m_r$ defined in (10). Its nuisance functions $a_{ik}(x), i = 0, 1, \cdots, k$ are functions of the conditional moments of $\eta \mid X$. Then we can derive their estimates $\hat{a}_{ik}(x)$ by directly plugging in the variance estimates: $\hat{\mu}_r(x) = \hat{V}(x)^{r/2}\mu_r^\star(x)$. The next theorem provides theoretical guarantee for the resulting estimate derived from this approach:

**Theorem I.2** (Error rate for heteroscedastic nonparametric regression). *Let $\mathcal{X} = [0,1]$, $C_\mathsf{T} \geq 1$ be a real number and $\mathcal{P}$ be the set of all distributions in $\mathcal{P}^\star$ that satisfy Assumption I.1 and $|T| \leq C_\mathsf{T}$. Let $\hat{g}, \hat{V}$ be defined above and $\hat{q}$ be some estimator of $q_0$ such that $\|\hat{q} - q_0\|_{L^2(P)} \leq \epsilon_2$. Also assume that $|\hat{\mu}_r(x)| \leq (2C_\mathsf{T})^r$[6]. Then the following statements hold:*

*(1). Consider the nuisance functions and their estimates specified in the previous paragraph. Then for all $i \in [r+1]$, we have*

$$\|a_{ir} - \hat{a}_{ir}\|_{L^2(P)} \leq \left(8eC_\mathsf{v}^{1/2}C_{\eta^\star}\right)^{r+1-i}\frac{r+1-i}{i!}\epsilon_\mathsf{v} \leq A_r\epsilon_\mathsf{v},$$

*where $A_r = r\left(8eC_\mathsf{v}^{1/2}C_{\eta^\star}\right)^{r+1}$.*

*(2). Let $\hat{\theta}_{\mathrm{OML}}$ be the solution to (59) with the moment function $m = m_r$ defined in (10). Then in the setting of Theorem I.1, for any $\gamma \in (0,1)$, there exists a constant $C_\gamma > 0$ such that*

$$\mathfrak{R}_{n,1-\gamma}(\hat{\theta}; \mathcal{P}_{s,\epsilon}(\hat{h})) \leq C_\gamma \delta_{\mathrm{id},r}^{-1}\left[\sqrt{\frac{V_{\mathsf{m},r}}{n}} + r\lambda_\star\left(\epsilon_1^{r+1} + \epsilon_1^r\epsilon_2 + A_r(\epsilon_1 + \epsilon_2)\epsilon_\mathsf{v}\right)\right]. \quad (142)$$

---

[6]This is without loss of generality, since otherwise we can replace $\hat{\mu}_r$ with $\min\{(2C_\mathsf{T})^r, \max\{-(2C_\mathsf{T})^r, \hat{\mu}_r\}\}$. By assumption we know that $|\mu_r| \leq (2C_\mathsf{T})^r$, so this would only reduce the estimation error.

Theorem I.2 is proven in Appendix I.4.1. Note that Theorem I.2 makes structural assumptions on $g_0, V_0$ but not on $q_0$. As a result, the assumption is stronger than the fully structure-agnostic setting, while being weaker than the Holder-smoothness setting. Even in this interpolated regime, to the best of our knowledge, there is no existing results that achieve faster rates than DML.

It is worth noticing that the rate $\epsilon_v$ can be faster than $\epsilon_1$ to arbitrary order. Thus, there exists an optimal $k$ that balances the dependency on $n$ and the magnitude of the constants $A_r, \delta_{\mathsf{id},r}, V_{\mathsf{m},r}, \lambda_\star$. For $r = 2, 3$, assuming that these constants are uniformly bounded and that $q$ also belongs to a Holder class, we can derive the following result:

**Corollary I.1.** *In the setting of Theorem I.2, if we additionally assume that $q \in \Lambda^\gamma(M_q)$ and $\|\hat{q} - q_0\|_{L^2(P_X)} \leq \epsilon_2 = \mathcal{O}(n^{-2\gamma/(2\gamma+1)})$, then the following holds:*

*(1). Let $r = 2$, then $\mathfrak{R}_{n,1-\gamma}(\hat{\theta}_{\mathrm{OML}}; \mathcal{P}_{s,\epsilon}(\hat{h})) = \mathcal{O}_P(n^{-1/2})$ as long as $\min\{\alpha, \gamma\} > \frac{1}{4}$ and $\min\{\alpha, \gamma\}\beta > \frac{1}{4}$.*

*(2). Let $r = 3$, then $\mathfrak{R}_{n,1-\gamma}(\hat{\theta}_{\mathrm{OML}}; \mathcal{P}_{s,\epsilon}(\hat{h})) = \mathcal{O}_P(n^{-1/2})$ as long as $\min\{\alpha, \gamma\} > \frac{\sqrt{3}-1}{4}$ and $\min\{\alpha, \gamma\}\beta > \frac{1}{4}$.*

Corollary I.1 implies that $\mathcal{O}_P(n^{-1/2})$ rate can be achieved under weaker smoothness requirements if one uses third-order orthogonal estimators. Its proof can be found in Appendix I.5.

## I.4 Technical details in Appendix I.3

### I.4.1 Proof of Theorem I.2

*Proof of part (1).* By definition, we have

$$\hat{a}_{ir}(x) = \frac{1}{(i-1)!(r+1-i)!} \sum_{\pi \in \Pi_{r+1-i}} (-1)^{|\pi|-1} \prod_{B \in \pi} \hat{\kappa}_{|B|}(x), \tag{143}$$

while $\hat{\kappa}_i$'s are cumulant estimates obtained by directly plugging in $\hat{V}(\cdot)$:

$$\hat{\kappa}_i = \hat{\mu}_i - \sum_{j=1}^{i-1} \binom{i-1}{j-1} \hat{\mu}_{i-j}\hat{\kappa}_j, \quad \hat{\mu}_1 = 0, \ \hat{\mu}_i = \mathbb{E}_n\big[\hat{V}(X)^{r/2}\mu_r^\star(X)\big] (i \geq 2).$$

By the mean value theorem, there exists some $\tilde{V}(\cdot)$ between $\hat{V}$ and $V_0$ such that $\hat{V}^{j/2}(x) - V_0^{j/2}(x) = \frac{j}{2}\tilde{V}^{j/2-1}(x)(\hat{V}(x) - V_0(x))$. Hence

$$\begin{aligned}
\big\|\hat{V}^{j/2} - V_0^{j/2}\big\|_{L^2(P)} &= \frac{j}{2}\big\|\tilde{V}^{j/2-1}(x)(\hat{V}(x) - V_0(x))\big\|_{L^2(P)} \\
&\leq \frac{j}{2}\big\|\tilde{V}^{j/2-1}\big\|_{L^\infty(P)} \cdot \big\|\hat{V} - V_0\big\|_{L^2(P)} \\
&\leq \frac{j}{2}C_{\mathsf{v}}^{j/2-1}\big\|V - \hat{V}_0\big\|_{L^2(P)},
\end{aligned}$$

where we recall that $C_{\mathsf{v}}$ is the assumed uniform upper bound on $|V_0(X)|$ and $|\hat{V}(x)|$. We can then bound the estimation error of $\hat{\mu}_r(\cdot)$ as follows:

$$\begin{aligned}
\|\hat{\mu}_r(X) - \mu_r(X)\|_{L^2(P)} &\leq \big\|\big(\hat{V}(X)^{r/2} - V_0(X)^{r/2}\big)\mu_r^\star(X)\big\|_{L^2(P)} \\
&\leq rC_{\mathsf{v}}^{r/2-1}C_{\eta^\star}^r\epsilon_{\mathsf{v}}.
\end{aligned}$$

Via a similar reasoning as in Lemma F.1, we can deduce by induction that

$$|\hat{\kappa}_j(X)| \leq \big(2jC_{\mathsf{v}}^{1/2}C_{\eta^\star}\big)^j, \quad \|\hat{\kappa}_j(X) - \kappa_j(X)\|_{L^2(P)} \leq \big((2j+1)C_T + 1\big)^j\epsilon_{\mathsf{v}} \leq \big(4jC_{\mathsf{v}}^{1/2}C_{\eta^\star}\big)^j,$$

which then implies an upper bound on the error of $\hat{a}_{ik}(\cdot)$:

$$\|a_{ir} - \hat{a}_{ir}\|_{L^2(P)}$$

$$= \left\|\frac{1}{(r+1-i)!i!} \sum_{\pi \in \Pi_{r+1-i}} (-1)^{|\pi|-1} \left(\prod_{B \in \pi} \kappa_{|B|}(X) - \prod_{B \in \pi} \hat{\kappa}_{|B|}(X)\right)\right\|_{L^2(P)}$$

$$\leq \frac{1}{(r+1-i)!i!} \sum_{\pi \in \Pi_{r+1-i}} \left\|\prod_{B \in \pi} \kappa_{|B|}(X) - \prod_{B \in \pi} \hat{\kappa}_{|B|}(X)\right\|_{L^2(P)}$$

$$\leq \frac{1}{(r+1-i)!i!} \sum_{\pi \in \Pi_{r+1-i}} \sum_{B \in \pi} \left\|\kappa_{|B|} - \hat{\kappa}_{|B|}\right\|_{L^2(P)} \cdot \sup_{x \in \mathcal{X}} \prod_{B' \in \pi \setminus \{B\}} \max\left\{|\kappa_{|B'|}(x)|, |\hat{\kappa}_{|B'|}(x)|\right\}$$

$$\leq \frac{1}{(r+1-i)!i!} \epsilon_{\mathsf{v}} \sum_{\pi \in \Pi_{r+1-i}} \sum_{B \in \pi} \left(4|B|C_{\mathsf{v}}^{1/2}C_{\eta^\star}\right)^{|B|} \prod_{B' \in \pi \setminus \{B\}} 2\left(2|B|C_{\mathsf{v}}^{1/2}C_{\eta^\star}\right)^{|B|}$$

$$\leq \frac{1}{(r+1-i)!i!} 2\left(2C_{\mathsf{v}}^{1/2}C_{\eta^\star}\right)^{r+1-i} \epsilon_{\mathsf{v}} \sum_{j=1}^{r+1-i} 4^j \sum_{(i_1,\cdots,i_j) \in \mathrm{P}_{r+1-i,j}} \binom{r+1-i}{i_1,\cdots,i_j} \prod_{s=1}^{j} i_s^{i_s}$$

$$\leq \frac{2}{i!}\left(2C_{\mathsf{v}}^{1/2}C_{\eta^\star}\right)^{r+1-i} \epsilon_{\mathsf{v}} \sum_{j=1}^{r+1-i} 4^j e^{r+1-i} p(r+1-i, j)$$

$$\leq \left(8eC_{\mathsf{v}}^{1/2}C_{\eta^\star}\right)^{r+1-i} \frac{r+1-i}{i!}\epsilon_{\mathsf{v}},$$

where we follow the arguments employed in [Lemma G.4](#).

*Proof of part (2).* This is a direct consequence of [Theorem I.1](#) and part (1).

## I.5    Proof of [Corollary I.1](#)

Let $\rho = \min\{\alpha, \gamma\} > \frac{1}{4}$, then we have $\|\hat{g} - g_0\|_{P_0,\infty}, \|\hat{q} - q_0\|_{P_0,\infty} = \tilde{\mathcal{O}}_P(n^{-\frac{\rho}{2\rho+1}})$, so that $\epsilon_1, \epsilon_2 = \tilde{\mathcal{O}}_P(n^{-\frac{\rho}{2\rho+1}}) = o_P(n^{-1/6})$.

Also [Proposition I.1](#) implies that

$$\max\{\epsilon_4, \tilde{\epsilon}\} = \mathcal{O}\left(n^{-\min\{2\alpha, \beta/(2\beta+1)\}}\right).$$

It remains to show that

$$\max\{\epsilon_1, \epsilon_2\} \cdot \max\{\epsilon_4, \tilde{\epsilon}\} = \mathcal{O}(n^{-1/2}) \Leftarrow \frac{\rho}{2\rho+1} + \min\left\{2\alpha, \frac{\beta}{2\beta+1}\right\} > \frac{1}{2}.$$

Since $2\alpha > \frac{1}{2}$, it remains to check that

$$\frac{\rho}{2\rho+1} + \frac{\beta}{2\beta+1} > \frac{1}{2} \quad \Leftrightarrow \beta\rho > \frac{1}{4}$$

which holds by assumption. This proves (1).

To prove (2), we use a similar argument except that the upper bound becomes

$$|\hat{\theta} - \theta_0| = \mathcal{O}_P\left(\epsilon_1^4 + \epsilon_1^3\epsilon_2 + \max\{\epsilon_1, \epsilon_2\}(\epsilon_4^2 + \tilde{\epsilon})\right).$$

Since $\rho \geq \frac{\sqrt{3}-1}{4} > \frac{1}{6}$, we have $\max\{\epsilon_1, \epsilon_2\}^4 = \mathcal{O}(n^{-\frac{4\rho}{2\rho+1}}) = \mathcal{O}(n^{-1/2})$. Moreover, the assumption guarantees that

$$\frac{\rho}{2\rho+1} + \min\left\{2\alpha, \frac{\beta}{2\beta+1}\right\} \geq \frac{\rho}{2\rho+1} + \min\left\{2\rho, \frac{\beta}{2\beta+1}\right\} \geq \frac{1}{2}$$

since $\frac{\sqrt{3}-1}{4}$ is the positive root of the equation $\frac{\rho}{2\rho+1} + 2\rho = \frac{1}{2}$. This concludes the proof.

## I.6 Comparison with Mackey et al. [2018]

In Mackey et al. [2018] the authors consider polynomial-based moment functions. These constructions can be derived from our Theorem I.1 by choosing $J_1(w, x) = w^k - \mu_k(x)$ where $k$ is some positive integer and $\mu_k(x) = \mathbb{E}[\eta^k \mid X = x]$. For $r = 2, 3$, we obtain the following special cases:

**Example I.1.** *By choosing $r = 2$, we recover the moment function*

$$m\big(Z, \theta, q(X), g(X), \{\mu_i(X)\}_{i=k}^{k+1}\big)$$
$$= \big[Y - q(X) - \theta(T - g(X))\big]\big[(T - g(X))^{k+1} - \mu_{k+1}(X) - (k+1)(T - g(X))\mu_k(X)\big]$$
(144)

*proposed by Mackey et al. [2018]. Thus, Theorem I.1 implies that this moment function satisfies all conditions in Assumption E.1 with the orthogonality set $S = \{\|\alpha\|_1 \leq 2\} \setminus \{(1, 0, 0, 1), (0, 1, 0, 1)\}$.*

**Example I.2.** *By choosing $r = 3$, we obtain*

$$m\big(Z, \theta, q(X), g(X), \mu_2(X), \{\mu_i(X)\}_{i=k}^{k+2}\big)$$
$$= \big[Y - q(X) - \theta(T - g(X))\big] \times \big[(T - g(X))^{k+2} - \mu_{k+2}(X) - (k+2)(T - g(X))\mu_{k+1}(X)$$
$$- (k+1)(k+2)\big((T - g(X))^2 - \mu_2(X)\big)\mu_k(X)/2\big]$$
(145)

*with nuisance functions $h(X) = (q(X), g(X), \mu_2(X), \mu_k(X), \mu_{k+1}(X), \mu_{k+2}(X)) \in \mathbb{R}^6$ is orthogonal with respect to $S = \{(a, b, 0, 0, 0, 0) \mid a + b = 3\} \cup \{\alpha \mid \|\alpha\|_1 \leq 2\} \setminus \{(a, b, c, d, 0, e) \in \mathbb{Z}_{\geq 0}^6 \mid a + b = c + d + e = 1\}$.*

## I.7 More discussion on Assumption E.1

The following result states that if the distribution of $\eta \mid X = x$ does not depend on $x$ and its density has certain good properties, then Assumption E.1 would not be violated with $J_2$, unless $\eta$ is Gaussian.

**Proposition I.2** (Identifiability v.s. orthogonality). *The moment function $m$ in Theorem I.1 (2) satisfies Assumption E.1 if and only if*

$$\mathbb{E}\left[(T - g_0(X))J_r(T - g_0(X), X)\right] \neq 0.$$

*Moreover, the following statements hold:*

*(1). If $\eta \mid X = x$ is Gaussian for all $x \in \mathcal{X}$, then $\mathbb{E}\left[(T - g_0(X))J_r(T - g_0(X), X)\right] = 0, \forall r \geq 2$.*

*(2). If $\eta \mid X = x$ is non-Gaussian with twice continuously differentiable density $p(\cdot)$ that does not depend on $x$, then there exists $J_1(w, x)$ in the form of (7), such that $\mathbb{E}[(T - g_0(X))J_3(T - g_0(X), X)] \neq 0$ and $\mathbb{E}[(1 + |\eta|)|\rho_{i2}(\eta)|] < +\infty, \forall 1 \leq i \leq M_2$.*

*(3). Suppose that $\eta \perp\!\!\!\perp X$ and $\eta$ is non-Gaussian, and let $\{J_r\}_{r=1}^{+\infty}$ be the sequence generated from $J_1(w, x) = w$ (assuming that all of them are well-defined). Then there exists $r \geq 2$ such that $\mathbb{E}[(T - g_0(X))J_r(T - g_0(X), X)] \neq 0$.*

The first part of Proposition I.2 can be directly derived from the Stein's Lemma, while the second part is derived from a characterization of the solutions property of the Gauss-Airy's equation $xy - ay' - by'' = 0$. [Durugo, 2014, Ansari, 2016] Generalizing this result to $k \geq 3$ requires characterizing the solution properties of higher-order Gauss-Airy's equation, which we leave for future work.

***Proof* :** (1) is straightforward from Stein's lemma. To prove (2), orthogonality implies that

$$0 = \mathbb{E}\left[D_q m(Z, \theta_0, h_0(X)) \mid X\right] = -\mathbb{E}\left[J_3(T - g_0(X), X) \mid X\right],$$
(146)

and

$$0 = \mathbb{E}\left[D_{qg} m(Z, \theta_0, h_0(X)) \mid X\right] = \mathbb{E}\left[J_3'(T - g_0(X), X) \mid X\right],$$
(147a)
$$0 = \mathbb{E}\left[D_{qgg} m(Z, \theta_0, h_0(X)) \mid X\right] = -\mathbb{E}\left[J_3''(T - g_0(X), X) \mid X\right],$$
(147b)

where we slightly abuse notation and use $J_3', J_3''$ to represent the partial derivatives with respect to the first argument.

Note that $m$ is partially linear in $\theta$, we have

$$m(Z, \theta, h_0(X)) = \theta \cdot \nabla_\theta m(Z, \theta_0, h_0(X)) + (Y - q_0(X))J_3(T - g_0(X), X),$$

so (2) and (3) are both equivalent to

$$\mathbb{E}\left[\nabla_\theta m(Z, \theta_0, h_0(X))\right] \neq 0 \quad \Leftrightarrow \quad \mathbb{E}\left[(T - g_0(X))J_3(T - g_0(X), X)\right] \neq 0. \tag{148}$$

We argue that there must exist some $J_3(w, x)$ that is polynomial in $w$, such that the equations (146), (147) and (148) hold simultaneously. Since by assumption, the density $p(\eta)$ of $\eta = T - g_0(X) \mid X$ does not depend on $X$, (147a) is equivalent to

$$0 = \int J_3'(w, x)p(w)\mathrm{d}w = -\int J_3(w, x)p'(w)\mathrm{d}w \quad \Leftrightarrow \quad \int J_3(w, x)p'(w)\mathrm{d}w = 0,$$

and similarly, (147b) is equivalent to

$$\int J_3(w, x)p''(w)\mathrm{d}w = 0.$$

Lemma I.1 at the end of this subsection implies that in $L^2(\mathbb{R})$, $wp(w) \notin \mathrm{span}\langle p'(w), p''(w)\rangle$, so by Lemma I.2, there must exists some function $\tilde{J}_3(w) \in \mathcal{C}^3(\mathbb{R})$ such that

$$\int \tilde{J}_3(w)p'(w)\mathrm{d}w = \int \tilde{J}_3(w)p''(w)\mathrm{d}w = 0 \quad \text{and} \quad \int \tilde{J}_3(w)wp(w)\mathrm{d}w > 0. \tag{149}$$

Moreover, we can assume WLOG that $\int \tilde{J}_3(w)p(w)\mathrm{d}w = 0$, since replacing $\tilde{J}_3$ with $\tilde{J}_2 - c_0, \forall c_0 \in \mathbb{R}$ does not affect the properties in (149). We define $\tilde{J}_2(w) = \tilde{J}_3'(w)$ and $\tilde{J}_1(w) = \tilde{J}_3''(w)$. In the following, we show that $J_1(w, x) = \tilde{J}_1(w)$ satisfies all the desired properties. First, since $\tilde{J}_3$ is a polynomial, so is $\tilde{J}_1$. Second, by (147b) we have

$$\mathbb{E}\left[\tilde{J}_1(T - g_0(X), X) \mid X\right] = \mathbb{E}\left[\tilde{J}_3''(T - g_0(X), X) \mid X\right] = 0.$$

Finally, recall that $p(\cdot)$ is the probability density function of $\eta = T - g_0(X)$, so

$$\mathbb{E}\left[(T - g_0(X))\tilde{J}_3(T - g_0(X))\right] = \int \tilde{J}_3(w)wp(w)\mathrm{d}w > 0,$$

concluding the proof of (2).

Finally, under the conditions of (3), Lemma G.6 implies that $\mathbb{E}[(T - g_0(X))J_r(T - g_0(X), X)] = \frac{1}{r!}\kappa_{r+1}$, where $\kappa_i$ is the $i$-th order cumulant of $\eta$. We know from Levy's Inversion Formula [Durrett, 2019, Theorem 3.3.11] that non-Gaussian distributions must have at least one non-zero cumulant, so the conclusion immediately follows. $\qquad\square$

**Lemma I.1** (Solution of second-order Airy equation). *Let $p(\cdot)$ be the probability density function of a random variable $\eta$ that is second-order continuous differentiable, and that*

$$xp(x) + 2ap'(x) + bp''(x) = 0 \tag{150}$$

*for some $a, b \in \mathbb{R}$, then we must have $b = 0, a > 0$ and thus $\eta$ must be Gaussian.*

**_Proof_** : Since $p(\cdot)$ is a density function, the Riemann–Lebesgue lemma implies that its Fourier transform $\hat{p}(\xi) = \int_\mathbb{R} e^{-ix\xi}p(x)\mathrm{d}x$ must vanish at infinity. On the other hand, applying Fourier transform to both sides of (150) yields

$$i\hat{p}'(\xi) + 2ai\xi\hat{p}(\xi) - b\xi^2\hat{p}(\xi) = 0 \quad \Rightarrow \quad \hat{p}(\xi) = Ce^{-a\xi^2 - ib\xi^3/3}.$$

Thus we must have $a > 0$. If $b \neq 0$, [Ansari, 2016, 4.2] then implies that there exists constants $c_1, c_2 \in \mathbb{R}$ such that $p(x)$ has the same sign as the Airy function $\mathrm{Ai}(c_1x + c_2)$. However, it is well-known that $\mathrm{Ai}(x)$ can take both positive and negative values, which is a contradiction. $\qquad\square$

**Lemma I.2** (Separability of inner products). *Suppose that $f_i(w), i = 0, 1, 2$ are continuous functions such that $f_0 \notin \text{span}\langle f_1, f_2 \rangle$. Then there exists a function $J(w) \in \mathcal{C}^3(\mathbb{R})$ such that*

$$\int |J(w)f_i(w)| \, \mathrm{d}w < +\infty, \quad i = 0, 1, 2$$

*and*

$$\int J(w)f_i(w)\mathrm{d}w = 0, i = 1, 2 \quad and \quad \int J(w)f_0(w)\mathrm{d}w > 0.$$

***Proof*** : Suppose that such $J(w)$ does not exist. For any finite interval $[a, b]$ and a sequence of $\mathcal{C}^3$ functions $S = \{g_j, j = 1, 2, \cdots, n\}$ supported on $[a, b]$ (we denote the set of such functions by $\mathcal{C}_0^3([a, b])$), we define the vector $u_i^S = \left(\int g_r(w)f_i(w)\mathrm{d}w\right)_{k=0}^{+\infty}, i = 0, 1, 2$. Then for any $\lambda \in \mathbb{R}^{|S|}$, by choosing $J(w) = \sum_{i=1}^n \lambda_i g_i(w)$, our assumption implies that

$$\lambda^\top u_1^S = \lambda^\top u_2^S = 0 \quad \Rightarrow \quad \lambda^\top u_0^S = 0.$$

Let $v_0^S$ be the orthogonal projection of $u_0^S$ onto $\text{span}\langle u_1^S, u_2^S \rangle$ and $\lambda = u_0^S - v_0^S$, then $\lambda^\top u_1^S = \lambda^\top u_2^S = 0$ and $\lambda^\top u_0^S = \|\lambda\|^2$, so that $\lambda = 0$ and $u_0^S \in \text{span}\langle u_1^S, u_2^S \rangle$.

We consider the following two cases:

1. For any finite $S$, $\text{rank}\left(\text{span}\langle u_1^S, u_2^S \rangle\right) \leq 1$, then $u_0^S$ is parallel to $u_1^S$ for all $S$. Since $u_1 \neq 0$, it is easy to show that $u_0 = \alpha u_1$ for some $\alpha \in \mathbb{R}$. As a result, we have $\int J(w)(f_0(w) - \alpha f_1(w))\mathrm{d}w = 0$ for any $J(\cdot) \in \mathcal{C}_0^3([a, b])$, so we must have $f_0 - \alpha f_1 \equiv 0$ on $[a, b]$.

2. There exists some finite $S$ such that $\text{rankspan}\langle u_1^S, u_2^S \rangle = 2$, then there exists unique $\alpha, \beta \in \mathbb{R}$ such that $u_0^S = \alpha u_1^S + \beta u_2^S$. By considering any set $S_1 = S \cup \{s\}, s \notin S$, one can show that $u_0^{S_1} = \alpha u_1^{S_1} + \beta u_2^{S_1}$ as well, so $u_0^S = \alpha u_1^S + \beta u_2^S$ for any finite set $S$. As a result, we have $\int J(w)(f_0(w) - \alpha f_1(w) - \beta f_2(w))\mathrm{d}w = 0$ for any $J(\cdot) \in \mathcal{C}_0^3([a, b])$, so we must have $f_0 - \alpha f_1 - \beta f_2 \equiv 0$ on $[a, b]$.

Now we have shown that for any interval $[a, b]$, there exists $\alpha, \beta \in \mathbb{R}$ such that $f_0 - \alpha f_1 - \beta f_2 \equiv 0$ on $[a, b]$. It is easy to derive from this fact that $f_0 - \alpha f_1 - \beta f_2 \equiv 0$ on $\mathbb{R}$, *i.e.*, $f_0 \in \text{span}\langle f_1, f_2 \rangle$, which is a contradiction. Hence the conclusion follows. $\square$

