# OpenReview forum: "It’s Hard to Be Normal: The Impact of Noise on Structure-agnostic Estimation"
_NeurIPS.cc/2025/Conference — NeurIPS 2025 poster_

### Official Review · Reviewer_7tt3 · 2025-06-04

**Clarity:** 2
**Significance:** 3
**Originality:** 3
**Rating:** 4
**Confidence:** 2

**Summary:**

This paper focuses on a particular estimation task: Estimating the treatment effect in the partially linear outcome model of Robinson [1988], where the treatment effect is a constant across all individuals.  In this setting, this paper introduces a few contributions: First, it introduces new lower-bounds on the minimax estimation error in the framework of structure-agnostic estimation.  It shows that in two settings (binary and Gaussian treatment) these lower-bounds match the upper bounds that already exist for DML-style estimators up to constant factors.  Second, this paper introduces "agnostic cumulant-based estimation" (ACE) which gives robustness to higher-order errors in non-Gaussian settings, and illustrates the use of this approach on synthetic data.

**Questions:**

A few points in priority order.

1. Regarding weakness #1: How robust is the theoretical performance of ACE to the assumption that the data-generating process follows the PLR setting?  What can we say about the performance of ACE in the more general setting considered in [Jin & Syrgkanis 2024] when using $l > 4$?  It is not strictly necessary for the authors to answer those particular questions, as they may be out-of-scope for this paper, but I would appreciate some additional discussion on this point: Getting more clarity on this point, even if the conclusion is "it's not clear that ACE is robust" would be helpful here.

2. Regarding weakness #2: Is ACE minimax-optimal, or can you characterize settings where it (theoretically) performs better than the DML estimator?  Some discussion on this point would be helpful...if some discussion already exists on this point in the paper (or appendix), could you point me to it?

3. Regarding weakness #3: Not a question, but a suggestion---I would have liked to see an experiment comparing DML against the "end-to-end" ACE procedure (including data-driven selection of $l$ as suggested in Appendix J).

4. Finally, (copied from above): I was a bit confused trying to understand Theorem 3.1. It seems that the lower-bound includes a term $c_q \epsilon_1^2$ as well as the product of errors $\epsilon_1 \epsilon_2$.  Am I correct to read this equation as implying that, if $\epsilon_2 = O(n^{-1/2})$ but $\epsilon_1 = O(1)$ (i.e., if the propensity score model is miss-specified), then the minimax estimation error does not go to zero?  That would seem counter-intuitive, in light of the "double-robustness" of DML methods like AIPW, which is consistent if either the outcome or propensity model is consistent.

**Ethical Concerns:**

["NO or VERY MINOR ethics concerns only"]

**Final Justification:**

I increased my score from 3 -> 4.

In my original review, I highlighted three weaknesses, in order of priority, and discuss here how my opinion has changed:
(1) Significance of PLR setting:  The authors have convinced me that the setting is of sufficiently general interest that this concern should not be load-bearing in accept/reject decisions.
(2) Unclear (theoretical) comparison to DML:  The authors gave a very clear answer to this question in the rebuttal, and my concerns are addressed.
(3) Unclear (empirical) comparison to DML:  While I would still like to see the "end-to-end" comparison to DML, I understand that's outside the scope of the rebuttal process, and that the main contribution of the paper is theoretical, rather than empirical.

**Limitations:**

I think the paper could be a bit clearer when it comes to limitations, primarily in the context of my Weakness #1 above.

**Quality:**

3

**Strengths And Weaknesses:**

## Strengths

In terms of originality, the results seem interesting and novel, although I'm not sure of the significance for practitioners (see weakness #1 below).  I particularly appreciated the results regarding the minimax optimality of DML in the binary and Gaussian settings.  The inclusion of simulations was helpful for demonstrating that the proposed method out-performs DML when including higher-order terms ($l = 5$) both in terms of bias and MSE.

Caveat: I did not verify the proofs.  Moreover, while I am familiar with double ML, semiparametrics, etc, I am less familiar with the more recent line of work on structure-agnostic estimation.  While I have skimmed the references in this work, it is possible that I am missing other recent context from the literature.

## Weaknesses

I have ordered these weaknesses in order of relative importance, and will summarize the most salient points as questions at the end of the review.

(1) **Significance of PLR setting**: The results in this paper apply in particular to the setting of partially-linear regression, where the treatment effect is homogeneous (the same for all individuals).  Moreover, the novelty of the results is tailored to this setting:  As noted by the authors, and as I understand it from my skim, [Jin & Syrgkanis 2024] establish the minimax optimality of DML in the more general (and realistic) setting where there is no homogeneity constraint.  Moreover, it is unclear to me how the ACE (with $l > 1$) and DML estimators compare in the setting where this assumption is violated.  To me, the assumption of homogeneous treatment effects is more worrisome than the assumption that $\eta$ and $X$ are independent.

(2) **Unclear (theoretical) comparison to DML**: I understand that when $l=1$, then ACE is equivalent to DML, and that ACE is novel in the case where $l > 4$ (see page 8, lines 206-209).  However, I found it difficult to see in Theorem 5.2 and elsewhere how the rates compare for different settings of $l$.  It is established that DML is **not** minimax optimal with non-Gaussian noise, but I didn't see a claim that ACE is minimax optimal, or a characterization of the settings where it provides a faster rate than DML. Some more discussion on this point would strengthen the paper.

(3) **Unclear (empirical) comparison to DML**: I appreciated the inclusion of simulations, and I agree that they show the method out-performs DML with $l = 5$, as noted in "strengths" above.  First, however, there is a lack of clarity in Section 6 of the main paper due to moving so much of the content in into the appendix [Appendix J, page 51, lines 1253-1273 in particular]. Some of these details are quite important for interpreting Figure 1.  For instance, it was not clear in the main text that Figure 1(a) **only shows those ACE procedures where the MSE is lower than DML**.  I would have liked to see an experiment comparing DML against the "end-to-end" ACE procedure (including data-driven selection of $l$ as suggested in Appendix J).

## Minor feedback

I was a bit confused trying to understand Theorem 3.1: It seems that the lower-bound includes a term $c_q \epsilon_1^2$ as well as the product of errors $\epsilon_1 \epsilon_2$.  Am I correct to read this equation as implying that, if $\epsilon_2 = O(n^{-1/2})$ but $\epsilon_1 = O(1)$ (i.e., if the propensity score model is miss-specified), then the minimax estimation error does not go to zero?  That would seem counter-intuitive, in light of the "double-robustness" of DML methods like AIPW, which is consistent if either the outcome or propensity model is consistent.

Other minor issues / questions:
* Unclear references in theorem statements:  Perhaps this is a personal preference, but I like for theorem statements to be relatively self-contained.  For instance, Theorem 5.2 contains the assumption that $\eta$ is independent of $X$, but requires the reader to remember that $\eta$ is defined in Equation (1), 7 pages earlier.  Likewise, $\Phi$ and $\Phi^*$ are defined in Definitions 2.2 and 3.1 respectively.
* In Algorithm 1, is it assumed that sample-splitting has already occurred?  Is it the idea that $\hat{q}$ and $\hat{g}$ were estimated on different data than $\mathcal{D}$?  If so that might be worth clarifying.

---

> ### Author Rebuttal · Authors · 2025-07-30
>
> We thank the reviewer for the positive and constructive feedback and are delighted that the reviewer found our work “interesting and novel.” We respond to each point raised below.
>
> **Weakness 1. The results in this paper apply in particular to the setting of partially-linear regression, where the treatment effect is homogeneous (the same for all individuals).**
>
> The reviewer is correct that the results of this work applies to the partially linear outcome model. In the revision, we will highlight that while the partially linear model is more restrictive than the fully non-parametric model, it is widely employed and applies to practical problems like demand estimation from purchase and pricing data (as studied in Mackey et al. and simulated in our Sec. 6 experiments). Here, $Y$ represents observed demand, the treatment $T$ corresponds to an observed product price, $g_0(X)$ denotes a baseline product price determined by covariates X that influence pricing policy, and the treatment noise $\eta$ represents a random discount offered to customers for demand assessment. Notably, $\eta$ is typically discrete (and thus distinctly non-Gaussian) and independent of $X$. In addition, [1, Section V] studied a partial linear outcome model where the outcome is the median property value of a county and the treatment is the TSP, a metric for air pollution, and the more recent work [2] used the same model to study the effect of income taxes on the property values.
>
> **Question 1.  How robust is the theoretical performance of ACE to the assumption that the data-generating process follows the PLR setting? What can we say about the performance of ACE in the more general setting considered in [1] when using $l>4$?**
>
> Thank you for this excellent question!  We will answer it from two different standpoints:
> - If we want to apply the same ACE estimator to data that might not satisfy the partial linearity assumption, say that the true outcome model is $$Y = g(X)+\Delta(X)T+\epsilon$$, and we want to estimate $\mathbb{E}[\Delta(X)]$, then the standard DML estimator would incur a bias $$\frac{Cov(Var(T|X),\Delta(X))}{\mathbb{E}[Var(T|X)]}.$$ For the ACE estimator, the second term in the moment equation becomes $J_k(T-m(X))$. Let $I_k(X) = \mathbb{E}[(T-m(X))J_k(T-m(X))|X]$, then the bias is instead $$\frac{Cov(I_k(X),\Delta(X))}{\mathbb{E}[I_k(X)]}.$$ As a result, there are two cases when the bias (of both DML and ACE) becomes zero. The first one is the PLR setting where $\Delta(X)$ is constant. The second is that when $\eta=T-m(X)$ is independent of $X$, then $I_k(X)$ becomes constant, while $\Delta(X)$ can be non-constant.
> We would like to point out that we were unaware of the second property of ACE, which might address the reviewer’s concern about the PLR assumption. However, we find that the higher-order robustness property no longer holds in this case.
> - If we want to design ACE-type estimators for the more general setting of [1], we must make additional assumptions on the model, since [1] shows that DML is already optimal without any further assumption. One possible assumption is the independent noise assumption; as noted in 1, this guarantees that ACE is asymptotically unbiased. However, it is unclear how and whether higher-order robustness can be achieved for this general model.
>
>
> **Question 2. Is ACE minimax-optimal, or can you characterize settings where it (theoretically) performs better than the DML estimator? Some discussion on this point would be helpful...if some discussion already exists on this point in the paper (or appendix), could you point me to it?**
>
> By theorem 5.2, ACE estimator performs better than DML in terms of the dependency on the nuisance estimation errors $\epsilon_i, i=1,2$. Specifically, the error is $O(\epsilon_1^l\epsilon_2+\epsilon_1^{l+1})$, improving upon DML’s rate $\epsilon_1\epsilon_2+\epsilon_1^2$. The price to pay is a possibly larger constant, which now depends on the magnitude of $(l+1)$-th cumulant of the treatment noise as well as some super-exponential terms in $l$ (as we noted in Remark 5.2, however, the dependency can be only exponential in some cases). Therefore, as the sample size goes to infinity, $\epsilon_i \to 0, i=1,2$ and ACE with order $l\geq 2$ would eventually outperform DML.
>
> The optimal choice of $l$ is discussed in Remark 5.3 and formally stated in eq.(15). Roughly speaking, when the nuisance estimation errors $\epsilon_i,i=1,2$ are polynomial in $n$, then one can choose the order of orthogonality $l$ to be $O(1)$ while making the bias to be $O(n^{-1/2})$. Hence, the final error rate would be $O(n^{-1/2})$. This is the well-known theoretically optimal rate given noisy observations. Therefore, the ACE estimator is minimax optimal up to constant.
>
> We will add more explanations to make these points more transparent in an updated version.
>
> **Question 3. First, however, there is a lack of clarity in Section 6 of the main paper due to moving so much of the content into the appendix [Appendix J, page 51, lines 1253-1273 in particular]. Some of these details are quite important for interpreting Figure 1. For instance, it was not clear in the main text that Figure 1(a) only shows those ACE procedures where the MSE is lower than DML. I would have liked to see an experiment comparing DML against the "end-to-end" ACE procedure (including data-driven selection as suggested in Appendix J).**
>
> We thank the reviewer for the suggestions. We will definitely make the experiment part of the main paper more self-contained in the revised version. Also, we will illustrate an end-to-end ACE procedure to compare with DML.
>
> **Question 4. I was a bit confused trying to understand Theorem 3.1. It seems that the lower-bound includes a term as well as the product of errors . Am I correct to read this equation as implying that, if $\epsilon_2=O_P(n^{-1/2})$  but $\epsilon_1=O(1)$  (i.e., if the propensity score model is miss-specified), then the minimax estimation error does not go to zero? That would seem counter-intuitive, in light of the "double-robustness" of DML methods like AIPW, which is consistent if either the outcome or propensity model is consistent.**
>
> This is a fantastic question! The reviewer’s confusion might be caused by different parameterizations of the causal model. As pointed out in [4, Section 4.1], there exists two different debiasing schemes for the PLR setting. The first uses the moment function $$(Y-\theta T - f(X))(T-g(X))=0$$ while the second is what we used. Here $f(X)=\mathbb{E}[Y\mid T=0, X]$. Although in [4], the authors did not explicitly state the rates in terms of the nuisance errors, the presence of an $\epsilon_1^2$ term can be seen from their Assumption 4.1(b), which is in contrast with a “truly doubly robust” rate in Assumption 4.1(a) for the alternative parameterization.
>
> Here are some explanations of why seemingly different rates are obtained. A notable feature of PLR is that, given either pair of nuisances $(f,g)$ and $(q,g)$, the model is not uniquely specified. This means that these two parameterizations are not equivalent. Moreover, the relationship of these three nuisances is given by $$f(X)=q(X)-\theta g(X)$$. Even if we can estimate $\theta$ accurately (e.g. via DML), having $\epsilon_1$ error for $g$ and $\epsilon_2$ error for $q$ would result in an $O(\epsilon_1+\epsilon_2)$ error for $f$. Plugging into the “doubly robust” rate we obtain $$\text{error of } g \times \text{error of } f = O(\epsilon_1(\epsilon_1+\epsilon_2))$$ precisely matching the bound we have. So, having a doubly robust error rate does not contradict with the lower bound we establish here.
>
> From a technical point of view, the reason why $\epsilon_1^2$ appears in the bound, at least for DML, is that the variance $\mathbb{E}[(T-g(X))^2]$ (see eq.(2)) needs to be accurately estimated. If the estimation error of $g$ is too large, then this step would fail.
>
> We hope that our response addresses all questions that the reviewer has.
>
> [1]. Jin, Jikai, and Vasilis Syrgkanis. "Structure-agnostic optimality of doubly robust learning for treatment effect estimation." arXiv preprint arXiv:2402.14264 (2024).
>
> [2]. Chay, Kenneth Y., and Michael Greenstone. "Does air quality matter? Evidence from the housing market." Journal of political Economy 113.2 (2005): 376-424.
>
> [3]. Hull, Isaiah, and Anna Grodecka-Messi. "Measuring the impact of taxes and public services on property values: a double machine learning approach." arXiv preprint arXiv:2203.14751 (2022).
>
> [4]. Chernozhukov, Victor, et al. "Double/debiased machine learning for treatment and structural parameters." The Econometrics Journal (2018): C1-C68.
>
> [5]. Robinson, Peter M. "Root-N-consistent semiparametric regression." Econometrica: journal of the Econometric Society (1988): 931-954.

---

> > ### Comment · Reviewer_7tt3 · 2025-08-02
> > **Thank you**
> >
> > Thank you for the thorough and clear response to my questions - these responses are very helpful and I would say that they've sufficiently addressed my concerns.  I will increase my score accordingly.
> >
> > As space permits, I would encourage the authors to include some of these explanations in the next revision of the paper (or at least in the appendix).  In addition, the explanation in the response to reviewer 9YPQ ("Weakness 1") is also very helpful, and I would encourage the authors to include it in some form in the next revision.

---

> > > ### Author Response · Authors · 2025-08-07
> > >
> > > We thank the reviewer for the positive feedback, and will follow the reviewer's suggestions in the revised version of our paper.

---

### Official Review · Reviewer_y6th · 2025-06-16

**Clarity:** 4
**Significance:** 4
**Originality:** 4
**Rating:** 5
**Confidence:** 4

**Summary:**

The authors investigate the problem of treatment effect estimation in causal inference, specifically studying a partially linear model. A popular choice in the literature is double machine learning (DML), in which case the nuisance parameters (e.g. treatment propensity and covariate effects on the outcome) are estimated via black-box ML algorithms and plugged-in. The authors show DML is optimal under Gaussian treatment noise. However, for non-Gaussian noise, the authors resort to constructing new, higher-order, cumulant-based estimators. The authors also derive minimax lower bounds for binary treatments.

**Questions:**

__Major__
- I have a high-level confusion about a comment made in Section 5.1. The authors note that the fast rate of Theorem 5.1 holds specifically for estimating cumulants, and that estimation of moments runs into a bottleneck (stated in Theorem H.1). At a high-level, moments and cumulants are in 1-1 correspondence (explicit formulas given in terms of Bell polynomials). Naively, I would have the impression that if there is no bottleneck for cumulant estimation, then there ought to be no bottleneck for moment estimation. Can the authors provide high-level comments why this naive impression is incorrect? It would also be nice to elaborate in the paper.

- Related to the previous comment, I see that the cumulant estimators in Theorem 5.1 are essentially just plug-in estimators (plug in $\hat{g}$ and then compute cumulants of the empirical residual distribution). Plug-in is sensible to me since we are in a structure-agnostic setting. But why can't one do the same for moment estimation? Why does the bottleneck appear for moments but not cumulants?

- In the proof of Theorem 3.1, it would be nice if the authors could provide some high-level discussion about what is the non-trivial part of constructing the set of hypotheses for using the method of fuzzy hypothesis (which they claimed right after the statement of Theorem 3.1).

- The lower bound construction of Theorem 3.2 involves moment matching, which is a technique that has been successfully used in other problems (especially, functional estimation problems). In these other settings, it is very commonly the case that the optimal estimator turns out to be based on polynomial-approximation ideas (i.e. a method-of-moments estimator). In fact, there are duality arguments in some settings which immediately give a certificate of optimality (see Wu and Yang, "Polynomial methods in statistical inference: theory and practice"). In this paper, the authors go for a cumulant-based estimator, instead of moments. Could their moment-matching lower bound construction, paired with the duality perspective, yield a moment-based estimator? Also, since the lower bound is based on moment-matching, does that imply that the moment estimation bottleneck the authors identify is not actually fundamental, but just an artifact of analysis?

- In the abstract, the authors claim DML is always suboptimal for independent non-Gaussian treatment noise, but in Section 3.1 they show optimality of DML for binary treatment?

- In Section 5, the authors claim Theorem 4.1 gives better rates than DML. But I don't see that they provided lower bounds specifically for DML that are worse than the upper bounds for ACE. Can they provide these?

 __Minor__
- On page 4, should "Gaussianility" be "Gaussianity"?
- Theorem E.1 has a broken hyperlink.

**Ethical Concerns:**

["NO or VERY MINOR ethics concerns only"]

**Final Justification:**

The authors addressed my questions, and I retain my score.

**Limitations:**

Yes

**Paper Formatting Concerns:**

Please check again for broken hyperlinks (e.g. in Theorem E.1).

**Quality:**

4

**Strengths And Weaknesses:**

The paper is quite strong in my opinion. DML is widely used, and the authors address a fundamental question about it, and obtain an interesting result which comments on a curiosity identified in the earlier work of Mackey. The paper is original, the writing and exposition is clear, and the mathematical execution is impressive. In my opinion, there are no major weaknesses.

---

> ### Author Rebuttal · Authors · 2025-07-30
>
> We thank the reviewer for the positive and constructive feedback and are elated that the reviewer found the paper “original” and “quite strong”, the writing and exposition clear, and the mathematical execution “impressive.” We respond to each point raised below.
>
> **Question 1. I have a high-level confusion about a comment made in Section 5.1. The authors note that the fast rate of Theorem 5.1 holds specifically for estimating cumulants, and that estimation of moments runs into a bottleneck (stated in Theorem H.1). At a high-level, moments and cumulants are in 1-1 correspondence (explicit formulas given in terms of Bell polynomials). Naively, I would have the impression that if there is no bottleneck for cumulant estimation, then there ought to be no bottleneck for moment estimation. Can the authors provide high-level comments why this naive impression is incorrect? It would also be nice to elaborate in the paper.**
>
> The most direct explanation for the differing error rates between moments and cumulants lies in the fact that when cumulants are expressed in terms of moments, the leading terms of the moment estimation error cancel out, leaving only higher-order residuals.
>
> The formal justification of this property of cumulants can be found on page 44 of the paper, starting from line 1140. The key observation is that the residual of nuisance estimates contributes additively to the cumulant generating function of the treatment noise, due to their assumed independence.
>
> **Question 2. In the proof of Theorem 3.1, it would be nice if the authors could provide some high-level discussion about what is the non-trivial part of constructing the set of hypotheses for using the method of fuzzy hypothesis (which they claimed right after the statement of Theorem 3.1).**
>
> The method of fuzzy hypothesis requires constructing two sets of distributions that are difficult to distinguish. In view of standard tools for applying this method (see e.g., Theorem 2.1), it is important to achieve the following simultaneously:
> The mixtures of the two groups of distributions are the same. This property is guaranteed by appropriately scaling the “bump functions” used to perturb the nuisances. We verify this property in eq.(24).
> These two groups yield well-separated values of the parameter of interest (which is the $\theta$ in our setting, with separation distance being the final lower bound that we want to prove.
>
> To the best of our knowledge, the only lower bounds for the structure-agnostic setting we consider are established in [2,3], for several different causal parameters. The constructions for these different parameters are established in a highly case-by-case manner.
>
> **Question 3. The lower bound construction of Theorem 3.2 involves moment matching, which is a technique that has been successfully used in other problems (especially, functional estimation problems). In these other settings, it is very commonly the case that the optimal estimator turns out to be based on polynomial-approximation ideas (i.e. a method-of-moments estimator). In fact, there are duality arguments in some settings which immediately give a certificate of optimality (see Wu and Yang, "Polynomial methods in statistical inference: theory and practice"). In this paper, the authors go for a cumulant-based estimator, instead of moments. Could their moment-matching lower bound construction, paired with the duality perspective, yield a moment-based estimator? Also, since the lower bound is based on moment-matching, does that imply that the moment estimation bottleneck the authors identify is not actually fundamental, but just an artifact of analysis?**
>
> We thank the reviewer for connecting our results with the duality approach in the literature. Here, the lower bound that applies moment matching technique is for the case of Gaussian treatment. In this case, we show that DML is already minimax optimal up to logarithmic factors (Theorem 3.2). DML can be viewed as the simplest moment-based estimator in our setting. By contrast, cumulant-based estimators are designed only for the non-Gaussian treatment setting, because all cumulants of a Gaussian distribution are zero, which would violate the identifiability assumption.
>
> **Question 4. In the abstract, the authors claim DML is always suboptimal for independent non-Gaussian treatment noise, but in Section 3.1 they show optimality of DML for binary treatment?**
>
> We will clarify in the revision that both of these claims hold simultaneously as the sub-optimality result concerns only distributions with independent treatment noise $\eta$, while the binary minimax lower bound allows for the treatment noise $\eta$ to depend on $X$.
>
> **Question 5. In Section 5, the authors claim Theorem 4.1 gives better rates than DML. But I don't see that they provided lower bounds specifically for DML that are worse than the upper bounds for ACE. Can they provide these?**
>
> That’s a great point – indeed, although we proved lower bounds for any estimator in the binary and Gaussian treatment settings, we did not prove lower bounds for DML under the assumptions of Section 5.1, i.e., independent non-Gaussian rate. Nonetheless, it is not hard to to see that in this setting, DML still has the same rate $\Omega(\epsilon_1^2+\epsilon_1\epsilon_2+n^{-1/2})$ even under these additional assumptions. Specifically, one can start from the equation following line 778, which provides an expansion of the error induced by the DML estimator. In the worst case, both $\Delta_g$ and $\Delta_q$ can be arbitrary functions with $L_2$-norm bounded by $\epsilon_1$ and $\epsilon_2$ respectively. Thus in the worst case, the second term can be $\Omega(\epsilon_1^2+\epsilon_1\epsilon_2)$ while the first term is $\mathbb{E}[\eta^2]+o_P(1)$. It follows that  DML could incur an $\Omega(\epsilon_1^2+\epsilon_1\epsilon_2)$ error in the worst case.
>
> Theorem 5.1, by contrast, gives upper bounds for ACE that have strictly better dependencies on $\epsilon_i,i=1,2$ when $l\geq 2$. Therefore, we can conclude that in the setting of
> Section 5, ACE is strictly better than DML.
>
> We will add these arguments into the revised version of this paper.
>
> **Other minor issues.**
>
> We thank the reviewer for pointing them out and will fix them in the revision.
>
> [1]. Robins, James, et al. "Semiparametric minimax rates." Electronic journal of statistics 3 (2009): 1305.
>
> [2]. Balakrishnan, Sivaraman, Edward H. Kennedy, and Larry Wasserman. "The fundamental limits of structure-agnostic functional estimation." arXiv preprint arXiv:2305.04116 (2023).
>
> [3]. Jin, Jikai, and Vasilis Syrgkanis. "Structure-agnostic optimality of doubly robust learning for treatment effect estimation." arXiv preprint arXiv:2402.14264 (2024).

---

> > ### Comment · Reviewer_y6th · 2025-08-05
> >
> > Thanks very much for the clarifications, especially about Q1 and Q5. I am satisfied, and I will retain my score. Thanks!

---

### Official Review · Reviewer_9YPQ · 2025-06-26

**Clarity:** 2
**Significance:** 3
**Originality:** 3
**Rating:** 5
**Confidence:** 3

**Summary:**

This paper studies structure-agnostic estimation of a linear outcome model from Robinson [1988]. They explore the role of Gaussian treatment noise in this problem, first showing that double machine learning is optimal for Gaussian noise and then showing that it is (always) sub-optimal for non-Gaussian noise. They develop a new approach using cumulants that is more robust in the non-Gaussian settings.

**Questions:**

Line 94 and Line 107: What is \mathcal{X} in Defn 2.1. Did you mean \mathcal{Z}?

Can you comment on the relevance of non-Gaussian treatment noise in real-world settings?

**Ethical Concerns:**

["NO or VERY MINOR ethics concerns only"]

**Final Justification:**

My only real criticism was a lack of accessibility, which the authors addressed in their rebuttal with a nice explanation. I argue for acceptance.

**Limitations:**

yes

**Paper Formatting Concerns:**

No major formatting concerns

**Quality:**

3

**Strengths And Weaknesses:**

The results provide insight into the limitations of DML under non-Gaussian treatment noise. I am not 100% sure how likely non-Gaussian (and non-Binary) treatment is in real-world applications.

Some might say the empirical results are limited, but I do not believe this is a limitation because the contributions are theoretical.

The paper heavily references two papers from 2023 and 2024. I personally have not read either of those papers (and did not have time to review them during the review period). The paper is not as accessible as it could be. It would be nice to have a concrete example, such as a set of structural equations and a description of what a nuisance function (relative to the treatment effect) is and what different approaches would do to estimate the treatment effect. This might also help provide some intuition for the role of normality and why you can make improvements when treatment noise is not normal.

The labels of the figures are a little small. It would be easier to read them if you made them a little larger.

---

> ### Author Rebuttal · Authors · 2025-07-30
>
> We thank the reviewer for providing insightful comments and for appreciating our work. Below is our response to the weaknesses and questions raised by the reviewer.
>
> **Weakness 1. The paper is not as accessible as it could be. It would be nice to have a concrete example, such as a set of structural equations and a description of what a nuisance function (relative to the treatment effect) is and what different approaches would do to estimate the treatment effect. This might also help provide some intuition for the role of normality and why you can make improvements when treatment noise is not normal.**
>
> We thank the reviewer for this nice suggestion. In this paper, we consider a semiparametric model with partial linear outcome: $$Y = \theta T + f(X) + \epsilon, T = g(X) + \eta.$$ The nuisance functions are regression functions that can be estimated from data and are used in estimating the target causal quantity. In our setting, we want to estimate $\theta$. This can be done in several different ways. Firstly, given observational data $(X_i,T_i,Y_i), i=1,2,...,n$ and some nuisance estimate $\hat{q}(x)$ for $q(X)=\mathbb{E}[Y|X]$, [1, Section 1.1] derived an estimator
> $$\hat{\theta} = \left(\sum_{i=1}^n T_i^2\right)^{-1}\sum_{i=1}^n T_i(Y_i-\hat{q}(X_i)) .$$
> The error of this estimator is linear in the error of $\hat{q}$. This error rate can be further improved by DML:
> $$\hat{\theta} = \left(\sum_{i=1}^n (T_i-\hat{g}(X_i))^2\right)^{-1}\sum_{i=1}^n (T_i-\hat{g}(X_i))(Y_i-\hat{q}(X_i)),$$ where $\hat{g}$ is an estimate of $g$.DML is less sensitive to the errors of the nuisance estimates. Mathematically, this insensitivity means that the first-order functional derivatives of these nuisance functions vanish, as in our Definition F.1.
>
> More generally, if the functional derivatives up to the $k$-th order all vanish, then we can achieve a $(k+1)$-th order error rate, according to Taylor’s formula. Our work shows that this is indeed possible when the treatment noise is not normal. Our estimator is of the form
> $$\hat{\theta} = \left(\sum_{i=1}^n (T_i-\hat{g}(X_i))J(T_i-\hat{g}(X_i))\right)^{-1}\sum_{i=1}^n J(T_i-\hat{g}(X_i))(Y_i-\hat{q}(X_i)),$$
> where $J(\cdot)$ is some fixed function. Obviously, this is a generalization of the DML estimator. Our key insight is that there exists a $k$-th order polynomial $J$ such that $(k+1)$-th order insensitivity can be achieved. However, we also require that $E[(T-g(X))J(T-g(X))]\neq 0$, since otherwise the denominator would converge to zero. For the $J$ that we choose, the left-hand side is shown to be the $(k+1)$-th cumulant of the treatment noise $\eta=T-g(X)$. Hence, when $\eta$ is Gaussian, this non-zero condition can never be satisfied. On the other hand, when $\eta$ is non-Gaussian, there must exists some $k$ to make this approach work.
>
>
> **Weakness 2. The labels of the figures are a little small. It would be easier to read them if you made them a little larger.**
>
> We thank the reviewer for this helpful feedback. We will make them larger in a revised version.
>
> **Question 1. Line 94 and Line 107: What is \mathcal{X} in Defn 2.1. Did you mean \mathcal{Z}?**
>
> Yes, it should be \mathcal{Z}. We thank the reviewer for pointing out this typo.
>
> **Question 2. Can you comment on the relevance of non-Gaussian treatment noise in real-world settings?**
>
> In real-world settings, Gaussian treatments are rare except when a controlled study is conducted, and non-Gaussian treatments are therefore much more common. As a result, we believe that assuming non-Gaussian noise is quite reasonable in practice. A canonical example (detailed in Mackey et al. and simulated in our Sec. 6 experiments) is demand estimation from purchase and pricing data. Here, Y represents observed demand, the treatment T corresponds to an observed product price, $g_0(X)$ denotes a baseline product price determined by covariates X that influence pricing policy, and the treatment noise $\eta$ represents a random discount offered to customers for demand assessment. Notably, $\eta$ is typically discrete (and thus distinctly non-Gaussian) and independent of X.
>
> [1]. Chernozhukov, Victor, et al. "Double/debiased machine learning for treatment and structural parameters." The Econometrics Journal (2018): C1-C68.

---

> > ### Comment · Reviewer_9YPQ · 2025-08-01
> >
> > The explanation provided by the authors was helpful. In particular, I appreciated the explanation of the key difference that normality has in the treatment noise. I also now understand what is meant by a nuisance function. It may be because I am seeing this the second time now, but I found the explanation posted above easier to understand.
> >
> > I retain my positive opinion of the paper and argue for its acceptance.
> >
> > One question (which is more out of curiosity, so feel free to focus on other rebuttals): Would it be appropriate to consider non-Gaussian treatment noise to be the key characteristic of ``natural experiments"? Of course, economists usually look for step functions for these natural experiments, but I believe this is generalizing this notion.

---

> > > ### Author Response · Authors · 2025-08-07
> > >
> > > We would like to thank the reviewer again for appreciating our work, and for suggesting the connection between natural experiments and our findings.
> > >
> > > Yes, we believe that it is correct to consider non-Gaussian treatment as a new characteristics of "natural experiments", but in a slightly different sense. The key value of natural experiments is that it makes causal effects identifiable. In our setting, identifiability always holds but there is a difference in the statistical efficiency of estimating the causal parameter. We show that estimating the causal effect becomes intrinsically harder if the treatment is Gaussian. As the reviewer suggests, we view our findings as an interesting complement of existing understandings of "natural experiments". We thank the reviewer for this insightful comment.

---

### Official Review · Reviewer_ev6C · 2025-06-30

**Clarity:** 3
**Significance:** 3
**Originality:** 3
**Rating:** 4
**Confidence:** 1

**Summary:**

The statistical optimality of estimation methods for causal inference is an important research question. The authors contribute to the study of the partially linear model of Robinson (1988). The distribution of the treatment noise plays an important role: while for Gaussian noise the DML estimator achieves (minimax-) optimal estimation rates, for non-Gaussian noise better estimators can be derived. This paper makes this contrast precise by showing that no better estimator exists in the Gaussian case and that better structure-agnostic estimators can be constructed in the non-Gaussian case. Further, the authors give additional results for the binary treatment case and in the standard non-parametric model. A small set of empirical experiments are conducted to verify the proposed estimators.

**Questions:**

Please see the section above for questions.

**Ethical Concerns:**

["NO or VERY MINOR ethics concerns only"]

**Final Justification:**

I believe the paper is of high quality, providing interesting results that are however of incremental nature since the ACE estimators were known and some of the observations regarding their statistical properties in Gaussian and non-Gaussian settings had been made before.

**Limitations:**

Yes.

**Paper Formatting Concerns:**

No formatting concerns.

**Quality:**

3

**Strengths And Weaknesses:**

The paper is clear, well-written, and to the best of my knowledge introduces new results that contribute to our understanding of optimal estimation methods in popular causal inference models.

## Strengths
- The theoretical results do address a gap in the literature and lead to novel estimators.
- While the ACE estimators were known for l=1,2,3, higher order ones are novel and have been shown to provide an experimental gain.

## Weaknesses
- [Minor] The paper seems to study exclusively the partially linear model of Robinson. While it is relatively general it remains constrained and possibly detracts from the generality of the proposed estimators. Further it is was always clear to me which results rely on Robinson's model and which do not. For example, is Sec. 5.1 applicable to arbitrary non-parametric outcome model, i.e., for E[Y | X, T]?
- The experimental section is a little bit light. It might be useful to supplement the experiments with additional simulations or semi-synthetic scenarios.
- The conclusion could also be improved with a summary of the contributions and their significance.

## Additional comments
- Figure 1 (b) is a little bit difficult to parse as the boxes overlap and the means are not always easily visible.
- Figure 1 (a) suggests that we need a minimum number of samples to realize the gain from high-order estimation methods. Could the authors comment on this observation? Is there a tradeoff between performance and sample size?
- Since ACE estimators of different orders give slightly different results, do you have a good way of choosing between them. The numerical experiments suggest that the variance is a good indicator of the induced MSE; is this an empirical observation or is there some theoretical result to back this up?
- What would it take to extend the binary treatment results to categorical treatments?

## Minor comments
- Define ACE in the abstract.
- line 81, should the sentence be "... existing estimators that do assume a partially linear outcome model." instead of "... do not assume ..."?
- line 83, it might be useful to further comment on what the "standard non-parametric model" is exactly.
- Definition 2.1: should the $\mathcal X$ be $\mathcal Z$?

---

> ### Author Rebuttal · Authors · 2025-07-30
>
> We thank the reviewer for the positive and constructive feedback and are delighted that the reviewer found the paper clear and well-written and our contributions novel. We respond to each point raised below.
>
> **Weakness 1. The paper seems to study exclusively the partially linear model of Robinson. While it is relatively general it remains constrained and possibly detracts from the generality of the proposed estimators. Further it was always unclear to me which results rely on Robinson's model and which do not. For example, is Sec. 5.1 applicable to arbitrary non-parametric outcome model, i.e., for E[Y | X, T]?**
>
> The reviewer is correct that this paper focuses on the partial linear model. Hence, all upper and lower bounds established in this paper, including Section 5.1, is for this specific model.
>
> The partial linear model is still widely adopted in the economics literature, see e.g. [1,2] for a few applications. It is true that compared with the fully non-parametric model, the partial linear model is more restrictive since the treatment effect is required to be homogeneous. Extending the results of this paper to the fully non-parametric model is an important future direction. It is worth noticing that even for the partial linear model, DML is minimax optimal in a structure-agnostic sense, as we prove in Theorem 3.1. Our cumulant-based estimator is novel, and to the best of our knowledge, no existing estimators are constructed based on similar ideas. Therefore, we believe that our approach may inspire extensions that have broader real-world applicability.
>
> **Weakness 2. The experimental section is a little bit light. It might be useful to supplement the experiments with additional simulations or semi-synthetic scenarios.**
>
> We thank the reviewer for pointing out this limitation. Currently, our experiment section is merely a proof-of-concept to show the effectiveness of our proposed approach. We will extend the experiment section to provide (1) more insights on how the estimation error depends on the cumulant and (2) comparison with other estimation strategies.
>
> **Weakness 3. The conclusion could also be improved with a summary of the contributions and their significance.**
>
> This is also a great suggestion. We will rewrite the conclusion part to make it a better summary of the contributions of this paper.
>
> **Comment 1. Figure 1 (b) is a little bit difficult to parse as the boxes overlap and the means are not always easily visible.**
>
> We thank the reviewer for pointing this out. We will make the means more visible in the revised version. The main purpose of this figure is to show that the standard DML estimator incurs a much larger bias than ACE estimators.
>
> **Comment 2. Figure 1 (a) suggests that we need a minimum number of samples to realize the gain from high-order estimation methods. Could the authors comment on this observation? Is there a tradeoff between performance and sample size?**
>
> The reviewer correctly identifies a tradeoff between performance and sample size. According to our Theorem 5.1, as $l$ increases, the bound would have a better dependency on nuisance errors $\epsilon_i, i=1,2$, but the constant becomes worse. Therefore, it outperforms standard DML, which corresponds to the case $l=1$, only when the nuisance errors $\epsilon_i, i=1,2$ are sufficiently small.
>
> **Comment 3. Since ACE estimators of different orders give slightly different results, do you have a good way of choosing between them? The numerical experiments suggest that the variance is a good indicator of the induced MSE; is this an empirical observation or is there some theoretical result to back this up?**
>
> As the reviewer points out, we used variance as a good indicator of the induced MSE. That is because in our example, the variance dominates the bias for estimators with order of orthogonality $l \geq 2$. Moreover, we do not have a good estimation of how large the bias could be. Therefore, our practical recommendation is that
> - When the sample size is small, only consider using ACE with $l=2$.
> - When the sample size is large and we believe that the nuisance function belongs to some benign function classes (e.g. linear functions with high sparsity), then theoretically, bias for $l\geq 2$ is likely to decay faster than variance. In this case, we recommend comparing the variance for $l\leq 5$ and pick the one with the smallest variance.
> - When the sample size is large and the nuisance functions are not so well-behaved, then bias would asymptotically dominate for small $l$. In this case, we recommend setting an upper threshold for variance (e.g. a constant multiple of the variance for $l=2$), and pick the largest possible $l\leq L_{max}$ with the estimated variance below that threshold.
>
> **Comment 4. What would it take to extend the binary treatment results to categorical treatments?**
>
> We suspect that, with a bit more effort, we could establish a similar lower bound for categorical treatments. The upper bound has been studied in a number of existing works. For example, [3] introduces a DML estimator with a group Lasso selection process for the propensity score. [4] studies an application in program evaluation. The main challenge in this setting lies in choosing a good model to fit the nuisances, and theoretical dependencies on nuisance errors are the same as in the binary treatment setting.
>
> **Minor Comments**
>
> We thank the reviewers for providing these comments. In line 81, it should be “do not” rather than “do”. There we wanted to emphasize that existing lower bounds do not apply to the partial linear model, since they are established for the standard non-parametric model, i.e., $E[Y|T,X] = q(D,X)$ where $q$ does not have a parametric form.
>
> [1]. Chay, Kenneth Y., and Michael Greenstone. "Does air quality matter? Evidence from the housing market." Journal of Political Economy 113.2 (2005): 376-424.
>
> [2]. Hull, Isaiah, and Anna Grodecka-Messi. "Measuring the impact of taxes and public services on property values: a double machine learning approach." arXiv preprint arXiv:2203.14751 (2022).
>
> [3]. Farrell, Max H. "Robust inference on average treatment effects with possibly more covariates than observations." Journal of Econometrics 189.1 (2015): 1-23.
>
> [4]. Knaus, Michael C. "Double machine learning-based programme evaluation under unconfoundedness." The Econometrics Journal 25.3 (2022): 602-627.

---

> > ### Comment · Reviewer_ev6C · 2025-08-05
> > **Response**
> >
> > Thank you for your rebuttal. It has largely addressed my remaining concerns. I maintain my score.

---

### Decision · Program_Chairs · 2025-09-17

**Decision:**

Accept (poster)

**Comment:**

This paper studies partially linear model of Robinson, and proves that DML achieves optimality under Gaussian noise assumption. When using non Gaussian noise assumption, the paper shows structure-agnostic causal estimator can achieve better results.

Strengths:
-- The paper studies a very important problem, considering the wide usage of DML in causal inference.
-- The paper provides solid theoretical justification.
-- Empirical results are also given.

Weaknesses:
-- There were some complains about the paper is not that accessible, which has been addressed in the rebuttal,

The reasons for accept: a good theoretical paper in causal inference.

All the reviewers have actively participated in the discussion. One reviewer raised their score. The main concerns were about the readability and the reviewers are happy about the clarification made by the authors in the rebuttal.